## Registered report

psychology

COVID-19, nursing home worker, care home worker, mental health, post-traumatic stress disorder, anxiety

**Author for correspondence:**
Elena Rusconi
e-mail: elena.rusconi@unitn.it

# Prevalence of post-traumatic symptomatology and anxiety among residential nursing and care home workers following the first COVID-19 outbreak in Northern Italy

Marianna Riello[1,2], Marianna Purgato[3], Chiara Bove[1], David MacTaggart[4] and Elena Rusconi[1]

[1]Department of Psychology and Cognitive Science, University of Trento, Corso Bettini, 31, 38068 Rovereto (TN), Italy
[2]Gruppo SPES, Trento, Italy
[3]Department of Neurosciences, Biomedicine and Movement Sciences, University of Verona, Verona, Italy
[4]School of Mathematics and Statistics, University of Glasgow, Glasgow, UK

ER, 0000-0003-2700-205X

The current COVID-19 pandemic has been officially linked to the deaths of hundreds of thousands of people across the globe in just a few months. It is particularly lethal for the elderly in general, as well as for populations residing in long-term stay facilities. By this time, those working and caring for high-risk populations have been exposed to very intense and sudden levels of physical and psychological strain. The situation has taken a particularly tragic turn in residential nursing and care homes (NCH), which were hit hard by the pandemic. In residential NCH, neither residents nor workers tend to have immediate access to the same expertise, medication and equipment as in hospitals, which exacerbates an already tense situation. Among the mental health conditions related to exposure to potentially traumatic events, post-traumatic stress disorder and anxiety are the most prevalent and scientifically recognized. In this survey-based epidemiological study, we test the prevalence of anxiety and post-traumatic symptomatology in residential nursing and care home workers—a group of individuals that has been largely neglected but who nonetheless plays a very important and sensitive role in our society. We do this by focusing on the North of Italy, the most affected region during the first COVID-19 outbreak in Italy. Using a single-stage cluster design, our study returns an estimate for the prevalence

of moderate-to-severe anxiety and/or post-traumatic symptomatology of 43% (s.e. = 3.09; 95% CI [37–49]), with an 18% (s.e. = 1.83; 95% CI [14–22]) prevalence of comorbidity among workers of Northern Italian NCH between 15 June and 25 July 2020 (i.e. 12–52 days after the end of national lockdown). Women and workers who had recently been in contact with COVID-19-positive patients/colleagues are more likely to report moderate-to-severe symptoms, with odds ratios of 2.2 and 1.7, respectively.

# 1. Introduction

The novel severe acute respiratory syndrome coronavirus-2 (SARS-CoV-2), otherwise known as the novel coronavirus (2019-nCoV), was first reported in December 2019 as a cluster of atypical viral pneumonia cases in the city of Wuhan, China [1]. In February 2020, the Word Health Organization (WHO) named the disease caused by SARS-CoV-2 the 2019 Coronavirus disease (COVID-19) and in March 2020 declared the outbreak a pandemic [2]. COVID-19 has been rapidly transmitted to Italy, and has been linked to high death rates, especially in elderly people. According to recent statistics from the Population Reference Bureau, Italy has the second-largest percentage of older adults (65+) in the world [3]. In 2020, almost one-fourth of the total population in Italy was estimated to be aged 65 years and older, with elderly people in Italian society growing constantly [4,5]. According to official data released by the Italian Ministry of Health [6], the number of deaths for COVID-19 in Italy went from 10 to 32 007 between 25 February and 18 May 2020. This represents 13% of the individuals who tested positive for the presence of the virus (see figure 1 showing a breakdown of death rates by age group as of 15 May) [6]. At least 86% of those deaths were located in the North of Italy [3].[1] The higher death rates connected with age are possibly due to a weaker immune system in the elderly that permits the faster progression of viral infection [8,9]. Additionally, younger patients tend to be prioritized, when there are limited resources, over older patients for access to intensive care units, equipment and drugs in hospitals [10]. Other potential factors involved in increasing the vulnerability of elderly individuals are the presence of concomitant physical illnesses associated with advanced age and the relatively confined space in which many of them live, such as residential nursing and care homes (NCH)[2] [11].

The latest national survey conducted by Istituto Superiore di Sanità (ISS) on the COVID-19 health emergency in NCH describes the scenario across 1082 public and private NCH (33% of the NCH contacted) from February to 14 April 2020 [12]. On the whole, the participating NCH hosted a total of 80 131 residents on 1 February, with an average of 74 residents each. During the period covered by this survey, a total of 6773 residents died, of whom 2724 (that is 40.2% of all the deceased) either tested positive for COVID-19 or showed COVID-19 compatible symptoms (swab policies differed across the territory). The majority of these deaths (91%) was concentrated in the Northern region, which was also the most affected by the COVID-19 contagion. On top of this, a number of healthcare and administrative staff working in NCH tested positive for COVID-19 [12].

All staff (i.e. including healthcare, administrative, technical and professional) working in NCH have been under a particularly intense physical and psychological pressure since the epidemic outbreak. The pressure is mounting from several directions. First, in NCH, elderly people with or without disabilities live close to each other and tend to form close bonds with all staff (including top management [13]), thus the effects of the COVID-19 emergency may become particularly taxing and difficult to manage. The elderly and/or those with concomitant physical illnesses are at greater risk of death or serious COVID-19 outcomes. In this context, even workers are more exposed to COVID-19 infection—either because they work directly with residents or because they have to liaise with those who work with residents—than the general population [14].

Second, administrative staff (including nursing directors and administrators [15]) have the extra responsibility to implement novel security protocols among workers (e.g. to maintain social distancing and use biosecurity equipment). Moreover, they have to manage communications with families of residents, who were not allowed to visit their relatives during Phase 1 (in Phase 1 the

---

[1]The North of Italy includes the following areas: Liguria, Lombardia, Piemonte, Valle d'Aosta, Emilia-Romagna, Friuli-Venezia Giulia, Trentino-Alto Adige, Veneto. Forty-six per cent of the Italian population resides in the North [7].

[2]In Italian, these are subsumed under the label of *strutture residenziali socio-assistenziali e socio-sanitarie per anziani*.

| age range | deceased | % mortality | |
|---|---|---|---|
| 0–9 | 3 | 0.0% | 0.2% |
| 10–19 | 0 | 0.0% | 0.0% |
| 20–29 | 12 | 0.0% | 0.1% |
| 30–39 | 60 | 0.2% | 0.4% |
| 40–49 | 259 | 0.9% | 0.9% |
| 50–59 | 1 069 | 3.6% | 2.7% |
| 60–69 | 3 135 | 10.5% | 10.4% |
| 70–79 | 8 258 | 27.6% | 25.5% |
| 80–89 | 12 186 | 40.8% | 30.9% |
| > 90 | 4 901 | 16.4% | 27.3% |
| unknown | 1 | 0.0% | 1.9% |
| total | 29 884 | 100.0% | 13.4% |

**Figure 1.** Official mortality rates by age group (percentage of people who died compared with the total registered positive cases in a given age group) as of 15 May 2020 [6]. About 94% of the deceased were aged 60 or over. Data source: Istituto Superiore di Sanità (ISS).

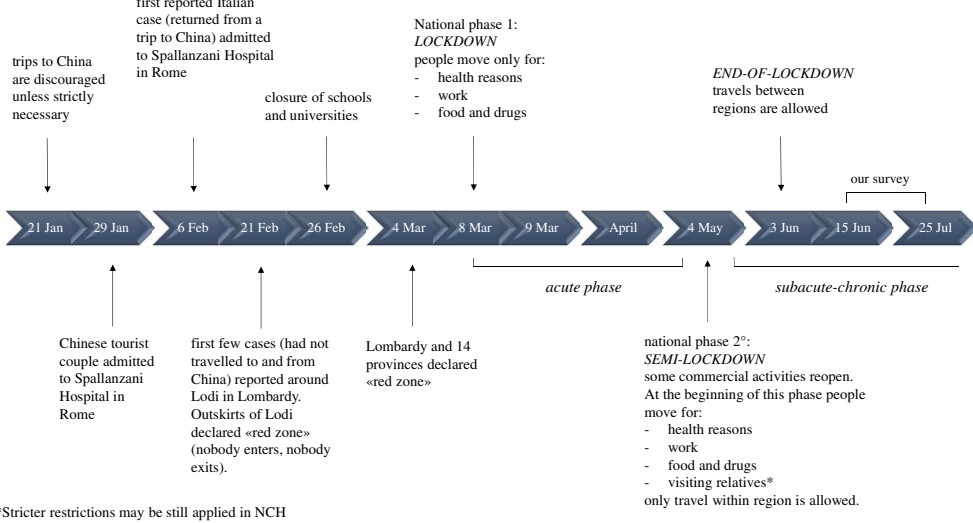

*Stricter restrictions may be still applied in NCH
°During this phase some measures will become more lenient, e.g. from the 18 May museums and parks are allowed to reopen.

**Figure 2.** Milestone events in the recent COVID-19 outbreak in Italy. From the beginning of March to 15 May 2020 the number of confirmed cases increased dramatically from 1694 to 223 885 [6]. According to official data from the ISS as of 15 May, COVID-19 had been related to 29 884 deaths (i.e. a mortality rate of about 13%) and 94% of the deceased were people aged above 60 [6]. We collected our data during the subacute-chronic phase, between mid-June and the end of July 2020. Although the country has entered Phase 2, more restrictive rules continue to be applied in NCH, where residents are particularly vulnerable to COVID-19.

Italian government implemented very strict lockdown rules with the possibility of leaving one's home only for health reasons, work duties or to buy food and drugs) and may be allowed to visit them with restrictions during Phase 2, in which the lockdown rules are becoming less strict (figure 2) [16].

Third, the Centers for Disease Control and Prevention (CDC) guidelines for preventing the spread of COVID-19 in retirement communities [16] recommended cancelling group activities. However, the elimination of social interactions may increase the risk of adverse mental health outcomes during a stressful event such as a disease outbreak. Therefore, for the group activities that are deemed as absolutely essential, CDC recommended introducing social distancing and limiting the number of attendees. This has had inevitable consequences for the management of residents. The imposition of keeping at a distance while wearing protective masks, for example, disrupts communications with

residents, in particular with those having cognitive or sensory impairments. Because lip-reading cues cannot be used by residents, interactions are more laborious and workers often have to resort to gestures and miming, to convey a message [17].

Fourth, all staff members are dealing with an unusually fast staff turnover because of the rapid transmission of the virus. This turnover implies novel interactions between existing staff members and new staff, who are either recruited from a variety of clinical/technical settings or have just graduated from healthcare institutions and universities.

Finally, staff in NCH reported a lack of information about procedures to contain the infection, a lack of medication and of biosecurity equipment, staff shortage, difficulties in transferring COVID-19 patients to hospitals and in isolating patients affected by COVID-19 from the others within NCH. They also reported the lack of any written standard operating procedure to contain and manage the virus [14]. In response to the reported difficulties, temporary guidelines for the prevention and control of SARS-CoV-2 infection in residential structures have now been published by the ISS [18].

In addition, family visits which represent a cornerstone of the daily management of patients in NCH were prohibited during Phase 1 (figure 2). In a number of centres, staff reported a lack of communication devices for patients to interact with family members/carers in the absence of in-person visits (e.g. video calls) [14]. Staff from other centres reported difficulties in communicating, remotely, health information to family members of COVID-19 patients, especially with regard to deaths that occurred quickly. The unusually high frequency of deaths, the absence of prompt communication from NCH with patients' families and the high rate of infection among staff and residents have led to several NCH currently being under investigation [19]. Furthermore, with the opening of Phase 2 and the subsequent increase of people's mobility (figure 2), the concerns for a possible contagion have increased and, consequently, so has the apprehension of becoming a vehicle for the virus inside NCH. We can thus expect the enormous amount of pressure that has been mounting in NCH during Phase 1 (i.e. between 8 April and 3 May 2020) to persist for quite some time.

In other countries, which have experienced a peak of contagion with some delay compared with Italy, it has only recently emerged that the situation in NCH may be much more critical than previously known. The UK government, for example, only recently updated its official death toll figures by including data from NCH, causing its known death toll to rise higher than that of any other country except the USA at the time [20].

The complex humanitarian emergency described above may generate increased levels of distress, fear of contamination and psychological suffering related to compassion towards at-risk patients. These psychological reactions are in line with those documented in the 2003 SARS outbreak that included the fear of infecting family members, friends and colleagues; the feeling of uncertainty and stigmatization and high levels of stress, anxiety and depression symptoms [21–23]. In the long-term, these issues might become clinically relevant, leading to a diagnosis of mental health disorder [24].

Among the mental health conditions related to the exposure to potentially traumatic events, post-traumatic stress disorder (PTSD) and anxiety are the most prevalent and scientifically recognized. Characteristic symptoms of PTSD include re-experiencing phenomena such as nightmares and recurrent distressing memories of the event(s), the avoidance of internal and external reminders of the trauma, negative alterations to thoughts and mood and hyperarousal symptoms including sleep disturbance, increased irritability and hypervigilance [25,26]. The US National Co-morbidity Survey found that 7.8% of 5877 adult respondents had PTSD at some time in their lives [27]. In individuals who had been exposed to traumatic events in humanitarian settings, rates of PTSD and anxiety symptomatology may reach a prevalence of 30–40% (according to the type and amount of stressors [28–32]). However, precise information on the time since trauma exposure, that could heavily impact symptom intensity [33], is often not available from the current literature, especially in humanitarian situations where people are exposed to multiple complex traumatic experiences. When PTSD is untreated, people can still suffer from the condition at least 10 years after the critical event [34]. Furthermore, and according to available data from populations exposed to potentially traumatic events, PTSD may generate medical retirement from work, changes in work volition and coping strategies [35], and difficulties in interacting with colleagues [36,37]. It has recently been stressed that trauma awareness, post-trauma social support and stressors are strongly related to a longer-term mental health status [38]. Generalized anxiety disorder (GAD) is characterized by the excessive worry about everyday events and problems to the point at which the individual experiences considerable distress in performing day-to-day tasks. In the general population, the lifetime prevalence of GAD is 5.1%, with increased values for individuals exposed to specific risk factors, i.e. potentially traumatic events [39]. Comorbidity is known to be a highly prevalent feature of

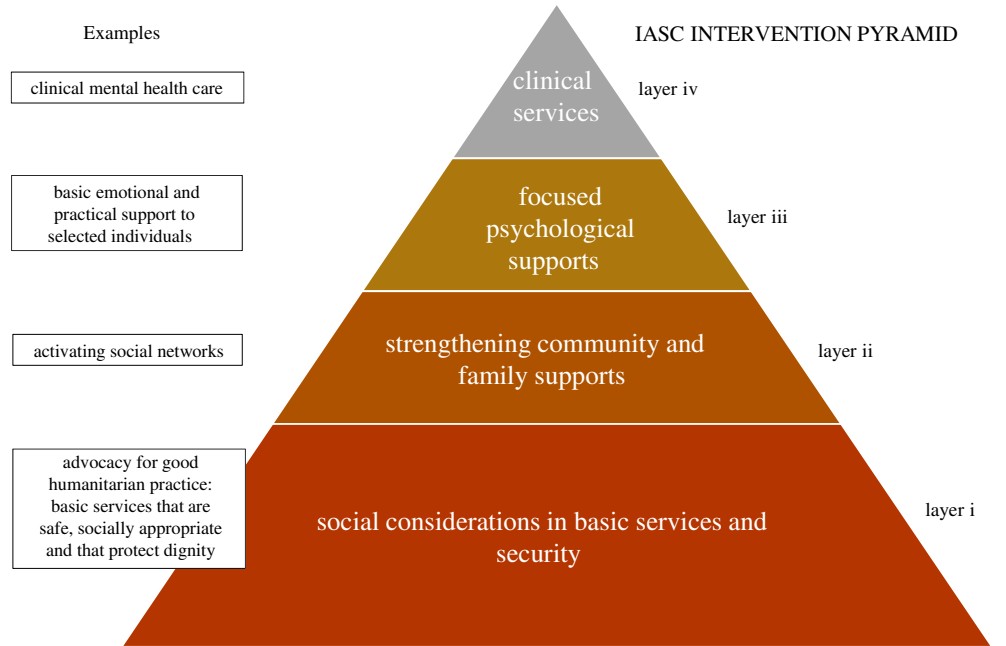

**Figure 3.** IASC pyramid of mental health and psychosocial support interventions in emergency settings [42]. Interventions across layers are intended to be complementary and not mutually exclusive. Interventions located at the top two levels of the pyramid (layers iii and iv) are the most specialized and focused on populations with high levels of symptomatology and/or mental disorders (e.g. psychotherapies, focused psychosocial interventions delivered by mental health specialists as psychologists). At the base of the pyramid, there are basic services and security (layer i), and general community support (layer ii), that are mainly focused on reinforcing and activating social networks and support [43].

anxiety, with up to 90% of patients with anxiety disorder showing concomitant symptoms of depression, dysthymia or other psychological symptoms [39]. PTSD and anxiety symptomatology will be considered in one prevalence index to capture a more comprehensive estimate of the psychological state of participants. Available studies on this topic, for example, found high levels of PTSD symptoms and lower levels of anxiety during the emergency peak [40], while in other studies conducted months after the acute emergency levels of anxiety were higher [22].

To date, despite the enormous burden of distress and potentially traumatic events experienced by people who work in NCH, no studies have documented the prevalence of mental health problems in this population group. Against this background, the aim of the present work is to examine the prevalence of self-reported PTSD symptomatology and anxiety in staff working in NCH in the immediate aftermath of the COVID-19 outbreak.

In the current literature void, these data will be a useful initial inroad into forming a more detailed representation of this population of workers. The knowledge of detailed socio-demographic and clinical characteristics of this population group has the potential added value of indirectly informing the choice of the intervention category (not the specific intervention type) according to the Inter-Agency Standing Committee (IASC) pyramid of mental health and psychosocial support interventions in emergency settings [41] (figure 3), and long-term follow-up assessments for NCH workers. For example, NCH workers with lower levels of PTSD and/or anxiety symptomatology might receive basic and/or low-intensity psychosocial interventions placed at the base of the pyramid (layers i and ii), while individuals with higher levels of PTSD and/or anxiety symptomatology might receive structured psychotherapeutic interventions according to resource availability at local level; (iv) layer of the pyramid on the top). Psychological interventions covering the areas of anxiety and PTSD symptomatology fall in the upper two layers of the IASC pyramid (layers iii and iv) and might include online psychotherapeutic interventions to help staff to deal with psychological problems, and online psychological group or individual activities to release stress [44]. Mental health front desks and helplines can be established to enable workers to receive personalized assistance by telephone or the internet. This approach is in line with the WHO mhGAP humanitarian intervention guide [45].

## 1.1. Research question

The research question addressed by this study is:

What is the prevalence of severe-to-moderate self-reported mental health symptoms of anxiety and/or PTSD among staff in NCH during the first COVID-19 outbreak in Northern Italy?

A respondent will be deemed to have moderate-to-severe symptoms if he or she reaches a set threshold in one of the two scales adopted for this survey: a score greater than or equal to 10 on the 7-item generalized anxiety disorder scale (GAD-7; see §2.2.2 for further details) and/or greater than or equal to 26 on the 22-item impact of event scale-revised (IES-R; see §2.2.2 for further details). The answer to this research question will help fill a gap in the current literature and provide much needed data on staff working in NCH.

# 2. Material and methods

## 2.1. Participants

### 2.1.1. Power analysis and sample size rationale

To estimate a suitable sample size, we will first make the simplifying assumption of simple random selection (SRS). For a large population, the required expression is

$$n_{\text{base}} \geq \frac{z_{\alpha/2}^2 p(1-p)}{\delta^2},$$

where $z_{\alpha/2} = 1.96$ for a probability of 0.95, $p$ is the proportion (prevalence) and $\delta$ is the margin of error. Based on comparable literature on healthcare workers (see table II of [40]), we expect a prevalence of $p = 0.35$.

The minimum sample size needed to estimate this proportion to within 0.05 at a probability of 0.95 is $n_{\text{base}} \approx 350$. Our survey design, however, involves clustering and weighting, so SRS is only a starting point. Although our calculation of the proportion estimate will not be affected strongly by the nature of our survey design, the estimated variances and standard errors can be. Therefore, $n_{\text{base}}$ is likely to under-represent the actual sample size that we require. In order to account for this, we need to consider the design effect $D^2$ for our survey. Following [46], the design effect can be expressed as

$$D^2 = 1 + f(L_{\text{weight}}, L_{\text{clust}}),$$

where $f$ is some function of the loss due to clustering $L_{\text{clust}}$ and the loss due to weighting $L_{\text{weight}}$. If $f = 0$ then we have SRS. For our study, we will follow [47,48] and approximate the design effect as

$$D^2 \approx (1 + cv^2)[1 + \rho(n_c - 1)],$$

where $cv^2$ is the relvariance (variance/squared mean) of the applied weights, $n_c$ is the cluster size and $\rho$ is the intracluster correlation.

The influence of weighting on the design effect can be determined using the following expression for the relvariance,[3]

$$cv^2 = \frac{n \sum_{i=1}^{n} w_i^2}{\left(\sum_{i=1}^{n} w_i\right)^2} - 1,$$

where $n$ is the sample size and $w_i$ is the weight of the $i$th element in the sample. An *a priori* estimate of $cv^2$ is difficult due to the lack of similar studies related to NCH. For the weights that we will apply (see §2.3.1), we will assume a typical estimate [47] of $cv^2 = 0.1$, although the value from our study may be much smaller.

The quantity $\rho$ is related to the rate of homogeneity within clusters [46] and depends on the characteristics of the population under study. Similar to the weighting loss estimate, there are very few suitable data related specifically to NCH that would allow us to make an accurate estimate of $\rho$. Based on studies measuring mental health outcomes and values indicated as typical in the specialized

---

[3]While preparing the Stage 2 manuscript we spotted a typographical error in the previous relvariance formula: this has now been rectified and the current manuscript reports the correct formula for relvariance [49]. None of the previous calculations has been affected by the typographical error.

literature [50–53] we will assume a rate of homogeneity of 0.05. Together with a cluster size of $n_c = 25$ and $cv^2 = 0.1$, our effective sample size $n_{\mathrm{eff}}$ is

$$n_{\mathrm{eff}} = D^2 n_{\mathrm{base}} = 847.$$

In our study, we will aim to exceed this value and seek a minimum sample size of 900. This would afford us an estimate of population parameters with the same level of confidence and margin of error up to a prevalence of 0.41. For any prevalence inferior to 0.41 or superior to 0.59, the minimum sample size would be inferior to 900. If the prevalence turned out to be (unexpectedly) between 0.41 and 0.59, the minimum required sample would be between 900 and 930. In any case, if we were unable to reach the minimum sample size, the data will be still reported, with an adjusted margin of error. For example, if the actual prevalence turned out to be 0.35 and we only received data from about 590 respondents, the margin of error of our estimate would increase from 0.05 to 0.06.

The design effect corresponding to the above parameters is 2.42. Another influence of a design effect greater than 1 is to amplify the standard error (s.e.) by a factor $D$. This, in turn, will increase the size of 95% confidence intervals compared with assuming only SRS.

One potential outcome of our survey will be to estimate $D^2$ (and, by extension, $cv^2$ and $\rho$) for use in future studies related to NCH. Such estimates can be found, for example, via

$$D = \frac{\text{s.e.}_{\mathrm{complex}}}{\text{s.e.}_{\mathrm{SRS}}},$$

where s.e.$_{\mathrm{SRS}}$ is the standard error assuming only SRS, and s.e.$_{\mathrm{complex}}$ is the standard error due to the inclusion of extra sampling effects, e.g. clustering. These estimates will be a useful output from our study but remain secondary to the main output of assessing the prevalence of anxiety and/or PTSD symptomatology in NCH.

## 2.2. Study design and materials

### 2.2.1. Design and recruitment

The study has been approved by the clinical research ethics committee of the Servizi Pastorali Educativi Sociali (SPES) Group after consultation with trade union representatives, and it follows the American Association for Public Opinion Research reporting guideline [54]. In this survey-based cross-sectional epidemiological study, we collected demographic data and information on mental health symptoms in Northern Italian NCH. Our primary focus was on establishing the prevalence of mental health symptoms in workers in the immediate aftermath of the first COVID-19 outbreak. To this aim, we adopted a single-stage cluster sampling from 15 June to 25 July 2020. Milestone events about the current COVID-19 outbreak in Italy are shown in figure 2. Notably, restrictions imposed within NCH are likely to remain stricter than those imposed on other work environments due to residents' high vulnerability to COVID-19.

Our sample was taken from the worst-hit geographic location (the North of Italy) during the recent COVID-19 outbreak. According to a recent ISS report, about 72% of all existing NCH (3420) are concentrated in Northern Italy. The same document reports that, in a sample of 1082 NCH (representing 33% of the 3276 NCH contacted) mostly based in the North, there are on average 48 healthcare and technical workers (with a large preponderance of healthcare workers) per single nursing home. This count does not include administrative staff, which we can estimate to represent about 10% of the staff [55], thus bringing the total to 53. For studies conducted on individuals working in healthcare organizations it is possible to expect average response rates of about 54% [56]. Adopting an estimate of a 48% response rate, we can expect to collect about 25 responses on average per single nursing home. Assuming a 30% response rate from top management in Italian NCH [12], we aimed to sample in the Northern region a minimum of 120 NCH whose contact details are publicly available via online databases [12].

To test feasibility and negotiate a format that would be seen as acceptable, minimally intrusive and even helpful to NCH, we preliminarily discussed our plans with the top management of a nursing home in Trento. They already agreed to circulate the survey among their workers once we were ready to commence the study. We have excluded from the study the top manager and those few workers (an administrator, a doctor, a physiotherapist, a psychologist, a lawyer, a nurse) which had been consulted for an opinion on the appropriateness of the survey at this stressful time. The top manager of one other nursing home in Verona, personally known to the authors, had already seen the survey and

expressed an interest in participating. Because no workers were involved in the decision only the top manager has been excluded in that case. We judged it necessary to undertake these preliminary steps to test and evaluate the likelihood of our reaching the target sample size for this registered report. Other NCH were randomly selected by assigning them a number with the rand() function in Excel, and approached in ascending order, by email and phone, by the researchers.

Within those NCH that decided to enroll in the study, the sampling was exhaustive. All administrative, medical/healthcare, technical and professional staff was invited via email to fill an anonymous survey. Administrative staff included: directors, nursing coordinators, administrative collaborators or assistants and secretaries. Medical/healthcare staff included: physicians, nurses, healthcare auxiliary staff, physiotherapists, experts in psychiatric rehabilitation, speech therapists and psychologists. Technical staff included: educators, entertainers, mediators, caseworkers, trainers, sociologists, specialized auxiliaries, technicians for the maintenance of the building and cleaning staff. Professional staff included: lawyers and religious assistants. The invitation message contained a link to the survey and informed the individual that, by completing the survey, he or she would contribute to a prevalence study on mental health conditions of workers in NCH.[4] Informed consent was requested from all participants prior to completing the online survey and participants were allowed to terminate the survey at any time. The survey was anonymous and no information was requested that may have led to the identification of single respondents.

## 2.2.2. Materials

We focused on symptoms of anxiety and PTSD for all participants using an online version of validated measurement tools: the 7-item GAD (GAD-7; range of total scores 0–21; [58,59]) and the 22-item impact of event scale-revised (IES-R; range of total scores 0–88; [60]) were used to test the presence and severity of symptoms of anxiety and PTSD respectively. The GAD-7 has good internal consistency (Cronbach's alpha = 0.92), test-retest reliability (0.83), procedural validity (0.83 between administered and self-report versions), and reaches very good values of sensitivity and specificity (89% and 82%, respectively). It convergences with other well-known and longer tests for anxiety (e.g. the Beck Anxiety Inventory and the Symptom Checklist-90) [59]. The IES-R has also been shown to have good internal consistency (Cronbach's alpha = 0.95), test-retest reliability ($r = 0.89$), sensitivity and specificity of 74.5% and 63% respectively and a good correlation with other validated scales (e.g. the PTSD Symptom Scale) [60–62].

Despite the availability of several other reliable instruments to assess the clinical psychological symptoms of participants, for example, the diagnostic MINI Neuropsychiatric Interview [63], we selected the GAD-7 and IES-R for feasibility issues, and in consideration of their psychometric properties. The GAD-7 and IES-R are short and easily self-administered. Psychometric evaluations of the GAD-7 suggest that it is a reliable and valid measure of GAD symptoms in both psychiatric [64,65] and general population [66,67] samples. In particular, the GAD-7 is used as a tool to capture an overall clinical picture and guide treatment planning. It measures the severity of anxiety symptoms following DSM-IV criteria A, B and C, for GAD [68]. Scores above 10 are considered to be in the clinical range [59]. The GAD-7 has shown good reliability and construct validity [58]. The IES score is associated with psychological and physical health following traumatic events. Weiss & Marmar [60] developed a new version of the scale (IES-R) to include a measure of hyperarousal, a core component of PTSD. This scale is not diagnostic for PTSD but it has been used to measure the subjective response to specific traumatic events, such as earthquakes or wars, in an adult population [69–71]. The IES-R instrument has been translated and validated in different studies [72,73] recruiting different population groups. There are several translations of this self-report scale, which have shown good psychometric properties [69,74,75]. Moreover, both scales were used in recent epidemiological studies on mental health among healthcare workers during the COVID-19 outbreak in China [40].

The GAD-7 measures the frequency and severity of GAD symptoms with reference to the last two weeks, using seven 4-point Likert-scale items with responses ranging from 0 (not at all), 1 (several

---

[4]The individual was informed that, based on a selection of her/his answers, s/he would receive a brief immediate feedback on her/his well-being state. The feedback built on the answers given to the final part of the survey (a short questionnaire on general well-being), which is not of clinical relevance and was not considered for the purpose of the current study. This was included, on request from the local ethics committee, as a way to provide respondents with advice to seek help if their well-being appears at risk, while respecting their anonymity and reserving the main part of the survey for scientific research purposes. Indeed, most of the workers appeared quite unaware of obvious signs of physical and psychological distress, as they were not only to continue working and but also deal with an increased workload during this emergency [57].

days), 2 (more than half the days) to 3 (nearly every day). Total scores range from 0 to 21. The IES-R assesses post-traumatic stress with reference to the last 7 days, using 22 5-point Likert-scale items with responses ranging from 0 (not at all), 1 (a little bit), 2 (moderately), 3 (quite a bit), to 4 (extremely). Total scores range from 0 to 88. Only scales with all items completed were considered and entered in the study for statistical analysis. Total scores were interpreted as follows: GAD-7, normal (0–4), mild (5–9), moderate (10–14) and severe (15–21) anxiety; IES-R, normal (0–8), mild (9–25), moderate (26–43), severe (44–88) PTSD symptomatology. These categories are based on values established in the literature [40,59,66,76–78].

We also collected self-reported demographic data from respondents including: age (18–25, 26–30, 31–40, 41–50, 51–60, ≥61), gender (male or female), education level (primary school, middle school, high school diploma or university degree), job title (administrative, healthcare, technical, professional, other; see §2.2.1 for more detail on the roles included in the first four categories). Information about the geographical location of a respondent's nursing/care home was also collected to make sure that we could exclude all those outside of the Northern region. We asked respondents to indicate how many days out of 7 they worked during the last week and also during the week before last, whether in the last two weeks they came directly in contact with COVID-19-positive colleagues or patients, if they had access to personal protective equipment at work, and whether visits from family members were suspended during the previous fortnight.

As a quick check for inattentive or careless responses, we added to our survey three extra questions (one at the end of the demographic questionnaire, one at the end of the GAD-7, one at the end of the IES-R) similar in spirit to those found in the Information section of the Italian Wechsler Memory scale [79]. These are very simple questions (Who is the current president of the United States of America? What is the capital of Italy? What is the name of the current Pope?). If a respondent did not answer correctly to at least two of the three questions, their data were excluded. We also used IP/device tracking, to detect and prevent duplicate responses (no more than one response could be sent from the same device). Participants were contacted by the researchers through their organization, they were requested to avoid environmental distraction when completing the survey and to possibly use their personal rather than work devices (as a further measure to preserve anonymity). Given the level of commitment required, the specialist/professional tuning of the demographic questionnaire and the circulation of the survey from within the organizational community, we did not expect a significant number of responses outside of the intended population.

## 2.3. Statistical analysis

### 2.3.1. Pre-registered data analysis

Our main output from this study is an estimate of the prevalence of moderate-to-severe symptoms of anxiety and/or PTSD in Northern Italian NCH, which may be a starting point for the implementation of effective treatments (table 1). As a reminder, a respondent was deemed to have 'moderate-to-severe' symptoms if he or she reached a set threshold in one of the two scales adopted for this survey: a score greater than or equal to 10 for GAD-7 and/or greater than or equal to 26 for IES-R.

Once all of the data were collected, before proceeding to analyse them, we excluded the following:

— incomplete surveys (i.e. surveys without complete responses to the demographic AND the GAD-7 AND the IES-R questionnaires);
— surveys in which two or more of the inattentive/careless response check items were not responded to correctly;
— surveys in which the job role was indicated as 'other' and which, after checks for local variants in calling different roles, did not belong to any one of the four categories of interest (administrative, healthcare, technical, professional);
— surveys indicating that the particular nursing/care home is not based in the North of Italy.

The rest of the data was retained for the estimate of prevalence. For the prevalence estimate calculation itself, we considered the effects of clustering and weighting in our (single-stage) survey design. As mentioned in §2.1.1, these elements do not influence the prevalence estimate strongly but can affect its standard error. Clusters were based on individual NCH.

All collected data were weighted by adjusting our sample to match population proportions from the cross-classification of available key socio-demographic variables [80,84]. In the absence of any known

**Table 1.** Summary of our research question and hypotheses, sampling, analysis and interpretation plan.

| | |
|---|---|
| question | What is the current prevalence of moderate-to-severe symptoms of PTSD symptomatology and/or anxiety among workers in Northern Italian NCH? |
| hypothesis | The prevalence of workers with psychological comorbidities is similar to that reported by recent studies on healthcare workers [40], i.e. 35%. |
| sampling | The target sample size is estimated using $$n \geq D^2 \frac{z_{\alpha/2}^2 p(1-p)}{\delta^2},$$ where $D^2$ is the design effect, $\delta$ is the margin of error, $p$ is the prevalence and $z_{\alpha/2} = 1.96$. With an estimated design effect (accounting for clustering and weighting) of $D^2 = 2.42$, an expected proportion $p = 0.35$ of workers reporting moderate-to-severe symptoms of distress and/or anxiety and a margin of error $\delta = 0.05$, the target sample size is $n = 847$. We will aim to collect at least 900 completed questionnaires from NCH (which are sampled from across the North of Italy). |
| analysis | Participants will be deemed to indicate moderate-to-severe symptoms if their scores surpass the threshold in at least one of the two scales (i.e. obtaining a score $\geq 10$ for the GAD-7 and/or $\geq 26$ for the IES-R) [40]. In the prevalence calculation, only complete surveys will be considered (no items missing from demographic questionnaire, GAD-7 and IES-R scales). In our single-stage cluster design, NCH will represent individual clusters. To reach a more representative prevalence estimate, data will be weighted by adjusting our sample to match population proportions on the cross-classification of the available key socio-demographic and geographic variables [80,81]. Our population of reference will be extracted from the available database of workers' demographics (gender and role) and location in the Italian health system, including workers in NCH, personnel from local health departments and from hospitals that are managed by local health departments [55]. The proportion of respondents reporting moderate-to-severe symptoms, together with 95% confidence intervals (estimated via the Taylor series linearization method for the variance), will be calculated in Stata® which can take account of both clustering and weighting. |
| interpretation | Prevalence portrays the potential burden that the proportion of individuals showing moderate-to-severe symptoms places on the population of reference. Information on prevalence should then be complemented with information on the individual severity of symptoms, functional impairment and socio-demographic variables. The following guide will be used for interpretation of the prevalence datum, as a starting point, and will be integrated with considerations on social determinants of mental health, feasibility factors, individual-level variables and preferences. **Prevalence ≥50%**: Probable need to implement a stepped-care approach, providing simple low-resource psychosocial interventions to all the symptomatic participants as a first step, and more structured psychological interventions for those staff members not responding to the first-line intervention. The stepped care model has been extensively studied in the scientific literature and in policy documents, and is feasible across different clinical settings [82,83]. **30% ≤ prevalence < 50%**: Probable need to explore in depth the psychological status of a sizeable proportion of participants and to provide low-resource interventions of different complexity and focus (in accordance with the identified needs and resource availability) at local level or to liaise with local public services providing psychological interventions. **10% < prevalence < 30%**: Probable need to explore in depth the psychological status of a small proportion of participants, to deliver tailored psychological interventions of different complexity and focus (in accordance with the identified needs and resource availability) at local level or to liaise with local public services providing psychological interventions. **Prevalence ≤ 10%**: Unlikely to be an emergency-related burden due to the pandemic. Probable need to liaise with local public health services to explore psychological state and provide appropriate intervention. |

**Table 2.** Weighting plan. Individual weights were assigned to match population proportions on the cross-classification of the available key socio-demographic variables (role and gender) for Northern Italy. Numbers in each cell indicate the (known) proportion of workers from the population of reference that belong to the corresponding cell. Letters a, b, c, d, e, f, g and h indicate the proportion of responses in the corresponding cell within our data. Weights were identified by dividing the population proportion by the proportion in our data. Individual scores from respondents in each cell were multiplied by the corresponding weight.

| HEALTHCARE | | PROFESSIONAL | | TECHNICAL | | ADMINISTRATIVE | |
|---|---|---|---|---|---|---|---|
| M | F | M | F | M | F | M | F |
| 0.1846/a | 0.5012/b | 0.0018/c | 0.0004/d | 0.0603/e | 0.1346/f | 0.0220/g | 0.0951/h |

comprehensive or representative database for socio-demographic characteristics of the selected population of workers from Northern Italian NCH,[5] the population of reference was that of Northern Italian workers from the Italian national healthcare system as described in a relevant subsection of the latest database published by the Ministry of Health [55] (namely, the subsection referring to personnel employed by local health departments, or Aziende Sanitarie Locali, A.S.L.; see p. 16 of the following document: http://www.salute.gov.it/imgs/C_17_pubblicazioni_2870_allegato.pdf). After the exclusion of hospitals, university hospitals and research centres connected with the national healthcare system from the database, the available subsection data does include, but is not limited to, NCH. Indeed, it still includes personnel working in local health departments and personnel from hospitals that are directly managed by local health departments (which, we estimate, may represent altogether about half of the personnel included in this subsection). This database provides exhaustive information on the proportion of workers in administrative, healthcare, technical or professional roles by region and by gender. Timestamped raw data have been made available via a dedicated link on the Open Science Framework (OSF) for future re-analysis with adjusted weightings, were an exhaustive or representative database to become available for the selected population of workers in NCH. The values of the weightings used in this study can be found in table 2.

The prevalence calculation was performed using Stata® 16, a commercial statistical software package that is used for survey analysis [46]. This software can account for all the aspects of our complex survey design that we outlined above. The variance was estimated via the Taylor series linearization method. Stata® 16 was also used to estimate our design effect.

The variables from our data collection that were used for the prevalence calculation had the following format:

| ID | RESULT | JOB | GENDER | CLUSTER | WEIGHT |
|---|---|---|---|---|---|

ID is a random identification number assigned to a response (not determined by the individual who submitted the response); RESULT is a binary variable (Y or N) specifying whether or not the threshold for moderate-to-severe self-reported symptoms is reached; JOB can be H, P, T or A corresponding to the occupation roles in table 2; GENDER is a binary variable (M or F); WEIGHT is the weighting applied to each response based on the algorithm identified in table 2. For the prevalence calculation itself, only RESULT, CLUSTER and WEIGHT are necessary (although WEIGHT is constructed making use of JOB, GENDER and the values in table 2).

# 3. Results

## 3.1. Pre-registered analysis

### 3.1.1. Responding rate and exclusions

Out of a total of 188 NCH contacted by email and phone, 33 (i.e. about 18%) agreed to participate. A total of 1140 responses were collected. After screening for incomplete surveys ($N = 61$; of which 36 interrupted

---

[5]Both the Italian Institute of Statistics (ISTAT) and the Italian Ministry of Health advise that key demographics (such as age, gender and education level; information about ethnicity, which is often requested in surveys in the UK, is usually replaced by information about citizenship in Italy) are only available for residents, and not for workers, in NCH at the national level.

soon after having provided consent, 8 after having completed the demographic questionnaire, 17 after having completed the GAD-7 questionnaire), surveys in which 2 or more of the inattentive/careless response check items were not responded correctly ($N = 0$), surveys in which the indicated job did not belong to any of the categories of interest ($N = 8$; 4 descriptions were too generic and 4 concerned the external delivery of meals), surveys from NCH outside of Northern Italy ($N = 0$), a total of 1071 complete surveys were retained. Within individual NCH, the average responding rate was 53% of the contacted workers.

### 3.1.2. Demographic and occupational characteristics of the sample

Of the 1071 included participants, 810 (75.6%) were healthcare staff, 146 (13.6%) were technical staff and 115 (10.8%) were administrative staff. Most participants were women (916 [85.4%]), were aged between 41 and 60 years (671 [62.7%]), and had an educational level up to high school (752 [70.1%]). Our participants worked for a median of 5 days (last week: median $M = 5$, inter-quartile range IQR = 1; prior to last week: $M = 5$, IQR = 1) in each of the last two weeks. The majority of the participants (946 [88.3%]) reported having had continuous access to PPE and a total of 343 (32%) participants reported having had contact with COVID-19-positive colleagues or having assisted COVID-19-positive patients for at least 1 day in the last two weeks. The data from the demographic questionnaire are summarized in table 3.

### 3.1.3. Prevalence of moderate-to-severe symptoms of anxiety and/or PTSD

Four-hundred and seventy-four participants (44%; s.e. = 2.95; 95% CI [38–50]; population size = 1071; design d.f. = 32) passed the set threshold for moderate-to-severe symptoms for the GAD-7 and/or the IES-R. After weighting, a prevalence of 43% (s.e. = 3.09; 95% CI [37–49]; population size = 1069; design d.f. = 32)[6] was estimated. The design effect was found to be $D^2 = 4.17$, the relvariance $cv^2 = 0.16$ and the rate of homogeneity (intra-cluster correlation) $\rho = 0.08$ (using $n_c = 33$, which is the ceil of the mean cluster size).

## 3.2. Additional exploratory analyses

### 3.2.1. Multivariate logistic regression analysis

In order to perform a multivariate logistic regression analysis for the dependent variable of the threshold for moderate-to-severe symptoms (including weights), we first performed bivariate analyses of this variable with all the potential independent (predictor) variables. We determined the Pearson $\chi^2$ F-statistic and $p$-value for each bivariate association. All variables with $p > 0.1$ were excluded from consideration for the multivariate logistic analysis. The variables with $p < 0.1$ are 'gender', 'number of days worked last week' and 'contact with COVID-positive patients or staff'. Before proceeding to the logistic analysis, we tested for collinearity among any two predictor variables using Pearson's $R$ correlation coefficient. None of the variables was found to be collinear. Therefore, the above three variables that we have listed were included in the regression analysis, the results of which are displayed in table 4.

It can be seen that the only statistically significant factors in the logistic regression, at $p < 0.05$, are 'gender' and 'contact with COVID-positive patients or staff' (for brevity we do not show the breakdown of 'days worked last week' since it is not statistically significant). The design-adjusted F-statistic for the multivariate regression model is equal to 6.88, with $p = 0.0012$.

The estimated odds ratios indicate that females are about twice as likely to report moderate-to-severe symptoms as males and those with recent contact with COVID-positive colleagues/patients are about 1.7 times as likely to report symptoms compared with those without recent contact. The confidence intervals for the odds ratios of both of these variables do not cross to below 1, which indicates that these results are reliable. Details on how these results have been obtained using Stata can be found in the OSF repository at the following link: https://osf.io/gmbsa/. Instructions on how to explore the data more generally are also given.

---

[6]All statistical analyses were also performed using a three-category weighting scheme, whereby the three categories accounted for 100% of the population of reference (healthcare, administrative, technical staff). Since no substantial differences were found in terms of outcomes, we only report the four-category weighting, as specified in the pre-registered protocol.

**Table 3.** Demographic and occupational characteristics of our sample of participants. Cells show raw numbers (percentages) or median and inter-quartile range (M;IQR), when explicitly specified. Totals are shown in bold.

| characteristic | role | | | |
| --- | --- | --- | --- | --- |
| | admin | health | tech | all |
| **N** | | | | |
| our sample | **115 (10.8)** | **810 (75.6)** | **146 (13.6)** | **1071 (100)** |
| *population used for weighting* | ***21 411 (11.7)*** | ***125 326 (68.6)*** | ***35 627 (19.5)*** | ***182 364 (99.8)[a]*** |
| **gender** | | | | |
| female—our sample | 95 (82.6) | 702 (86.7) | 119 (81.5) | **916 (85.5)** |
| *female—population used for weighting* | *17 387 (81.2)* | *91 587 (73.1)* | *24 596 (69)* | ***133 570 (73.2)*** |
| male—our sample | 20 (17.4) | 108 (13.3) | 27 (18.5) | **155 (14.5)** |
| *male—population used for weighting* | *4024 (18.8)* | *33 739 (26.9)* | *11 031 (31)* | ***48 794 (26.8)*** |
| **age** | | | | |
| 18–25 | 4 (3.5) | 45 (5.6) | 3 (2.1) | **52 (4.9)** |
| 26–30 | 4 (3.5) | 79 (9.7) | 10 (6.9) | **93 (8.7)** |
| 31–40 | 17 (14.8) | 154 (19.0) | 26 (17.8) | **197 (18.4)** |
| 41–50 | 50 (43.5) | 247 (30.5) | 57 (39.0) | **354 (33.0)** |
| 51–60 | 38 (33.0) | 235 (29.0) | 44 (30.1) | **317 (29.6)** |
| >60 | 2 (1.7) | 50 (6.2) | 6 (4.1) | **58 (5.4)** |
| **education level** | | | | |
| primary school | 0 (0) | 3 (0.3) | 1 (0.7) | **4 (0.4)** |
| middle school | 4 (3.5) | 225 (27.8) | 37 (25.3) | **266 (24.8)** |
| high school diploma | 58 (50.4) | 370 (45.7) | 54 (37.0) | **482 (45.0)** |
| university degree | 53 (46.1) | 212 (26.2) | 54 (37.0) | **319 (29.8)** |
| **days worked in the past week (M;IQR)** | 5;1 | 5;1 | 5;1 | **5;1** |
| **days worked in the prior week (M;IQR)** | 5;1 | 5;1 | 5;2 | **5;2** |
| **contact with COVID+** | | | | |
| no | 97 (84.3) | 519 (64.1) | 112 (76.7) | **728 (68.0)** |
| yes[b] | 18 (15.7) | 291 (35.9) | 34 (23.3) | **343 (32.0)** |
| **access to PPE** | | | | |
| no[c] | 19 (16.5) | 87 (10.7) | 19 (13.0) | **125 (11.7)** |
| yes | 96 (83.5) | 723 (89.3) | 127 (87.0) | **946 (88.3)** |
| **family visits suspended** | | | | |
| no | 43 (37.4) | 282 (35.8) | 56 (38.4) | **381 (35.6)** |
| yes | 72 (62.6) | 528 (65.2) | 90 (61.6) | **690 (64.4)** |

[a]The population of reference (total $N = 182\,786$) includes also a minority (0.2%) of professionals (not present in our sample).
[b]The yes group includes both respondents who had continuous contact ($N = 153$) and respondents who had intermittent contact ($N = 190$) with COVID-positive colleagues/patients in the past two weeks.
[c]The no group includes both respondents who had no access ($N = 65$) and respondents who had only intermittent access ($N = 60$) to PPE at work.

### 3.2.2. Anxiety and/or PTSD?

Two-hundred and thirty-five participants (22%; s.e. = 1.88; 95% CI [18–26]; population size = 1071; design d.f. = 32) passed the set threshold for moderate-to-severe symptoms at the GAD-7, whereas 433 participants (40%; s.e. = 3; 95% CI [35–47]; population size = 1071; design d.f. = 32) passed the set threshold for moderate-to-severe symptoms at the IES-R. Of these, 194 participants (18% of the total

**Table 4.** Regression analysis for the moderate-to-severe symptoms threshold.

| | odds ratio | s.e. | $t$ | $p > |t|$ | 95% CI |
|---|---|---|---|---|---|
| gender (female) | 2.182 | 0.4745 | 3.59 | 0.001[a] | [1.401 3.398] |
| days worked last week | 1.0197 | 0.0477 | 0.42 | 0.678 | [0.927 1.122] |
| contact with COVID+ (y) | 1.6657 | 0.3889 | 2.19 | 0.036[a] | [1.035 2.6799] |
| y-intercept | 0.326 | 0.09 | −4.04 | <0.0001 | [0.185 0.574] |

[a]Significant at $p < 0.05$.

**Table 5.** Regression analyses for anxiety and PTSD symptomatology.

| anxiety ($F = 2.84$, $p = 0.042$) | odds ratio | s.e. | $t$ | $p > |t|$ | 95% CI |
|---|---|---|---|---|---|
| gender (female) | 1.599 | 0.367 | 2.04 | 0.049[a] | [1.002 2.552] |
| days worked last week | 1.064 | 0.084 | 0.79 | 0.437 | [0.906 1.25] |
| contact with COVID+ | 1.453 | 0.308 | 1.76 | 0.087 | [0.981 1.665] |
| education | 1.278 | 0.166 | 1.89 | 0.068 | [0.981 1.665] |
| y-intercept | 0.06 | 0.032 | −5.29 | <0.0001 | [0.02 0.1767] |
| PTSD ($F = 5.34$, $p = 0.0046$) | odds ratio | s.e. | $t$ | $p > |t|$ | 95% CI |
| gender (female) | 2.186 | 0.523 | 3.27 | 0.003[a] | [1.342 3.558] |
| contact with COVID+ | 1.714 | 0.369 | 2.5 | 0.018[a] | [1.106 2.656] |
| occupation | 1.176 | 0.141 | 1.35 | 0.188 | [0.92 1.503] |
| y-intercept | 0.214 | 0.082 | −4.05 | <0.0001 | [0.099 0.465] |

[a]Significant at $p < 0.05$.

sample; s.e. = 1.87; 95% CI [15–22]; population size = 1071; design d.f. = 32) passed the threshold for both anxiety and PTSD at the same time.

After weighting, a prevalence of 22% (s.e. = 1.87; 95% CI [18–26]; population size = 1069; design d.f. = 32) was estimated for anxiety and a prevalence of 39% (s.e. = 3.12; 95% CI [33–46]; population size = 1069; design d.f. = 32) was estimated for PTSD, with design effects of 2.21 and 4.36 and intra-cluster correlations of 0.03 and 0.09, respectively. The prevalence of comorbidity was estimated to be 18% (s.e. = 1.83; 95% CI [14–22]; population size = 1069; design d.f. = 32). In this case, the design effect is equal to 2.46 and the intra-cluster correlation to 0.04.

For the weighted results, we performed logistic regression analyses for anxiety and PTSD. We applied the same methodology as in the previous section to select variables for the analyses and, for brevity, we do not display all the steps here. The results of the logistic regression analyses are displayed in table 5.

For anxiety, only 'gender' is statistically significant, whereas for PTSD, both 'gender' and 'contact with COVID-positive patients or staff' are significant. In the logistic regression analysis for anxiety, the odds ratio for gender suggests that females are about 1.6 times more likely to report moderate-to-severe symptoms than males. For PTSD, females are about 2.2 times more likely to report symptoms than males and those with recent contact with COVID-positive colleagues/patients are about 1.7 times as likely to report symptoms compared with those without recent contact.

### 3.2.3. Severity of symptoms and comparisons between groups

A considerable proportion of participants (87%; s.e. = 0.13; 95% CI [85–90]; population size = 1071; design d.f. = 32; weighted: 87%; s.e. = 0.14; 95% CI [84–89]; population size = 1069; design d.f. = 32) reported at least mild symptoms of anxiety and/or PTSD. Table 6 shows the proportion of responses across symptom severity categories for the entire sample and for selected subgroups of interest. Responses to the GAD-7 and to the IES-R were ranked (0–3, according to their severity; see §2.2.2. for cut-offs) and entered in Mann–Whitney or Kruskal–Wallis tests, depending on whether the test was performed on

**Table 6.** Severity categories of psychological symptoms. Cells show raw numbers (percentages). *Remains significant after Bonferroni correction ($\alpha_{FW} = 0.05$). Totals are shown in bold.

| N (%) | overall (GAD-7 AND/OR IES-R) | | | | | GAD-7 | | | | | IES-R | | | | |
|---|---|---|---|---|---|---|---|---|---|---|---|---|---|---|---|
| | normal | mild | moderate | severe | p-value | normal | mild | moderate | severe | p-value | normal | mild | moderate | severe | p-value |
| **whole sample** | **134 (12.5)** | **463 (43.2)** | **319 (29.8)** | **155 (14.5)** | | **389 (36.3)** | **447 (41.8)** | **160 (14.9)** | **75 (7.0)** | | **169 (15.8)** | **469 (43.8)** | **303 (28.3)** | **130 (12.1)** | |
| **occupation** | | | | | 0.0859 | | | | | 0.1054 | | | | | 0.1338 |
| administrative | 14 (12.2) | 47 (40.8) | 40 (34.8) | 14 (12.2) | | 37 (32.2) | 48 (41.7) | 20 (17.4) | 10 (8.7) | | 21 (18.3) | 43 (37.4) | 41 (35.6) | 10 (8.7) | |
| healthcare | 106 (13.1) | 361 (44.6) | 227 (28.0) | 116 (14.3) | | 307 (37.9) | 335 (41.3) | 114 (14.1) | 54 (6.7) | | 130 (16.0) | 369 (45.6) | 210 (25.9) | 101 (12.5) | |
| technical | 14 (9.6) | 55 (37.7) | 52 (35.6) | 25 (17.1) | | 45 (30.8) | 64 (43.9) | 26 (17.8) | 11 (7.5) | | 18 (12.3) | 57 (39.1) | 52 (35.6) | 19 (13.0) | |
| **gender** | | | | | <0.0001* | | | | | 0.0001* | | | | | <0.0001* |
| female | 103 (11.3) | 385 (42.0) | 286 (31.2) | 142 (15.5) | | 310 (33.9) | 396 (43.2) | 140 (15.3) | 70 (7.6) | | 131 (14.3) | 393 (42.9) | 272 (29.7) | 120 (13.1) | |
| male | 31 (20.0) | 78 (50.3) | 33 (21.3) | 13 (8.4) | | 79 (51.0) | 51 (32.9) | 20 (12.9) | 5 (3.2) | | 38 (24.5) | 76 (49.0) | 31 (20.0) | 10 (6.5) | |
| **contact with COVID+** | | | | | <0.0001* | | | | | 0.0052 | | | | | <0.0001* |
| no | 103 (14.1) | 331 (45.5) | 208 (28.6) | 86 (11.8) | | 279 (38.3) | 306 (42.1) | 102 (14.0) | 41 (5.6) | | 133 (18.3) | 331 (45.5) | 195 (26.8) | 69 (9.4) | |
| yes | 31 (9.0) | 132 (38.5) | 111 (32.4) | 69 (20.1) | | 110 (32.1) | 141 (41.1) | 58 (16.9) | 34 (9.9) | | 36 (10.5) | 138 (40.2) | 108 (31.5) | 61 (17.8) | |
| **access to PPE** | | | | | 0.7296 | | | | | 0.5503 | | | | | 0.2187 |
| no | 119 (12.6) | 407 (43.0) | 280 (29.6) | 140 (14.8) | | 345 (36.5) | 397 (41.9) | 139 (14.7) | 65 (6.9) | | 147 (15.5) | 411 (43.5) | 269 (28.4) | 119 (12.6) | |
| yes | 15 (12.0) | 56 (44.8) | 39 (31.2) | 15 (12) | | 44 (35.2) | 50 (40.0) | 21 (16.8) | 10 (8.0) | | 22 (17.6) | 58 (46.4) | 34 (27.2) | 11 (8.8) | |
| **family visits** | | | | | 0.8401 | | | | | 0.6066 | | | | | 0.9755 |
| no | 46 (12.1) | 164 (43.0) | 118 (31.0) | 53 (13.9) | | 131 (34.4) | 169 (44.3) | 54 (14.2) | 27 (7.1) | | 60 (15.7) | 166 (43.6) | 110 (28.9) | 45 (11.8) | |
| yes | 88 (12.8) | 299 (43.3) | 201 (29.1) | 102 (14.8) | | 258 (37.4) | 278 (40.3) | 106 (15.4) | 48 (6.9) | | 109 (15.8) | 303 (43.9) | 193 (28.0) | 85 (12.3) | |

**Table 7.** Median scores of anxiety and PTSD in the total sample and subgroups. Cells show median (IQR). *Remains significant after Bonferroni correction ($\alpha_{FW} = 0.05$). Totals are shown in bold.

| | median (IQR) score | |
|---|---|---|
| | **GAD-7** | **IES-R** |
| **all** | **6 (6)** | **22 (21)** |
| **role** | | |
| administrative | 6 (6) | 22 (21) |
| healthcare | 6 (6) | 24 (22) |
| technical | 7 (6) | 25 (21) |
| *p*-value | *0.0595* | *0.1428* |
| **gender** | | |
| female | 6 (5) | 23 (21.5) |
| male | 4 (6) | 15 (18) |
| *p*-value | *<0.0001** | *<0.0001** |
| **contact with COVID+** | | |
| no | 6 (5) | 20 (18.5) |
| yes | 6 (6) | 25 (25) |
| *p*-value | *0.0189* | *<0.0001** |
| **access to PPE** | | |
| no | 6 (6) | 22 (21) |
| yes | 6 (5) | 21 (19) |
| *p*-value | *0.5564* | *0.4439* |
| **family visits** | | |
| no | 6 (6) | 22 (22) |
| yes | 6 (4) | 23 (20) |
| *p*-value | *0.4854* | *0.7746* |

two or more independent groups. The analysis revealed that females reported more severe symptom levels compared with males, and workers who had contact with COVID-positive colleagues/patients reported more severe symptom levels than workers who had not had contact with COVID-positive colleagues/patients in the past two weeks. A significant difference between males and females was present both across scales and for each individual scale. A significant difference between workers who had contact with COVID-positive colleagues/patients and workers who had not had contact with COVID-positive colleagues/patients was found across scales and for the IES-R but not for the GAD-7 after a Bonferroni correction was applied ($\alpha_{FW} = 0.05$). No difference in the severity of the reported symptoms was found between groups according to role, access to PPE or whether family visits were allowed (all $p$s > 0.08; the tests were run also for age and education, although these are not shown in table 6, and the resulting $p$s are equal to 0.11 and 0.37, respectively).

### 3.2.4. Measurement scores

The median (IQR) scores on the GAD-7 and IES-R for all respondents are 6 (6) and 22 (21) respectively. Cronbach's alpha was 0.89 for GAD-7 and 0.93 for IES-R (Intrusion: 0.90, Avoidance: 0.81, Hyperarousal: 0.84). Similar to the findings about the severity of symptoms, participants who were female had higher scores for both the GAD-7 and the IES-R, whereas those who had contact with COVID-positive colleagues/patients at work had higher scores for the IES-R only (table 7). Both females and workers who had contact with COVID-positive colleagues or patients had significantly higher scores for all the three subscales of the IES-R (Intrusion, Avoidance, Hyperarousal; table 8).

**Table 8.** Median scores of PTSD symptoms in the total sample and subgroups. Cells show median (IQR). *Remains significant after Bonferroni correction ($\alpha_{FW} = 0.05$). Totals are shown in bold.

| | median (IQR) score | | |
| --- | --- | --- | --- |
| | intrusion | avoidance | hyperarousal |
| **all** | **8 (10)** | **7 (8)** | **6 (7)** |
| **sex** | | | |
| female | 8 (10) | 8 (8) | 6 (7) |
| male | 5 (8) | 5 (8) | 5 (5) |
| *p*-value | *<0.0001** | *0.0003** | *<0.0001** |
| **contact with COVID+** | | | |
| no | 7 (8) | 7 (8) | 6 (6) |
| yes | 10 (11) | 9 (9) | 7 (8) |
| *p*-value | *0.0001** | *<0.0001** | *<0.0001** |

# 4. Discussion

In this study, we set out to reach a largely ignored population, i.e. nursing and care home staff, that plays a very sensitive and crucial role in our society and on whom the outbreak of the COVID-19 pandemic has imposed a particularly sudden and heavy burden. We collected evidence on the mental health state of a large sample of workers from NCH in Northern Italy, the most widely affected region by the spread of the pandemic, across six weeks during the subacute phase (Phase 2). Firstly, we aimed to estimate the prevalence of moderate-to-severe PTSD symptomatology and/or anxiety. This was the focus of our pre-registered analysis, which also returns valuable information about the rate of homogeneity for mental health symptomatology and demographic characteristics of workers in NCH. In the exploratory analyses, we calculated separate prevalence estimates for PTSD symptomatology and anxiety. We also probed the relation between prevalence estimates and additional survey data, to explore possible associations between demographic/work characteristics and the reported symptoms.

In this section, we discuss first the identified parameters of the survey design and how they affect our prevalence estimates. We will then move on to discuss the outcome of the pre-registered and the exploratory analyses, with reference to recent studies focused on healthcare workers. We will finish by providing suggestions for possible interventions.

## 4.1. Survey design parameters

Our complex survey design is based on one stratum (the North of Italy) divided into clusters (individual NCH). As explained in §2.1.1, the SRS estimate for the number of participants needs to be multiplied by the design effect $D^2$ in order to account for the clustering and weighting of participants. We modelled the design effect as being based on two main factors—one to take into account the population weighting of results, whose main parameter is the relvariance $cv^2$; the other to model the intra-cluster correlation, whose main parameter is the rate of homogeneity $\rho$. Without *a priori* information about these parameters from other surveys related to NCH (studies sufficiently close to our own, such as [40], do not report such quantities) we chose values in line with typical values that we could find elsewhere in the literature related to human studies. In the following, we focus on parameters related to the main prevalence estimate.

Our original assumptions for the survey parameters underestimated, but only slightly, the values calculated after the study: $cv^2_{before} = 0.1$, $cv^2_{after} = 0.16$; $\rho_{before} = 0.05$, $\rho_{after} = 0.08$. Thus these parameters will be useful for the design of future surveys related to NCH, since no similar results are currently available in the literature. For the other prevalence estimates (GAD-7 and IES-R), the values of $\rho$ are of the same order of magnitude as that of the main prevalence result, suggesting that the cluster response to all these factors is similar.

The values of the above parameters for the main prevalence result, being slightly larger than their initial assumed values, have an effect on the confidence intervals of our prevalence estimate. The

upper and lower bounds of the CIs are a distance of 0.06, with and without the inclusion of weighting in the calculation. The upper (lower) bound can be reduced (increased) by 0.01 by including, at least, another 389 participants—due to the enhanced design effect of $D^2 = 4.17$. As mentioned before, however, the estimated prevalence itself would remain very close to the value we have found.

Another factor that has inflated the design effect compared with our initial estimate is the cluster size $n_c$. We initially expected about 25 respondents from individual NCH; however, the actual (mean) value in our study is 33. Despite this enhanced participation, the effect of the complex survey design means that, for a fixed value of $\rho$, larger clusters require an overall larger number of participants in order to achieve the same statistical goals.

The above points highlight the importance of taking survey design (e.g. clustering and weighting) into account in studies of this kind. This is important in order to convey valid statistical results. It is particularly important in rapidly changing situations such as the COVID-19 pandemic, for which survey studies may act as prompts for swift interventions.

## 4.2. Outcome of the pre-registered analysis

To our knowledge, this study represents the first survey focused on the mental health of all types of workers in NCH in Italy.

In the first instance, we investigated a combined index that subsumes both anxiety and PTSD symptomatology to provide a more comprehensive estimate of the psychological state of our participants (e.g. a similar approach in [49]). Indeed, several weeks after the emergency peak, while PTSD symptomatology may still be present, it is also possible that participants have developed symptoms of anxiety. In previous studies, high levels of PTSD symptoms and lower levels of anxiety were found around the peak of contagions [40], whereas higher levels of anxiety were found several months after the emergency [22].

We report that a significant proportion of participants experienced psychiatric symptoms at different levels of intensity soon after the peak of the first COVID-19 outbreak (i.e. in Phase 2; figure 2). In particular, we find a prevalence of 43% for anxiety and/or PTSD symptoms of moderate-to-severe intensity, which represents not only a significant emotional burden, but also a risk factor for the development of a psychiatric diagnosis.

The prevalence we detect is not too far off the combined prevalence of anxiety and PTSD symptomatology reported among Chinese healthcare workers during the peak of contagions. In the most comparable study (using our same measures and a partly similar sampling method), the combined prevalence of moderate-to-severe symptoms of anxiety and post-traumatic symptoms among first- and second-line workers reaches 47.25% [40]. In this case, we do not know what proportion of respondents endorsed both anxiety and PTSD symptomatology. However, assuming a prevalence of comorbidity similar to that in our sample, the prevalence of anxiety and/or post-traumatic symptomatology should approach 38–39% in first- and second-line healthcare workers in Chinese hospitals during their Phase 1.

The high levels of anxiety and/or PTSD symptoms found in the present survey might be influenced by the interplay of several factors, in addition to the higher likelihood of being exposed to the virus as healthcare professionals. First, the type of working environment in NCH facilitates the emotional involvement of staff. Whereas the hospital setting is mainly characterized by short-term stays, with patients transferring to specialized wards/departments according to their physical conditions, NCH host individuals for much longer-term periods. This protracted contact can help build closer relationships between nursing home residents and staff, who may feel engaged and more emotionally involved in the individual care and management of their guests. Second, the elderly population in NCH is at higher risk of contracting and suffering the most severe consequences of COVID-19 compared with the more heterogeneous hospital population. This issue, together with the prolonged experience of assisting seriously ill people, may have intensified the emotional response to the pandemic of all workers in NCH. Third, the geographical location where the study was conducted might have played a role in influencing the level of psychiatric symptoms in NCH.

Recent online surveys on the psychological state of Italian healthcare workers conducted in the midst of Phase 1, outline a dramatic picture in general. De Sio *et al.* [85], for example, report a prevalence of individuals with any level of severity of psychological distress as high as 89% on a sample of 695 respondents, around the peak of contagions in Italy. The survey was administered via public website and physicians were recruited via mailing lists of healthcare research departments from Central Italy. Psychological distress was measured with the General Health Questionnaire (GHQ-12 [86]), which has a more general focus than our measures. Notably, the proportion of participants reporting at least

mild symptoms of anxiety and/or PTSD in our study is also considerable, taking into account that it was conducted during Phase 2, as it reaches 87%. Di Tella et al. [87] recruited 72 doctors and 73 nurses in Northern Italy via convenience and snowball sampling, with code-restricted access to the survey. They find a total prevalence of 97% for respondents with any level of anxiety (as measured with the state anxiety scale STAI-Y [88]) plus respondents meeting the criterion for a provisional diagnosis of PTSD (based on PCL-5 [89]). The comorbidity proportion is unknown. Finally, Rossi et al. [90] find a 69% total prevalence of respondents endorsing severe anxiety symptoms, at the STAI-Y, and respondents reporting significant post-traumatic stress symptoms, at the GPS-PTSD subscale, across 1379 healthcare workers (comorbidity unknown). In this case, respondents were from across the whole of Italy but mostly from Northern and Central Italy. Recruitment was performed via social networks using snowballing and sponsored adverts near the peak of the COVID-19 contagions. In our own study (Phase 2, NCH) the same calculation would return a total prevalence of 19% for respondents endorsing severe anxiety and respondents reporting severe PTSD symptomatology (uncorrected for comorbidity); in Lai et al.'s, (Phase 1, hospital workers) it would return 16% [40].

The different times at which data were collected (Phase 2 versus Phase 1), the different sampling approaches (exhaustive sampling via employer versus snowball, websites or social media), the different tests employed and the peculiarity of our target population (workers in NCH versus hospital healthcare workers) make it difficult to draw common clinical conclusions. Based on the above comparisons with the extant literature on Italian healthcare workers, we speculate that our prevalence datum may represent a conservative estimate of self-reported psychological symptoms in Northern Italian NCH. Indeed, it is not unlikely that recruitment via social media would have returned a more dramatic picture than recruitment via organizations, as organizations that are less worker-oriented would be unlikely to endorse participation in a study such as this one. This, however, would not enable control on sample representativeness or return any useful information on design parameters— valuable features for future studies.

## 4.3. Outcome of the exploratory analysis

The exploratory analyses reveal that 22% and 40% of our participants reported moderate-to-severe symptoms of anxiety and PTSD, respectively. Among them, 18% reported both anxiety and PTSD symptoms at the same time.

The anxiety figure appears consistent with a recent systematic review and meta-analysis evaluating the prevalence of depression, anxiety and insomnia among healthcare workers during the COVID-19 pandemic [91]. The authors identified 12 studies assessing anxiety symptoms and reported a prevalence rate of 23%. However, heterogeneity measured with the $I^2$ statistic was 99%, suggesting that the collected studies were extremely different in terms of methodological and clinical characteristics: some calculated prevalence based on different levels of severity, some included mild disorders, others considered just moderate-to-severe symptoms [91].

The higher prevalence of PTSD symptomatology is also consistent with survey-based studies on the effects of the COVID-19 pandemic on healthcare workers (e.g. [40,90] but see [87]) and positions itself at the upper end of the range reported for other well-known emergency situations. In our exploratory analyses, we also find that for moderate-to-severe symptoms of anxiety and/or PTSD, females are approximately twice as likely to report these symptoms as males (OR = 2.2, 95% CI [1.4 3.4], $p$ = 0.001). A similar pattern emerges from the analyses of anxiety alone (OR = 1.6, 95% CI [1.002 2.6], $p$ = 0.049) and PTSD symptomatology alone (OR = 2.2, 95% CI [1.3 3.6], $p$ = 0.003).

These findings are in line with epidemiological research on gender and PTSD, which identifies women being diagnosed with PTSD approximately twice as often as men, independently of the number of traumatic experiences [92]. Further, a recent systematic review and meta-analysis of Kisely et al., who collected 59 studies assessing the psychological reactions of healthcare staff in an outbreak of any emerging virus, identified gender (being woman) among the predisposing factors for PTSD and other psychological conditions [30]. Finally, women develop stronger and longer-term PTSD symptoms, and this should be carefully considered in planning psychological intervention contents and delivery [30].

Gender differences have been identified also for anxiety. A large survey of more than 20 000 participants (11 463 women and 8550 men) examined lifetime and past-year occurrences of anxiety disorders, and found that women were significantly more likely to meet diagnostic criteria for anxiety disorders [93]. The study of Lai et al. confirms significantly increased levels of anxiety (severe symptoms) in women compared with men, with a prevalence of 5.8% versus 3.4% ($p$ = 0.001), respectively [40]. A similar pattern was identified by Du et al. in a regression analysis accounting for

gender in a sample of 134 healthcare workers in Wuhan (53 males, 81 females, with a prevalence of 11% versus 27%) [94].

In our study, workers who had contact with COVID-positive colleagues/patients report more severe symptom levels compared with workers who had not had such contact (OR = 1.7, 95% CI [1.04 2.7], $p$ = 0.036). This applies also when considering moderate-to-severe symptoms of PTSD alone (OR = 1.7, 95% CI [1.1 2.7], $p$ = 0.018). These findings are consistent with results from the literature on healthcare workers, in which exposure to COVID-19-positive patients emerges as a potential risk factor for psychiatric symptoms during and immediately after the first COVID-19 outbreak [24,40,87,90,95–97].

Interestingly, recent contact with COVID-positive persons is not associated with reporting moderate-to-severe symptoms of anxiety alone. Lai *et al.* [40] found that treating COVID-19-positive patients appeared to be an independent factor associated with all psychological symptoms (depression, anxiety, insomnia and distress), and discussed this finding in light of the increased and direct engagement of frontline workers. Our finding can be interpreted by considering the specific nature of the relationship between staff in NCH and COVID-19-positive persons, as described above (see Introduction). The majority of staff reporting symptoms in our study may have experienced the COVID-19 pandemic as particularly traumatic, thus developing serious post-traumatic symptoms as a reaction [98]. Only in some of them, acute traumatic reactions may have also generated anxiety as measured here, which may be a more generalized reaction connected to chronic stressors rather than only to specific traumatic events [99]. Hence the higher prevalence of PTSD associated with exposure to COVID-19.

As expected, no difference in the severity of psychological symptoms is found in our study between administrative staff versus healthcare versus technical staff. This result is in contrast with findings from a study assessing the psychological status of the medical workforce in the South of China [96]. That study found that the medical staff experienced greater fear, anxiety and depression than the non-clinical administrative staff. Also, these apparently contradictory results might be explained by the different settings in which studies were conducted (hospitals versus NCH), as previously described. Moreover, the administrative staff in NCH often have an educational background in healthcare and, in the best-case scenarios, tend to display a preference towards person-centred care in their approach to management. This aspect significantly favours the adoption of a compassionate attitude in long-term care settings.

Unexpectedly, we do not identify significant differences between groups according to access to PPE, to whether family visits were allowed or the number of days worked in the last two weeks. Lack of access to PPE was found to be associated with depression, anxiety and PTSD symptoms in a sample of US nurses during the lockdown period [100]. However, an important detail to be taken into consideration for our study is that access to PPE in Phase 2 was already mostly guaranteed in Northern Italy. In fact, just a low percentage (about 12%) of participants in our study, compared with the US sample (about 25% [100]), reports not having access to PPE during the two weeks investigated. We suspect that the full emergency period in which the PPE was insufficient had already passed by the time of our data collection and the PPE factor was, therefore, no longer a risk element for the development of psychiatric symptoms.

Regarding access of family members into the structures, NCH faced the dilemma of infection prevention versus allowing personal contact for residents. In a Dutch study, part of the staff worried about their own health during family visits, but in general, these experiences were considered very positive by staff in NCH who recognized the added value of real and personal contact between residents and their loved ones [101]. In our study, 36% of the participants worked in NCH in which family visits were not suspended. We can thus expect that, in addition to representing a vehicle for possible infections, family members might have been seen as a positive factor and were welcome in the structure. From previous literature, we know that PTSD is linked closely to a sense of responsibility of death prevention [102]. Therefore, sharing such desperate feelings with the family might have decreased this sense of guilt and the suffering of seeing guests dying isolated. This may have contributed to the ambiguity of this factor.

## 4.4. Possible next steps

One of the most interesting findings of our study is that the prevalence of moderate-to-severe symptoms of PTSD is almost double the prevalence of anxiety symptoms. As mentioned previously, this result is likely to be influenced by the multiple traumatic experiences of staff in NCH (in terms of chronic stressors but especially of acute events in the aftermath of the infections peak) and the time since trauma exposure (all very recent experiences), which play a crucial role in determining the type and intensity of psychological symptoms [103]. PTSD is characterized by a specific pattern of psychological symptoms that is different in its etiopathology and clinical course from other psychiatric conditions (The ICD-11 Working Group [104]), and this could also subtend the difference in the

prevalence rate of anxiety versus PTSD. Furthermore, the literature establishes that work-related PTSD is common in health and social services, and PTSD is linked to the exposure to unexpected extreme traumatic events, as it was for staff during the death of beloved residents in NCH [105]). Indeed, the exposure to some life-changing events, such the unexpected death of a loved one, is a highly traumatic event associated with higher prevalence of PTSD, especially when it is linked to the personal perception/sense of responsibility of death prevention [102].

From a clinical perspective, we can expect a substantial proportion of participants to recover naturally from PTSD symptoms, showing, over time, mild or subsyndromal symptoms and/or resilience [106]. Some longitudinal studies on the psychological consequences of exposure to multiple traumatic experiences in humanitarian settings, and studies focused on resilience, have shown spontaneous improvements of psychological symptoms in subgroups of individuals [107,108]. However, the psychological impact of trauma is not distributed equally across population groups. Other studies have highlighted the risk of a significant proportion of individuals developing a formal PTSD [109]. Regarding the specific factors contributing to different clinical trajectories, there is still uncertainty on the role of clinical and social determinants (both at proximal and distal levels) [110]. In general, contextual factors, psychosocial support and time since trauma exposure seem to be variables with a healing effect, while gender (being woman), low income, and educational level have been shown to be associated with increased vulnerability [111]. For these reasons, it is imperative to focus on evidence-based strategies that could prevent people exposed to potentially traumatic events and with early psychological symptoms from developing a formal psychiatric diagnosis.

Among the mental health and psychosocial interventions listed in the IASC guidelines, those interventions placed in the upper two layers of the IASC pyramid (figure 3) might be appropriate for population groups showing psychological symptoms of PTSD (and of anxiety). There is a solid base of evidence on the effectiveness of these interventions, confirmed by recent systematic reviews, meta-analyses and network meta-analyses [28,112–114]. For example, a systematic review and network analysis of Coventry et al. [114] identified significant benefits for individuals receiving psychological interventions versus control conditions for outcomes such as anxiety, PTSD, depression and sleep quality. Regarding the focus of psychological interventions, a recent Cochrane review collecting 44 randomized control trials (7974 participants) focused on fostering resilience in healthcare professionals and found that healthcare professionals receiving resilience training may report higher levels of resilience and lower levels of stress/ stress perception compared with controls [115].

In reference to the interpretation plan presented in table 1, we observe that a large proportion of staff in NCH is in need of receiving interventions either in the form of psychosocial support and/or as a more structured psychotherapeutic intervention. Our finding that 43% of workers reported moderate-to-severe symptoms of anxiety and/or PTSD calls for an urgent in-depth assessment of the psychological status of staff in NCH and their social environment (i.e. proximal level social determinants of mental health), and to implement large-scale intervention strategies [116]. In addition to the clinical attention, our results suggest that PTSD symptoms in particular are likely to represent a response to the COVID-19 emergency instead of a transient and generic reaction to stressors. Policy makers should consider the importance of providing preventative psychological intervention strategies to all the staff in NCH during the COVID-19 emergency (that is still ongoing in Italy, especially in NCH, e.g. [117]) and beyond. Trans-diagnostic psychological interventions represent a valuable option for staff with psychological symptoms of anxiety and/or PTSD. There is already promising evidence for the implementation of these approaches for individuals with PTSD, with particular advantages in terms of acceptability (lower treatment drop-out [118]).

Finally, and with regard to the format of psychological interventions, online strategies could be a first choice delivery option during periods of social distancing rules for limiting the risk of infection. Recently, there has been growing attention paid to low-intensity and cost-effective interventions delivered through online technology (i.e. smartphone app) [119–122] that could represent a valid resource for large-scale implementation during the COVID-19 emergency and beyond.

## 4.5. Limitations

Despite the many positive aspects of our survey, we can highlight some limitations. First, we did not explore the symptoms of conditions other than anxiety and PTSD, as the study prioritized the most frequent reactions during a state of emergency. It may be worth investigating: depression (e.g. PHQ-9 [123]), substance/alcohol use, indicators of general functioning and quality of life (e.g. SF-12/36 for physical and mental quality of life indices [124]). Indeed, high levels of depression compared with

anxiety and distress were found among the healthcare staff in China [95], and high levels of depressive symptoms were also reported in Italy [87].

Second, we did not use diagnostic tools (e.g. Mini International Neuropsychiatric Interview) or other clinician-administered tools but self-administered instruments which could be vulnerable to symptom over-reporting when compared with structured clinical assessment, especially in emergency settings [125]. We were also unable to distinguish pre-existing mental health disturbances from new symptoms caused essentially by the pandemic, as we did not have pre-pandemic data from the participants involved in this survey [126]. This reasoning applies particularly to anxiety, that may represent a chronic condition, while our assessment of PTSD symptomatology is anchored in work-related potentially traumatic events during the course of the COVID-19 pandemic. Moreover, potentially traumatic events are sudden and unexpected, by definition, with an impairment that is strong but temporary, and the psychological symptoms that we detected might represent normal reactions during a situation like the COVID-19 pandemic. In any case, even though our participants already suffered from pre-existing symptoms we might have been able to detect heightened stress responses in subgroups of individuals. It should, however, be highlighted that our study was not designed to (and does not) establish causal relations.

Third, the average response rate of this study was 53% per structure, thus we might have missed the responses of either those who were too stressed to respond or those who were not stressed at all and lacked interest in the study.

Fourth, the prevalence rates of the present survey were calculated on data from NCH that accepted to be involved in a project exploring the psychological status of their workers and that facilitated staff participation: they may thus represent a conservative estimate, being collected in collaborative organizations only.

Finally, our survey is cross-sectional. Longitudinal evaluations may help to understand the course of psychological symptoms and functioning over time, with important implications in terms of the types of psychological interventions to be delivered. Moreover, follow-up assessments would help responding to clinical questions related to the spontaneous improvement of psychological symptoms versus the development of a psychiatric diagnosis in a population group that may suffer serious psychological impairment but is still under-represented in clinical studies.

# 5. Conclusion

This work is the first detailed study on the prevalence of psychiatric symptoms among workers in NCH in Italy immediately after the first COVID-19 outbreak. We find high levels of PTSD symptomatology and/or anxiety among all workers in NCH, with a higher level of PTSD and anxiety symptoms in females and a higher level of PTSD symptoms in those who were in recent contact with COVID-19-positive patients. No neat difference among workers in different job roles was found—rather, it appears that healthcare staff is not the only category to have been affected from the COVID-19 emergency in this work environment. Since all staff are part of a close-knit community sharing high levels of distress, the entire staff in NCH should be considered as a group with a higher risk of psychiatric symptoms during the COVID-19 pandemic, and be offered suitable interventions such as online psychosocial and psychotherapeutic interventions, which are respectful of social distancing rules and should be tailored according to the specific needs of participants. Interventions for the mental well-being of NCH workers should be implemented swiftly, with a particular focus on women and those who took care of COVID-positive patients. Preventative interventions could also be considered by NCH to improve staff resilience in a developing situation.

Ethics. The clinical research ethics committee of SPES Group has approved the protocol, after consultations with trade union representatives. The research was conducted according to the principles expressed in the Declaration of Helsinki and informed consent was obtained from all subjects before access to the survey. All subjects were able to interrupt the survey at any time.
Data accessibility. We have used data from the following governmental document for weightings: http://www.salute.gov.it/imgs/C_17_pubblicazioni_2870_allegato.pdf (please see p. 16 of the above doc., p. 10 and table 2 in our manuscript). Following in-principle acceptance, the approved Stage 1 version of this manuscript was preregistered on the OSF at https://osf.io/4fpqa. This pre-registration was performed prior to data collection and analysis. Our article's supporting data and research materials are available through the following link: https://osf.io/gmbsa/.
Authors' contributions. M.R. conceptualization, data curation, investigation, methodology, project administration, supervision, writing; M.P. conceptualization, methodology, writing; C.B. investigation; D.M. data curation, formal analysis, methodology, writing; E.R. conceptualization, data curation, formal analysis, methodology, software, supervision, writing. All authors gave final approval for publication.

Competing interests. We have no competing interests.

Funding. We received no funding for this study.

Acknowledgements. The authors thank all the participating NCH, the individuals who facilitated respondent enrolment within each centre and all the workers who, at these challenging times, found the time to respond to our survey. We also wish to thank the editor and the five reviewers for their very timely, insightful and constructive comments on our Stage 1 manuscript.

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
