## [Reviewer comments · Royal Society Open Science]

Review History

RSOS-200880.R0 (Original submission)

Review form: Reviewer 1

Do you have any ethical concerns with this paper?

No

Recommendation?

Major revision

Comments to the Author(s)

Summary:

In this registered report, the authors describe a planned study of employees of nursing and care homes (NCH) throughout Italy in order to estimate the self-reported prevalence of moderate to severe anxiety and/or distress among staff working in NCH. As a result of the COVID-19 epidemic in Italy, the authors hypothesize higher levels of anxiety and distress among NCH workers. The authors plan to conduct a cross-sectional survey of NCH workers throughout Italy, stratified by broad geographic region (North, Centre, South Italy). The authors seem to suggest they will use a single-stage cluster sampling design, stratified by geographic region, to sample all

NCH workers from up to 32 different NCH throughout Italy. The authors also plan to weight their sample so it is as representative as possible of the population of Italian NCH workers. Given that residents of NCH/long-term care homes have had a disproportionate share of COVID-19 cases and fatalities across many countries of the world, it is important to assess the mental health of NCH employees who are providing critical care to an especially susceptible population. While I think the planned study is important, I have some concerns about the sample size, sampling methods, and planned statistical analysis that will affect the results of this study. Major comments follow.

Major concerns:

1. In section 3.1, the authors describe the estimated sample size needed to detect an estimated proportion of 35% of NCH employees having moderate to severe self-reported symptoms of anxiety and/or distress. They based this expectation on a prevalence survey of anxiety/distress as reported by healthcare workers exposed to COVID-19 in Chinese hospitals. Given the extent of the COVID-19 epidemic in Italy, this may well be reasonable. However, the formula used to estimate the required sample size with a 5 percentage point margin of error (+/- 5%) doesn't seem to recognize that the authors will be relying on a single-stage cluster design. In these types of designs, observations within the same cluster are not independent, but typically correlated in some way. Thus, the expected sample size should account for this intracluster correlation. If it does not, the expected margin of error may well be greater than +/- 5 percentage points. (Indeed, a general rule of thumb is that simple random samples will require about 400 respondents to estimate a proportion of 0.5 with a margin of error around 5 percentage points). Thus, the expected sample size of 350 may well be too low to obtain a margin of error of 5 percentage points in this particular survey that seems to employ a single stage cluster design. The authors should consider what the expected intracluster correlation might be and how that might influence estimated variances of estimated proportions when they conduct their statistical analysis. Put another way, while 350 employees might be surveyed, as a result of the clustering of employees within NCH and the similarity of employees within a given NCH means that the "effective" sample size is < 350 (for example, if, in a given NCH, personal protective equipment (PPE) is readily available, this would mean that all employees having direct contact with patients have access to the required PPE, meaning these employees are more alike than two randomly selected employees from two randomly selected NCH).
2. Table 1 mentions that a mixture of convenience, snowball, and random sampling methods will be used. However, the methods section implies that probabilistic sampling methods only will be used within three geographic strata. Why does this discrepancy exist? The authors need to consistently describe the sampling methods in both the table and the methods section. Indeed, if the authors only plan to use a single-stage cluster sampling design, then this should be stated as such in Table 1 as well as in the methods section.
3. On page 6, lines 59-60, the authors state they will sample a minimum of 32 NCH from an existing frame of NCH in Italy; 23 of these NCH will be sampled from the North (given 72% of all NCH in Italy are from the Northern region), 6 NCH from the Centre and 3 from the South of Italy. While it's somewhat implied, it would better to explicitly state that these 32 NCH will be randomly selected within each geographic stratum. If NCH are not randomly selected, might the possibility of some type of "selection" bias exist?
4. The authors expect to collect data from 37 employees per nursing home, based on an expected response rate of 70% (as reported in the Chinese study by Lai et al). It's not clear whether an expected 70% response rate is reasonable for Italy, given the vastly different cultures of the two countries. In addition, the Lai et al. study was conducted among healthcare workers, while the planned study is among NCH workers, which may be a more difficult population to survey during the COVID-19 epidemic in Italy. This ultimately could affect the final sample size obtained influencing the statistical reliability of the estimated prevalence of moderate to severe anxiety/distress among NCH throughout Italy. Again, it might be necessary to reconsider the expected sample size for this survey of NCH employees.
5. The main measures the authors will use to estimate the prevalence of self-reported moderate to severe anxiety/distress are the Generalized Anxiety Disorder (GAD-7) scale and the Impact of Event-Scale (revised: IES-R). The authors state both of these scales are widely used in the existing

literature. However, it would be beneficial to report additional information about these scales, such as their reliability and construct validity.

6. In addition to these two scales, respondents will also be asked about their working conditions in the previous two weeks. However, the questions used to assess working conditions (as outlined in the appendix) seem to lack important detail. For example, the amount of anxiety reported by an NCH employee might well be related to the total amount worked in the previous two weeks, including the number of days worked in the previous two weeks as well as the usual length of a typical shift worked in the previous two weeks. Both of these factors could influence levels of anxiety and it would seem to be beneficial to include this additional information. In addition, given the GAD-7 scale asks about anxiety in the last 7 days and the IES-R scale asks about distress in the past two weeks, it would also seem beneficial to ask respondents how many days (out of 7) they work in the past week and how many days (out of 7) they worked in the week prior to the last week.

7. Another question that might be beneficial to ask is whether employees of NCH have access to personal protective equipment (PPE). If so, this could also influence the prevalence of self-reported anxiety/distress. For example, it could well be the case that employees lacking access to PPE experience higher levels of anxiety/distress than employees who have access to PPE. If the data have not yet been collected, it would seem a shame to miss out on an opportunity to ask such a question, since it is potentially related to self-reported levels of anxiety/distress.

8. In the statistical analysis, I wonder whether the objective of this study is to estimate the prevalence of self-reported anxiety/distress overall (among all NCH workers) or by type of work and/or by geographic region and type of worker. The latter seems important information to obtain, as administrative staff not having direct contact with patients might potentially face less risk of contracting SARS-CoV-2 than employees having direct contact with NCH residents. This would then influence overall prevalence (especially if there were an imbalance in the proportion of employees sampled, or, put another way, if the sample is not representative of the types of employees across all NCH in Italy). It's also important to stratify by region, as the North region of Italy was the hardest hit by the COVID-19 epidemic in Italy.

9. The authors state that they will weight their sample to match population proportions of workers from the Italian national healthcare system. The authors will exclude from this database all workers from hospital, university hospitals, and research centres. While this is reasonable (and I can appreciate that the authors are attempting to produce weighted estimates), as a limitation, it would be necessary to consider and potential bias introduced by weighting to a population that could potentially be different than the subset of NCH workers.

10. For the analysis, the authors state that incomplete surveys will be discarded. However, it's not clear what the criteria will be for excluding incomplete surveys. Do the authors mean that all questionnaire items must be complete to consider a survey "complete"? Or do they mean that respondents failing to answer key questions will be considered incomplete (e.g., employee type, hours worked)? Also, given the GAD-7 and IES-R scales employ a larger number of questions, will complete data for these scales be required to be included for analysis? For example the IES-R scale contains 22 items. It's plausible that some respondents may not answer all items, and so useful data may be discarded. (1) The authors need to clearly define what constitutes and "incomplete" survey and (2) whether it might be possible to retain as much data as possible, possibly using multiple imputation methods. This would seem to be especially important for key measures of anxiety/distress (although if the number of exclusions is low, then this is probably less necessary). In addition, having a larger sample size could potentially mean more data from "complete" surveys for statistical analysis.

11. In the statistical analysis, the authors do not seem to recognize that their data are collected from a cluster sampling design and seem to ignore the need to account for the clustering of respondents within NCH. Doing so will produce estimates of variance that are too small and conservative (i.e., narrower) confidence intervals. The clustering of respondents within sampled NCH needs to be recognized in the statistical analysis of these data.

12. The authors assume that individual scores (on the GAD-7 and IES-R scales?) will be "added up and presented as a percentage with a 95% confidence interval" assuming a Poisson distribution. Why is a Poisson distribution assumed rather than assuming a binomial distribution? The Poisson distribution assumes the mean equals the variance, but have the

authors considered whether estimated means might exhibit over- or under-dispersion (such that the mean does not equal the variance)? If so, might a negative binomial assumption be more relevant, or at least used for a sensitivity analysis (of estimated confidence intervals?) More detail should be added to address these issues.

Minor comment:

1. The introduction seems unnecessarily long and could be tightened up to discuss the most important points and to clearly identify the aims of the study more quickly.

Review form: Reviewer 2

Do you have any ethical concerns with this paper?

No

Recommendation?

Reject

Comments to the Author(s)

Please see attached document (Appendix A).

Review form: Reviewer 3 (Claire Nollett)

Do you have any ethical concerns with this paper?

No

Recommendation?

Accept with minor revision

Comments to the Author(s)

The rationale and need for this study is comprehensively argued, the aims are clear and the methods are suitable for the research question. I have the following questions/suggested revisions:

Methods/Science

1. Use of the IES-R: In the instructions you state that the spread of COVID-19 infections in Italy is the stressful event. I wonder a) if this is too vague and participants need to state or at least think about a specific event they have found stressful eg. death of a resident, lack of PPE equipment b) similarly, whether you want to focus this on stress related to working in a nursing home, rather than the spread of COVID-19 in general?

2. Regarding the Interpretation section and recommendations for intervention in Table 1: You seem to suggest that the type of intervention required will depend solely on the prevalence of anxiety and psychological distress symptoms eg. a prevalence of 30-50% would require first and second layers of intervention (psychotherapy and focused psychosocial interventions) and prevalence of 50%+ would need interventions from the first layer (psychotherapy). I have two comments about this. Firstly, does the type of intervention required not depend at least in part on the severity of the symptoms and functional impairment, not just the prevalence? For example, as in an evidence based stepped care model

<https://www.nice.org.uk/guidance/cg113/chapter/Key-priorities-for-implementation>.

Secondly, the more prevalent, surely the more difficult it is to deliver first layer interventions like

psychotherapy for everyone, and interventions from lower levels which can be rolled out more widely should be considered?

3. For the GAD-7, you have included the additional 8th question. As you are making decisions based on cut-off scores using the 7 questions, how will this 8th question be used in the analysis and decision making?
4. You have included a quality of life measure in the appendix but I did not see mention of that in the methods and it doesn't seem to fit your research questions. Is this necessary?
5. Would it be helpful to include a question about access to PPE in your demographics questionnaire?
6. Is there any possibility of having a qualitative element to the the study? I think this would really help to understand the specifics of what might be causing distress in nursing home workers and inform development of suitable interventions.

Ethics

1. Feedback to participants on their scores: I don't fully understand where the traffic light percentages have come from - are these based on the outcome measure scores? For amber, when you say seek 'our' advice, does our refer to the research team? But for red the participant should speak to their centre manager? Will the centre managers be trained in how to respond, how to manage any distress? I think it might be helpful to signpost participants to organisations where they can get support for their mental health should they wish to.
2. As the questionnaire titles are not the 'real' titles, this could be a little misleading - could you leave them out?

Length

1. The introduction is perhaps longer than needed and I feel could be condensed to make it punchier. In particular from page 3 line 10 (all staff...) to line 24. Lines 25-30 are better and sum up the problems succinctly.
2. Again, I think the introduction could be condensed from pg 5 line 1 to 17.

Clarity

1. Pg 3 line 12: turnover - is this turnover of staff? Perhaps make this clearer
2. Pg 3 line 19: You mention phase 1 and 2 here without referring to figure 2. Perhaps refer to it here for clarity, or not mention the phases as this point (too much detail?).

Figures

1. As you refer to the IASC pyramid of support throughout the manuscript, it would be helpful to have a figure showing the pyramid with layers of support. I think it would be more helpful than figure 1 which shows the mortality rates by age, as this doesn't add much more to the text you have written about this.

Typos/missing words

1. Pg 2 line 27: suggest adding 'just a' so it reads 'in just a few months'
2. Pg 4 line 23: I would replace 'at' with 'on' when referring to scores on a scale
3. Pg 6 line 23: In table, 'q' is missing from required
4. Pg 6 line 42: Can the dates be added now where there are ??

Review form: Reviewer 4 (Catherine Hobbs)

Do you have any ethical concerns with this paper?

No

Recommendation?

Major revision

Comments to the Author(s)

The proposed research aims to address an important research question regarding the prevalence of mental health conditions in residential nursing and care home workers during the COVID-19 pandemic. However, several aspects of the design of this study are unclear.

The authors have chosen to focus on anxiety and distress in this sample, but do not offer a rationale for why these symptoms have been chosen over other symptoms experienced widely in the general population (such as depression). Additionally, more consideration is needed as to whether it is appropriate to combine these separate symptoms into a single composite prevalence rate. Previous studies have reported higher prevalence rates for distress than anxiety (e.g. Lai et al., 2020). Finally, all staff members within care homes are invited to participate in this survey. Due to the large variation in job roles it is possible that mental health outcomes would differ according to occupation. Currently, the authors propose to treat these individuals as a homogenous group, but I feel that further justification is required for this approach.

The authors hypothesise that they will observe a prevalence rate of 35% for severe to moderate self-reported mental health symptoms of anxiety and/or distress. There are several issues with the hypothesis in its present state:

- 1) The hypothesis is not operationalised. The authors previously discuss prevalence of severe to moderate self-reported mental health symptoms of anxiety and/or distress, however they do not state which measures or cut-off scores will be used to define this (although this is outlined in later sections).
- 2) It is also unclear whether the hypothesised prevalence rate of 35% relates to the unweighted or weighted prevalence rate analyses later outlined in the pre-registered data analysis.
- 3) Most problematically, it is unclear whether the authors hypothesis of a 35% prevalence rates originated from. In the paper cited (Lai et al., 2020), prevalence rates are reported as follows: depression (50.4%), anxiety (44.6%), insomnia (34%), and distress (71.5%). The confusion may have originated from Lai et al (2020) discussing a 35% prevalence rate for psychological comorbidities within their sample size calculation. However, this estimate refers to a paper looking at survivors of SARS (Lee et al., 2007). In this paper prevalence rates were 36.3% for depressive symptoms and 36.7%, for anxiety.
- 4) Finally, even if the hypothesised prevalence rate was accurately reflected in the cited papers the samples in these papers differs from the target sample of this study. Lai et al (2020) examined health care workers in hospitals (nurses and physicians), and Lee et al (2007) examined survivors of SARS. The rationale for the submitted study is that workers in care homes may be particularly susceptible to psychological difficulties. It therefore seems unconvincing that the prevalence rates in the aforementioned studies can be extrapolated to the target sample for this study.

There are similar issues in the authors' estimations of response rates. They report response rates of 90% in healthcare organisations. However, in the cited materials a mean response rate of 53.8% is reported in health care with a maximum of 94% (Baruch & Holtom, 2008).

The authors provide sufficient materials that would allow reproducibility of their methods. However, the pre-registered data analysis plan is unclear. The authors state that they will calculate the prevalence of anxiety/distress as a first step, and they will then exclude from the database workers who do not hold an administrative, healthcare, technical or professional job

before calculating weighted prevalence rates. However, in the design and recruitment section the authors state that only these individuals will be recruited. It is therefore unclear why these individuals would be retained in the initial prevalence rate calculation. The section on calculating weighted prevalence is rather brief at present which would make replication difficult. Including example analysis code here may be useful.

The materials outlined also include a number of potential moderators not considered in the data analysis plan (e.g. COVID-19 status of care home, engaging in work-related communications, pre-existing staff vs. recruited during the pandemic), increasing the potential for later undisclosed flexible analyses. The authors state that they wish to conduct further exploratory analyses, which presumably include these further measures. However, pre-registered analyses could be considered for these variables.

There are no details in this submission regarding outcome-neutral conditions for ensuring that the results obtained are able to test the stated hypotheses.

As a minor point, there are errors in the GAD-7 functioning question possible responses ('Not difficult at all' and 'Very difficult' are repeated twice).

Review form: Reviewer 5 (Alessandro Massazza)

Do you have any ethical concerns with this paper?

No

Recommendation?

Accept with minor revision

Comments to the Author(s)

Please find my comments in the attached Word file (Appendix B).

Decision letter (RSOS-200880.R0)

Dear Professor Rusconi,

The Editors assigned to your Stage 1 Registered Report ("Prevalence of psychological distress and anxiety among residential nursing and care home workers during the first COVID-19 outbreak in Italy") have now received comments from reviewers. We would like you to revise your paper in accordance with the referee and editors suggestions which can be found below (not including confidential reports to the Editor). Please note this decision does not guarantee eventual acceptance.

When submitting your revised manuscript, you must respond to the comments made by the referees and upload a file "Response to Referees" in "Section 2 - File Upload". Please use this to document how you have responded to the comments, and the adjustments you have made. In order to expedite the processing of the revised manuscript, please be as specific as possible in your response.

Kind regards,
Andrew Dunn
Royal Society Open Science
openscience@royalsociety.org

on behalf of Professor Chris Chambers (Registered Reports Editor, Royal Society Open Science)
openscience@royalsociety.org

Associate Editor Comments to Author (Professor Chris Chambers):

Associate Editor: 1

Comments to the Author:

Five reviewers with a range of specialist expertise have now assessed the Stage 1 manuscript. All find merit in the study, albeit to varying degrees, while also raising a wide range of issues that will need to be addressed to achieve Stage 1 acceptance. Major concerns are identified with the rationale for the research question (including the focus on anxiety), the clarity and precision of the hypothesis, justification and sufficiency of the planned sample size, sampling methods and statistical analyses, justification of the prospective interpretation, inclusion of data quality checks, and overall feasibility (particularly in achieving a sufficiently high survey response rate). The reviews are highly detailed and constructive, and although serious concerns are raised, I judge that they fall within an amendable scope of a Stage 1 RR. On this basis, a Major Revision is invited.

My sincere thanks to each of the reviewers for providing such detailed and high-quality assessments on such a rapid timescale.

Comments to Author:

Reviewer: 1

Comments to the Author(s)

Summary:

In this registered report, the authors describe a planned study of employees of nursing and care homes (NCH) throughout Italy in order to estimate the self-reported prevalence of moderate to severe anxiety and/or distress among staff working in NCH. As a result of the COVID-19 epidemic in Italy, the authors hypothesize higher levels of anxiety and distress among NCH workers. The authors plan to conduct a cross-sectional survey of NCH workers throughout Italy, stratified by broad geographic region (North, Centre, South Italy). The authors seem to suggest they will use a single-stage cluster sampling design, stratified by geographic region, to sample all NCH workers from up to 32 different NCH throughout Italy. The authors also plan to weight their sample so it is as representative as possible of the population of Italian NCH workers. Given that residents of NCH/long-term care homes have had a disproportionate share of COVID-19 cases and fatalities across many countries of the world, it is important to assess the mental health of NCH employees who are providing critical care to an especially susceptible population. While I think the planned study is important, I have some concerns about the sample size, sampling methods, and planned statistical analysis that will affect the results of this study. Major comments follow.

Major concerns:

1. In section 3.1, the authors describe the estimated sample size needed to detect an estimated proportion of 35% of NCH employees having moderate to severe self-reported symptoms of anxiety and/or distress. They based this expectation on a prevalence survey of anxiety/distress as reported by healthcare workers exposed to COVID-19 in Chinese hospitals. Given the extent of the COVID-19 epidemic in Italy, this may well be reasonable. However, the formula used to estimate the required sample size with a 5 percentage point margin of error (+/- 5%) doesn't seem to recognize that the authors will be relying on a single-stage cluster design. In these types of designs, observations within the same cluster are not independent, but typically correlated in some way. Thus, the expected sample size should account for this intracluster correlation. If it does not, the expected margin of error may well be greater than +/- 5 percentage points. (Indeed, a general rule of thumb is that simple random samples will require about 400 respondents to estimate a proportion of 0.5 with a margin of error around 5 percentage points). Thus, the expected sample size of 350 may well be too low to obtain a margin of error of 5 percentage points in this particular survey that seems to employ a single stage cluster design. The authors should consider what the expected intracluster correlation might be and how that might influence estimated variances of estimated proportions when they conduct their statistical analysis. Put another way, while 350 employees might be surveyed, as a result of the clustering of employees within NCH and the similarity of employees within a given NCH means that the "effective" sample size is < 350 (for example, if, in a given NCH, personal protective equipment (PPE) is readily available, this would mean that all employees having direct contact with patients have access to the required PPE, meaning these employees are more alike than two randomly selected employees from two randomly selected NCH).
2. Table 1 mentions that a mixture of convenience, snowball, and random sampling methods will be used. However, the methods section implies that probabilistic sampling methods only will be used within three geographic strata. Why does this discrepancy exist? The authors need to consistently describe the sampling methods in both the table and the methods section. Indeed, if the authors only plan to use a single-stage cluster sampling design, then this should be stated as such in Table 1 as well as in the methods section.
3. On page 6, lines 59-60, the authors state they will sample a minimum of 32 NCH from an existing frame of NCH in Italy; 23 of these NCH will be sampled from the North (given 72% of all NCH in Italy are from the Northern region), 6 NCH from the Centre and 3 from the South of Italy. While it's somewhat implied, it would better to explicitly state that these 32 NCH will be randomly selected within each geographic stratum. If NCH are not randomly selected, might the possibility of some type of "selection" bias exist?
4. The authors expect to collect data from 37 employees per nursing home, based on an expected response rate of 70% (as reported in the Chinese study by Lai et al). It's not clear whether an expected 70% response rate is reasonable for Italy, given the vastly different cultures of the two countries. In addition, the Lai et al. study was conducted among healthcare workers, while the planned study is among NCH workers, which may be a more difficult population to survey during the COVID-19 epidemic in Italy. This ultimately could affect the final sample size obtained influencing the statistical reliability of the estimated prevalence of moderate to severe anxiety/distress among NCH throughout Italy. Again, it might be necessary to reconsider the expected sample size for this survey of NCH employees.
5. The main measures the authors will use to estimate the prevalence of self-reported moderate to severe anxiety/distress are the Generalized Anxiety Disorder (GAD-7) scale and the Impact of Event-Scale (revised: IES-R). The authors state both of these scales are widely used in the existing literature. However, it would be beneficial to report additional information about these scales, such as their reliability and construct validity.
6. In addition to these two scales, respondents will also be asked about their working conditions in the previous two weeks. However, the questions used to assess working conditions (as outlined in the appendix) seem to lack important detail. For example, the amount of anxiety reported by an NCH employee might well be related to the total amount worked in the previous two weeks, including the number of days worked in the previous two weeks as well as the usual length of a typical shift worked in the previous two weeks. Both of these factors could influence

levels of anxiety and it would seem to be beneficial to include this additional information. In addition, given the GAD-7 scale asks about anxiety in the last 7 days and the IES-R scale asks about distress in the past two weeks, it would also seem beneficial to ask respondents how many days (out of 7) they work in the past week and how many days (out of 7) they worked in the week prior to the last week.

7. Another question that might be beneficial to ask is whether employees of NCH have access to personal protective equipment (PPE). If so, this could also influence the prevalence of self-reported anxiety/distress. For example, it could well be the case that employees lacking access to PPE experience higher levels of anxiety/distress than employees who have access to PPE. If the data have not yet been collected, it would seem a shame to miss out on an opportunity to ask such a question, since it is potentially related to self-reported levels of anxiety/distress.

8. In the statistical analysis, I wonder whether the objective of this study is to estimate the prevalence of self-reported anxiety/distress overall (among all NCH workers) or by type of work and/or by geographic region and type of worker. The latter seems important information to obtain, as administrative staff not having direct contact with patients might potentially face less risk of contracting SARS-CoV-2 than employees having direct contact with NCH residents. This would then influence overall prevalence (especially if there were an imbalance in the proportion of employees sampled, or, put another way, if the sample is not representative of the types of employees across all NCH in Italy). It's also important to stratify by region, as the North region of Italy was the hardest hit by the COVID-19 epidemic in Italy.

9. The authors state that they will weight their sample to match population proportions of workers from the Italian national healthcare system. The authors will exclude from this database all workers from hospital, university hospitals, and research centres. While this is reasonable (and I can appreciate that the authors attempt to produce weighted estimates), as a limitation, it would be necessary to consider and potential bias introduced by weighting to a population that could potentially be different than the subset of NCH workers.

10. For the analysis, the authors state that incomplete surveys will be discarded. However, it's not clear what the criteria will be for excluding incomplete surveys. Do the authors mean that all questionnaire items must be complete to consider a survey "complete"? Or do they mean that respondents failing to answer key questions will be considered incomplete (e.g., employee type, hours worked)? Also, given the GAD-7 and IES-R scales employ a larger number of questions, will complete data for these scales be required to be included for analysis? For example the IES-R scale contains 22 items. It's plausible that some respondents may not answer all items, and so useful data may be discarded. (1) The authors need to clearly define what constitutes and "incomplete" survey and (2) whether it might be possible to retain as much data as possible, possibly using multiple imputation methods. This would seem to be especially important for key measures of anxiety/distress (although if the number of exclusions is low, then this is probably less necessary). In addition, having a larger sample size could potentially mean more data from "complete" surveys for statistical analysis.

11. In the statistical analysis, the authors do not seem to recognize that their data are collected from a cluster sampling design and seem to ignore the need to account for the clustering of respondents within NCH. Doing so will produce estimates of variance that are too small and conservative (i.e., narrower) confidence intervals. The clustering of respondents within sampled NCH needs to be recognized in the statistical analysis of these data.

12. The authors assume that individual scores (on the GAD-7 and IES-R scales?) will be "added up and presented as a percentage with a 95% confidence interval" assuming a Poisson distribution. Why is a Poisson distribution assumed rather than assuming a binomial distribution? The Poisson distribution assumes the mean equals the variance, but have the authors considered whether estimated means might exhibit over- or under-dispersion (such that the mean does not equal the variance)? If so, might a negative binomial assumption be more relevant, or at least used for a sensitivity analysis (of estimated confidence intervals?) More detail should be added to address these issues.

Minor comment:

1. The introduction seems unnecessarily long and could be tightened up to discuss the most important points and to clearly identify the aims of the study more quickly.

Reviewer: 2

Comments to the Author(s)

Please see attached document.

Reviewer: 3

Comments to the Author(s)

The rationale and need for this study is comprehensively argued, the aims are clear and the methods are suitable for the research question. I have the following questions/suggested revisions:

Methods/Science

1. Use of the IES-R: In the instructions you state that the spread of COVID-19 infections in Italy is the stressful event. I wonder a) if this is too vague and participants need to state or at least think about a specific event they have found stressful eg. death of a resident, lack of PPE equipment b) similarly, whether you want to focus this on stress related to working in a nursing home, rather than the spread of COVID-19 in general?

2. Regarding the Interpretation section and recommendations for intervention in Table 1: You seem to suggest that the type of intervention required will depend solely on the prevalence of anxiety and psychological distress symptoms eg. a prevalence of 30-50% would require first and second layers of intervention (psychotherapy and focused psychosocial interventions) and prevalence of 50%+ would need interventions from the first layer (psychotherapy). I have two comments about this. Firstly, does the type of intervention required not depend at least in part on the severity of the symptoms and functional impairment, not just the prevalence? For example, as in an evidence based stepped care model

<https://www.nice.org.uk/guidance/cg113/chapter/Key-priorities-for-implementation>.

Secondly, the more prevalent, surely the more difficult it is to deliver first layer interventions like psychotherapy for everyone, and interventions from lower levels which can be rolled out more widely should be considered?

3. For the GAD-7, you have included the additional 8th question. As you are making decisions based on cut-off scores using the 7 questions, how will this 8th question be used in the analysis and decision making?

4. You have included a quality of life measure in the appendix but I did not see mention of that in the methods and it doesn't seem to fit your research questions. Is this necessary?

5. Would it be helpful to include a question about access to PPE in your demographics questionnaire?

6. Is there any possibility of having a qualitative element to the the study? I think this would really help to understand the specifics of what might be causing distress in nursing home workers and inform development of suitable interventions.

Ethics

1. Feedback to participants on their scores: I don't fully understand where the traffic light percentages have come from - are these based on the outcome measure scores? For amber, when you say seek 'our' advice, does our refer to the research team? But for red the participant should speak to their centre manager? Will the centre managers be trained in how to respond, how to manage any distress? I think it might be helpful to signpost participants to organisations where they can get support for their mental health should they wish to.

2. As the questionnaire titles are not the 'real' titles, this could be a little misleading - could you leave them out?

Length

1. The introduction is perhaps longer than needed and I feel could be condensed to make it punchier. In particular from page 3 line 10 (all staff...) to line 24. Lines 25-30 are better and sum up the problems succinctly.
2. Again, I think the introduction could be condensed from pg 5 line 1 to 17.

Clarity

1. Pg 3 line 12: turnover - is this turnover of staff? Perhaps make this clearer
2. Pg 3 line 19: You mention phase 1 and 2 here without referring to figure 2. Perhaps refer to it here for clarity, or not mention the phases as this point (too much detail?).

Figures

1. As you refer to the IASC pyramid of support throughout the manuscript, it would be helpful to have a figure showing the pyramid with layers of support. I think it would be more helpful than figure 1 which shows the mortality rates by age, as this doesn't add much more to the text you have written about this.

Typos/missing words

1. Pg 2 line 27: suggest adding 'just a' so it reads 'in just a few months'
2. Pg 4 line 23: I would replace 'at' with 'on' when referring to scores on a scale
3. Pg 6 line 23: In table, 'q' is missing from required
4. Pg 6 line 42: Can the dates be added now where there are ??

Reviewer: 4

Comments to the Author(s)

The proposed research aims to address an important research question regarding the prevalence of mental health conditions in residential nursing and care home workers during the COVID-19 pandemic. However, several aspects of the design of this study are unclear. The authors have chosen to focus on anxiety and distress in this sample, but do not offer a rationale for why these symptoms have been chosen over other symptoms experienced widely in the general population (such as depression). Additionally, more consideration is needed as to whether it is appropriate to combine these separate symptoms into a single composite prevalence rate. Previous studies have reported higher prevalence rates for distress than anxiety (e.g. Lai et al., 2020). Finally, all staff members within care homes are invited to participate in this survey. Due to the large variation in job roles it is possible that mental health outcomes would differ according to occupation. Currently, the authors propose to treat these individuals as a homogenous group, but I feel that further justification is required for this approach.

The authors hypothesise that they will observe a prevalence rate of 35% for severe to moderate self-reported mental health symptoms of anxiety and/or distress. There are several issues with the hypothesis in its present state:

- 1) The hypothesis is not operationalised. The authors previously discuss prevalence of severe to moderate self-reported mental health symptoms of anxiety and/or distress, however they do not state which measures or cut-off scores will be used to define this (although this is outlined in later sections).
- 2) It is also unclear whether the hypothesised prevalence rate of 35% relates to the unweighted or weighted prevalence rate analyses later outlined in the pre-registered data analysis.
- 3) Most problematically, it is unclear whether the authors hypothesis of a 35% prevalence rates originated from. In the paper cited (Lai et al., 2020), prevalence rates are reported as follows: depression (50.4%), anxiety (44.6%), insomnia (34%), and distress (71.5%). The confusion may

have originated from Lai et al (2020) discussing a 35% prevalence rate for psychological comorbidities within their sample size calculation. However, this estimate refers to a paper looking at survivors of SARS (Lee et al., 2007). In this paper prevalence rates were 36.3% for depressive symptoms and 36.7%, for anxiety.

4) Finally, even if the hypothesised prevalence rate was accurately reflected in the cited papers the samples in these papers differs from the target sample of this study. Lai et al (2020) examined health care workers in hospitals (nurses and physicians), and Lee et al (2007) examined survivors of SARS. The rationale for the submitted study is that workers in care homes may be particularly susceptible to psychological difficulties. It therefore seems unconvincing that the prevalence rates in the aforementioned studies can be extrapolated to the target sample for this study.

There are similar issues in the authors' estimations of response rates. They report response rates of 90% in healthcare organisations. However, in the cited materials a mean response rate of 53.8% is reported in health care with a maximum of 94% (Baruch & Holtom, 2008).

The authors provide sufficient materials that would allow reproducibility of their methods. However, the pre-registered data analysis plan is unclear. The authors state that they will calculate the prevalence of anxiety/distress as a first step, and they will then exclude from the database workers who do not hold an administrative, healthcare, technical or professional job before calculating weighted prevalence rates. However, in the design and recruitment section the authors state that only these individuals will be recruited. It is therefore unclear why these individuals would be retained in the initial prevalence rate calculation. The section on calculating weighted prevalence is rather brief at present which would make replication difficult. Including example analysis code here may be useful.

The materials outlined also include a number of potential moderators not considered in the data analysis plan (e.g. COVID-19 status of care home, engaging in work-related communications, pre-existing staff vs. recruited during the pandemic), increasing the potential for later undisclosed flexible analyses. The authors state that they wish to conduct further exploratory analyses, which presumably include these further measures. However, pre-registered analyses could be considered for these variables.

There are no details in this submission regarding outcome-neutral conditions for ensuring that the results obtained are able to test the stated hypotheses.

As a minor point, there are errors in the GAD-7 functioning question possible responses ('Not difficult at all' and 'Very difficult' are repeated twice).

Reviewer: 5

Comments to the Author(s)

Please find my comments in the attached Word file.

Author's Response to Decision Letter for (RSOS-200880.R0)

See Appendix C.

RSOS-200880.R1 (Revision)

Review form: Reviewer 1

Do you have any ethical concerns with this paper?

No

Recommendation?

Accept in principle

Comments to the Author(s)

Thank-you for revising the manuscript and for providing sufficient responses and revisions to all of my original concerns.

Review form: Reviewer 2

Do you have any ethical concerns with this paper?

No

Recommendation?

Accept with minor revision

Comments to the Author(s)

The authors have provided a fast revision, with thoughtful responses to my comments and those of the other (4!) reviewers. I consider the authors' responses to many of my points to be satisfactory, and have therefore focused this review on what I see as the remaining issues.

R5#1 (my comments regarding the utility of the study): Some of what the authors write in their response seems very sensible. However, in their response letter they state "4) the present study will partially help to disclose the causal effect of COVID-19 on distress or anxiety for this population." This is not the case; in Cook and Campbell's terminology (Shadish, Cook, & Campbell, 2001), the study is a "one-shot case study". The authors won't be able to establish a correlation between anxiety or distress and COVID-19, and certainly can't establish a causal effect (not even tentatively - no statistic in the planned analysis is an estimator of a causal effect). While the causal inference does not seem to be implied in the current manuscript (just the response letter) it is important that the authors do not claim that the study tells us something about the causal effect of COVID-19 in either the discussion section of the eventual manuscript or any ensuing media coverage.

R5#7: I appreciate the author's useful revisions to the instructions to the participants for the IES-R. However, it remains the case that the questions in the IES-R ask participants about a single event (e.g., "pictures about it popped into my mind"), while the customised instructions relate to a (plural) set of multiple possible events ("any stressful events in your nursing/care home due to the recent COVID-19 pandemic"). The customised instructions thus don't quite fit the items grammatically. Whether this problem is obviated in the Italian translation, I don't know. I'm also left a little unsure whether participants will have a clear idea of which events they're meant to be reflecting on given the rather open-ended instructions.

As per my comment in the first round there is still inconsistency within the manuscript in relation to what the IES-R is meant to be measuring ("distress", "psychological distress", "post-traumatic stress").

R5#16: I appreciate the author's attempt to add some more nuance to the interpretation scheme in Table 1, but my concern here remains. Mental health varies from person-to-person. For example, even with a prevalence of moderate distress/anxiety under 10%, some individuals may still experience severe psychological dysfunction and require structured and intensive psychological (or psychiatric) interventions. I don't believe that the inferences the authors are seeking to draw about which services should be offered are ones that logically follow from the statistic they plan to report (i.e., a prevalence estimate). I leave this to the editor's judgment but my recommendation is that this interpretation scheme be removed.

R5#21: I can appreciate the author's attention to the feasibility of the sample size planning, but this wasn't quite what I was asking for (perhaps I didn't express myself entirely clearly). In a registered report, the inferential criteria need to make it clear how a failure to achieve the target sample size will affect the inferences drawn. I.e., will the study still be reported if the sample size is less than 900? How would this affect how you describe the results? (The editor may have suggestions here).

References

Shadish, W. R., Cook, T. D., & Campbell, D. T. (2001). *Experimental and quasi-experimental designs for generalized causal inference* (2nd ed.). Belmont, California: Wadsworth.

Review form: Reviewer 4 (Catherine Hobbs)

Do you have any ethical concerns with this paper?

No

Recommendation?

Accept with minor revision

Comments to the Author(s)

Thank you to the authors for taking the time to address my concerns in detail, and for clarifying the sources of the estimated prevalence rate. The majority of my previous concerns have been addressed, however I believe it may still be worth considering whether it would be appropriate to report anxiety and distress prevalence rates separately. The authors have provided a clear rationale in the introduction as to why they have chosen to focus on anxiety and distress symptoms. However, there is little detail for the decision to combine these into a single prevalence rate and why this is more beneficial than considering these symptoms independently. As similar studies in this area (e.g. Lai et al, 2020) have reported distress and anxiety separately I think it would be worth outlining the decision process for this.

As a minor point, the introduction would benefit from further refinement at times. For example, the last paragraph on page 26 of the submission, line 38 onwards, the authors conclude 'This has had inevitable consequences for the management of residents'. However, it is unclear exactly what exactly these inevitable consequences are.

There are also some minor typos in the English translation of study materials:

- Item 7 of the GAD-7 English translation, 'MORE THAN AHLF THE DAYS'
- Question 3 on the quality of life scale, 'Where you productive (comleting many tasks) in most activities'.

Review form: Reviewer 5 (Alessandro Massazza)

Do you have any ethical concerns with this paper?

No

Recommendation?

Accept with minor revision

Comments to the Author(s)

See attached file (Appendix D).

Decision letter (RSOS-200880.R1)

Dear Professor Rusconi,

On behalf of the Editors, I am pleased to inform you that your Manuscript RSOS-200880.R1 entitled "Prevalence of psychological distress and anxiety among residential nursing and care home workers following the COVID-19 outbreak in Northern Italy" has been accepted in principle for publication in Royal Society Open Science subject to minor revision in accordance with the referee and editor suggestions. Please find their comments at the end of this email.

The reviewers and handling editors have recommended publication, but also suggest some minor revisions to your manuscript. Therefore, I invite you to respond to the comments and revise your manuscript.

Please you submit the revised version of your manuscript within 7 days (i.e. by the 11-Jun-2020). If you do not think you will be able to meet this date please let me know immediately.

When submitting your revised manuscript, you will be able to respond to the comments made by the referees and you should upload a file "Response to Referees". You can use this to document any changes you make to the original manuscript. In order to expedite the processing of the revised manuscript, please be as specific as possible in your response to the referees.

Full author guidelines can be found here <https://royalsocietypublishing.org/rsos/registered-reports>.

Kind regards,
Lianne Parkhouse

Editorial Coordinator
 Royal Society Open Science
 openscience@royalsociety.org

on behalf of Professor Chris Chambers (Subject Editor, Royal Society Open Science)
 openscience@royalsociety.org

Associate Editor Comments to Author (Professor Chris Chambers):

Four of the original reviewers have now assessed the manuscript. All agree that the proposal is significantly improved, recommending either in-principle acceptance (IPA) or minor revision. There are some remaining issues to address, primarily in to maximise clarity of the presentation/terminology and certain design features (e.g. inference criteria), as well as addressing questions surrounding the potential interpretation of the results. Provided the authors can respond comprehensively to these remaining issues in a final revision, IPA should be forthcoming without requiring further in-depth Stage 1 review.

Reviewer comments to Author:

Reviewer: 1
 Comments to the Author(s)

Thank-you for revising the manuscript and for providing sufficient responses and revisions to all of my original concerns.

Reviewer: 2
 Comments to the Author(s)

The authors have provided a fast revision, with thoughtful responses to my comments and those of the other (4!) reviewers. I consider the authors' responses to many of my points to be satisfactory, and have therefore focused this review on what I see as the remaining issues.

R5#1 (my comments regarding the utility of the study): Some of what the authors write in their response seems very sensible. However, in their response letter they state "4) the present study will partially help to disclose the causal effect of COVID-19 on distress or anxiety for this population." This is not the case; in Cook and Campbell's terminology (Shadish, Cook, & Campbell, 2001), the study is a "one-shot case study". The authors won't be able to establish a correlation between anxiety or distress and COVID-19, and certainly can't establish a causal effect (not even tentatively - no statistic in the planned analysis is an estimator of a causal effect). While the causal inference does not seem to be implied in the current manuscript (just the response letter) it is important that the authors do not claim that the study tells us something about the causal effect of COVID-19 in either the discussion section of the eventual manuscript or any ensuing media coverage.

R5#7: I appreciate the author's useful revisions to the instructions to the participants for the IES-R. However, it remains the case that the questions in the IES-R ask participants about a single event (e.g., "pictures about it popped into my mind"), while the customised instructions relate to a (plural) set of multiple possible events ("any stressful events in your nursing/care home due to the recent COVID-19 pandemic"). The customised instructions thus don't quite fit the items grammatically. Whether this problem is obviated in the Italian translation, I don't know. I'm also left a little unsure whether participants will have a clear idea of which events they're meant to be reflecting on given the rather open-ended instructions.

As per my comment in the first round there is still inconsistency within the manuscript in relation to what the IES-R is meant to be measuring (“distress”, “psychological distress”, “post-traumatic stress”).

R5#16: I appreciate the author’s attempt to add some more nuance to the interpretation scheme in Table 1, but my concern here remains. Mental health varies from person-to-person. For example, even with a prevalence of moderate distress/anxiety under 10%, some individuals may still experience severe psychological dysfunction and require structured and intensive psychological (or psychiatric) interventions. I don’t believe that the inferences the authors are seeking to draw about which services should be offered are ones that logically follow from the statistic they plan to report (i.e., a prevalence estimate). I leave this to the editor’s judgment but my recommendation is that this interpretation scheme be removed.

R5#21: I can appreciate the author’s attention to the feasibility of the sample size planning, but this wasn’t quite what I was asking for (perhaps I didn’t express myself entirely clearly). In a registered report, the inferential criteria need to make it clear how a failure to achieve the target sample size will affect the inferences drawn. I.e., will the study still be reported if the sample size is less than 900? How would this affect how you describe the results? (The editor may have suggestions here).

References

Shadish, W. R., Cook, T. D., & Campbell, D. T. (2001). *Experimental and quasi-experimental designs for generalized causal inference* (2nd ed.). Belmont, California: Wadsworth.

Reviewer: 4

Comments to the Author(s)

Thank you to the authors for taking the time to address my concerns in detail, and for clarifying the sources of the estimated prevalence rate. The majority of my previous concerns have been addressed, however I believe it may still be worth considering whether it would be appropriate to report anxiety and distress prevalence rates separately. The authors have provided a clear rationale in the introduction as to why they have chosen to focus on anxiety and distress symptoms. However, there is little detail for the decision to combine these into a single prevalence rate and why this is more beneficial than considering these symptoms independently. As similar studies in this area (e.g. Lai et al, 2020) have reported distress and anxiety separately I think it would be worth outlining the decision process for this.

As a minor point, the introduction would benefit from further refinement at times. For example, the last paragraph on page 26 of the submission, line 38 onwards, the authors conclude ‘This has had inevitable consequences for the management of residents’. However, it is unclear exactly what exactly these inevitable consequences are.

There are also some minor typos in the English translation of study materials:

- Item 7 of the GAD-7 English translation, ‘MORE THAN A HALF THE DAYS’
- Question 3 on the quality of life scale, ‘Where you productive (comleting many tasks) in most activities’.

Reviewer: 5

Comments to the Author(s)

See attached file.

Author's Response to Decision Letter for (RSOS-200880.R1)

See Appendix E.

Decision letter (RSOS-200880.R2)

Dear Professor Rusconi

On behalf of the Editor, I am pleased to inform you that your Manuscript RSOS-200880.R2 entitled "Prevalence of post-traumatic symptomatology and anxiety in residential nursing and care home workers following the COVID-19 outbreak in Northern Italy" has been accepted in principle for publication in Royal Society Open Science.

You may now progress to Stage 2 and complete the study as approved.

Please read the following email carefully

Your accepted Stage 1 manuscript has been registered under the requested 4-year private embargo at: <https://osf.io/4fpqa>

This embargo will be released, and the accepted Stage 1 manuscript made public, at the point of Stage 2 submission or manuscript withdrawal.

Following completion of your study, we invite you to resubmit your paper for peer review as a Stage 2 Registered Report. Please note that your manuscript can still be rejected for publication at Stage 2 if the Editors consider any of the following conditions to be met:

- The results were unable to test the authors' proposed hypotheses by failing to meet the approved outcome-neutral criteria.
- The authors altered the Introduction, rationale, or hypotheses, as approved in the Stage 1 submission.
- The authors failed to adhere closely to the registered experimental procedures. Please note that any deviations from the approved experimental procedures must be communicated to the editor immediately for approval, and prior to the completion of data collection. Failure to do so can result in revocation of in-principle acceptance and rejection at Stage 2 (see complete guidelines for further information).
- Any post-hoc (unregistered) analyses were either unjustified, insufficiently caveated, or overly dominant in shaping the authors' conclusions.
- The authors' conclusions were not justified given the data obtained.

We encourage you to read the complete guidelines for authors concerning Stage 2 submissions at <https://royalsocietypublishing.org/rsos/registered-reports#ReviewerGuideRegRep>. Please especially note the requirements for data sharing, reporting the URL of the independently registered protocol, and that withdrawing your manuscript will result in publication of a Withdrawn Registration.

Once again, thank you for submitting your manuscript to Royal Society Open Science and we look forward to receiving your Stage 2 submission. If you have any questions at all, please do not hesitate to get in touch. We look forward to hearing from you shortly with the anticipated submission date for your stage two manuscript.

on behalf of Professor Chris Chambers (Registered Reports Editor, Royal Society Open Science)
openscience@royalsociety.org

Author's Response to Decision Letter for (RSOS-200880.R2)

See Appendix F.

RSOS-200880.R3 (Revision)

Review form: Reviewer 2

Is the manuscript scientifically sound in its present form?

Yes

Are the interpretations and conclusions justified by the results?

Yes

Is the language acceptable?

Yes

Do you have any ethical concerns with this paper?

No

Have you any concerns about statistical analyses in this paper?

Yes

Recommendation?

Major revision

Comments to the Author(s)

Please see the attached Word document (Appendix G).

Review form: Reviewer 4 (Catherine Hobbs)

Is the manuscript scientifically sound in its present form?

Yes

Are the interpretations and conclusions justified by the results?

Yes

Is the language acceptable?

Yes

Do you have any ethical concerns with this paper?

No

Have you any concerns about statistical analyses in this paper?

No

Recommendation?

Accept with minor revision

Comments to the Author(s)

The introduction, rationale and stated hypotheses are the same as the approved stage 1 submission, and the authors adhered to the registered procedures. The data collected by the authors adequately allowed them to test their proposed hypotheses. The unregistered exploratory statistical analyses seem justified. I feel that the discussion could be more concise and at times more objective (e.g. referring to organisations that chose not to participate as 'less virtuous' seems speculative). The confidence intervals for the effect of gender on anxiety in the logistic regression model came close to overlapping with the null. Highlighting the strength of evidence for this particular effect would be useful.

Review form: Reviewer 5 (Alessandro Massazza)

Is the manuscript scientifically sound in its present form?

Yes

Are the interpretations and conclusions justified by the results?

Yes

Is the language acceptable?

Yes

Do you have any ethical concerns with this paper?

No

Have you any concerns about statistical analyses in this paper?

No

Recommendation?

Major revision

Comments to the Author(s)

See Word file attached (Appendix H).

Decision letter (RSOS-200880.R3)

Dear Professor Rusconi,

The editors assigned to your paper ("Prevalence of post-traumatic symptomatology and anxiety in residential nursing and care home workers following the COVID-19 outbreak in Northern Italy") has now received comments from reviewers. We would like you to revise your paper in accordance with the referee and Subject Editor suggestions which can be found below (not including confidential reports to the Editor).

Please submit a copy of your revised paper within three weeks (i.e. by the 02-Oct-2020).

- Data accessibility

If you wish to submit your supporting data or code to Dryad (<http://datadryad.org/>), or modify your current submission to dryad, please use the following link:
<http://datadryad.org/submit?journalID=RSOS&manu=RSOS-200880.R3>

- Competing interests

- Authors' contributions

- Acknowledgements

- Funding statement

Kind regards,
Andrew Dunn
Royal Society Open Science Editorial Office
Royal Society Open Science

on behalf of Chris Chambers
Subject Editor, Royal Society Open Science
openscience@royalsociety.org

Associate Editor's comments (Professor Chris Chambers):

Associate Editor: 1

Comments to the Author:

Three of the original Stage 1 reviewers were available to assess the Stage 2 manuscript. The overall assessment is positive, but the reviewers raises a number of issues that will need to be addressed to achieve Stage 2 acceptance. Reviewer 2 questions the suitability and robustness of the extensive exploratory analyses, suggesting that the majority of them are cut or consigned to supplementary information (whereas Reviewer 4 feels they are justified). Please consider these concerns carefully. Reviewer 2 is broadly correct that the preregistered outcomes should be centered in a Stage 2 RR; however, exploratory analyses are welcomed where they are methodologically sound and appropriately caveated. Reviewer is concerned that neither of these conditions are met for at least some of the analyses reported.

The second substantive point to address is the collapsing of PTSD and anxiety into a single prevalence rate (Reviewer 5). The reviewer argues that these should be reported separately (with the separate findings highlighted). However, since the collapsed approach is part of the preregistered design, please take care in responding to this point that you do not alter the approved preregistered part of the Stage 1 analysis plan. This point should therefore be

addressed in the exploratory analyses and Discussion (or in a suitable rebuttal), but sections of the manuscript that report preregistered analyses should not be removed.

Please also ensure that the OSF project is made available to the reviewers in a revision by either making the project public or providing a private view-only link in the manuscript if you prefer to keep the content embargoed until acceptance. Please also supply data files in non-proprietary format, and code in a format that can be easily read outside STATA (there is no need to re-run the analyses in R, as Reviewer 2 suggests, but the content of the code and data should be observable without needing STATA).

Given the substantive issues raised, it is likely that a revised manuscript will be returned to at least one reviewer for another look, provided reviewers are available on a similarly tight timescale.

Comments to Author:

Reviewer: 2

Comments to the Author(s)

Please see the attached Word document.

Reviewers' Comments to Author:

Reviewer: 4

Comments to the Author(s)

The introduction, rationale and stated hypotheses are the same as the approved stage 1 submission, and the authors adhered to the registered procedures. The data collected by the authors adequately allowed them to test their proposed hypotheses. The unregistered exploratory statistical analyses seem justified. I feel that the discussion could be more concise and at times more objective (e.g. referring to organisations that chose not to participate as 'less virtuous' seems speculative). The confidence intervals for the effect of gender on anxiety in the logistic regression model came close to overlapping with the null. Highlighting the strength of evidence for this particular effect would be useful.

Reviewer: 5

Comments to the Author(s)

See Word file attached.

Author's Response to Decision Letter for (RSOS-200880.R3)

See Appendix I.

Decision letter (RSOS-200880.R4)

Dear Professor Rusconi:

It is a pleasure to accept your manuscript entitled "Prevalence of post-traumatic symptomatology and anxiety in residential nursing and care home workers following the COVID-19 outbreak in Northern Italy" in its current form for publication in Royal Society Open Science.

COVID-19 rapid publication process:

We are taking steps to expedite the publication of research relevant to the pandemic. If you wish, you can opt to have your paper published as soon as it is ready, rather than waiting for it to be published the scheduled Wednesday.

This means your paper will not be included in the weekly media round-up which the Society sends to journalists ahead of publication. However, it will still appear in the COVID-19 Publishing Collection which journalists will be directed to each week (<https://royalsocietypublishing.org/topic/special-collections/novel-coronavirus-outbreak>).

If you wish to have your paper considered for immediate publication, or to discuss further, please notify openscience_proofs@royalsociety.org and press@royalsociety.org when you respond to this email.

on behalf of Professor Chris Chambers (Subject Editor)
openscience@royalsociety.org

Appendix A

Review of RSOS-200880 (Stage 1 Registered Report)

Article title: Prevalence of psychological distress and anxiety among residential nursing and care home workers during the first COVID-19 outbreak in Italy.

In this Stage 1 Registered Report, the authors set out a plan for a study estimating the prevalence of moderate to severe psychological distress and anxiety in residential and care home workers in Italy. This is a population that must surely be experiencing great stress during the COVID-19 pandemic, and I think it's fantastic that the authors wish to do something to help. I'm also always pleased to see the RR format being used. I nevertheless do have some major concerns regarding the planned study. I have structured my comments according to the topics in the peer review guidelines.

The scientific validity of the research question(s)

The research question of this study is “What is the prevalence of severe-to-moderate self-reported mental health symptoms of anxiety and/or distress among staff in NCH during the first COVID-19 outbreak in Italy?”

This research question seems reasonably clearly specified. However, I'm somewhat unconvinced that having an answer to it would be of significant scientific or practical utility. All we would learn from the study is that *at one point in time*, some percentage of a sample of residential nursing and care homes in Italy experienced moderate or greater distress or anxiety. We would not know anything about the causal *effect* of COVID-19 on distress or anxiety for this population. Nor would we know whether this population experiences higher levels of distress than other populations. We wouldn't even really know what levels of anxiety or distress might still be present in this population at the time the results are released to decision-makers who might *do* something to address this anxiety and distress. The authors say “these data [...] will also provide a useful guide for the implementation of timely evidence-based psychological interventions and long-term follow up assessments for NCH workers”—but I struggle to see how that could be the case since these are not things that the planned study directly investigates.

As a related note on the original contribution of the study, the authors say “To date, despite the enormous burden of distress and potentially traumatic events experienced by people who work in [nursing and care homes], no studies have documented the prevalence of mental health problems in this population group.” It would be useful to know what search strategy the authors used to determine this.

The logic, rationale, and plausibility of the proposed hypotheses

This study does not have any explicit hypotheses. The authors do say “Based on comparable literature on healthcare workers [27], we expect a prevalence of 35%”, but this isn't a hypothesis per se. The authors do not set any inferential criteria in relation to this prediction.

The soundness and feasibility of the methodology and analysis pipeline (including statistical power analysis where applicable)

The study would use a cross-sectional survey design, which seems appropriate to the research question. The authors have also provided some useful detail about the planned sampling plan and materials. I nevertheless have several specific concerns, which I've organised by topic.

Participants and procedures

- The authors plan to use sample weighting, but the power analysis is for a simple unweighted design with a random sample. Would the weighting not affect the sample size necessary to achieve the desired precision of $\pm 5\%$ around the prevalence estimate?
- The authors suggest that based on previous studies they expect a response rate of 70% amongst invited participants. I am not familiar with the local cultural context, but based on my own experience this percentage does sound very optimistic. The authors also seem to require permission from “top management” at the various NCHs that would be sampled, but thus far have only spoken with a couple of these managers. As such the feasibility of the recruitment plan is hard to judge.

Measures

- There seems to be a lack of clarity about what the IES-R is intended to measure – the authors variously refer to “psychological distress” and “post-traumatic stress”. I do think the IES-R is a slightly awkward fit for this study in that it is usually intended to be used with people who have experienced a *specific* traumatic event that happened in a particular distinct (past) time period—e.g., a car accident, an earthquake, etc. In this study, the stated stimulus is “the spread of COVID-19 infections in Italy”, which is an ongoing and somewhat abstract country-wide phenomenon, not a specific event. This then causes problems in terms of the applicability of the items. How much sense does it make, for example, to ask participants whether they “found myself acting or feeling like I was back at that time” or “I tried to remove it from my memory” when COVID-19 infections in Italy *are still happening*?
- The authors say that participants “obtaining a score ≥ 10 at the GAD-7 AND/OR obtaining a score ≥ 26 at the IES-R” will be classified as having “moderate to severe symptoms”. These cutoffs have a crucial role in this study, so a clear rationale would have been useful. The authors just say “These categories are based on values established in the literature”, without citations. What *evidence* is there that scores above these cutoffs indicate “moderate to severe symptoms”?
- More broadly, sumscores from self-report scales have significant limitations as indicators of the presence of mental health conditions. The study aims to estimate “prevalence”, but the authors will be unable to determine the prevalence of any specific mental health disorder or condition (because the scales used are not diagnostic tools). Obviously, every method for measuring psychopathology has limitations, and maybe scores on self-report scales are the best that can be done in this context. But I would have liked to have seen some understanding that alternative methods are available (e.g., diagnostic interviews) and a rationale *for* the selected method.
- It appears that translated Italian versions of the measures will be used. This could be fine, but what evidence is there of the reliability and validity of the translated versions?
- The survey contains a number of detailed questions about participants’ demographic characteristics, education, work, and living situation. I would be concerned that the information obtained might be sufficient to identify many of the participants if shared openly. It might be necessary to have a separate (and more restrictive) data sharing plan for the demographic information, or to reconsider whether all the information is necessary to collect.

Data analysis

- I was pleased to see the plan to apply sample weighting. However, the authors only attempt to weight on a small number of relevant variables (role type, geographical region, gender; not age or income). Furthermore, the authors do not have access to the proportions in each cross-classification cell in the population about which they intend to make inferences, but only a related population (workers from the Italian national healthcare system as a whole, excluding “hospitals, university hospitals and research centres connected with the national healthcare system”). I can appreciate that the unavailability of relevant population data makes this challenging, but it does add substantial uncertainty to the degree to which the results from the sample can be generalised to the *actual* population of interest.
- How is it that the population proportions in Table 2 do not sum to 1?
- The authors say that they will discard incomplete surveys (i.e., listwise deletion). Listwise deletion will cause biased estimates unless the data is missing completely at random, which is rarely a plausible assumption (see Allison, 2001; Schlomer et al., 2010). It would be good to see a more sophisticated plan for dealing with missing data (e.g., multiple imputation), or an argument for why listwise deletion is the most appropriate strategy.
- The authors say that a 95% confidence interval will be reported based on an assumed Poisson distribution. I think they might mean a binomial distribution.
- Regarding inferential criteria (bottom row of Table 1): The authors plan to infer what variety of mental health care is necessary for this sector of workers based on the prevalence of moderate to severe anxiety or distress. For example, if the prevalence is between 10 and 30%, they will infer that “focused psychosocial interventions” are required; if the prevalence is greater than 50% then “psychotherapy” will be required; etc. Put bluntly, this interpretation scheme does not make sense. Mental health *varies* from person to person. Even if the prevalence of moderate to severe anxiety or distress is low, some individuals may have very severe needs, and thus require highly specialised supports. You cannot reasonably infer what services are required by considering only one feature of the distribution of mental health conditions in the population (i.e., the proportion above some cutoff on some numeric scale).

Whether the clarity and degree of methodological detail would be sufficient to replicate exactly the proposed experimental procedures and analysis pipeline

The study is reasonably simple in nature (cross-sectional survey, no experimental manipulation), so replicating it theoretically wouldn't be overly difficult. Nevertheless, the fact that the actual (Italian) materials aren't currently available is one obvious issue.

Whether the authors provide a sufficiently clear and detailed description of the methods to prevent undisclosed flexibility in the experimental procedures or analysis pipeline

The authors have provided plans that are specific enough to shut down some potential avenues for flexibility, but there remain some issues:

- The authors allude to exclusions in a few places, but no comprehensive list of exclusion criteria seems to have been provided. It would be useful to consider issues like detecting careless responses (see Meade & Craig, 2012), whether there are sufficient checks to detect participants outside of the intended population, detecting duplicate responses, etc.

- Relatedly, the authors say that incomplete responses will be discarded, but it's unclear what "counts" as incomplete (Missing an item on the GAD-7 or IES-R? Missing *any* item on the survey?)
- The algorithm/method and software that will be used for the sample weighting is not specified.

A data processing and analysis script could have allowed for a much clearer indication that the analysis pipeline has been adequately specified.

Whether the authors have considered sufficient outcome-neutral conditions (e.g. positive controls) for ensuring that the results obtained are able to test the stated hypotheses

This is not an experiment so some of the considerations that might typically apply here (e.g., positive controls, manipulation checks) are not relevant. The primary concern I have is the lack of a plan to deal with the scenario of slow recruitment – e.g., if the data collection end date is reached and the sample size is less than the target of 350, what would the authors do? Would they conclude that the sample is not sufficient to answer the stated research question?

Miscellanea

- The authors say that "Anxiety is characterized by the excessive worry about everyday events and problems to the point at which the individual experiences considerable distress in performing day to day tasks." This sounds more like generalized anxiety disorder than "anxiety" the emotion or a general description of anxiety disorders.
- The authors say "Last but not least, this survey will represent a first important step for workers to open up about their mental health at work and to become aware that, even in their work environment, there are psychologists who can listen to their difficulties, in person, and provide support accordingly to help them face the remainder of this emergency and beyond." I find it hard to see how a participant filling out an anonymous online survey with fixed-choice items would be likely to receive those benefits.
- Page 4, "A Cochrane systematic review of 33 randomized trials on psychological therapies for individuals living in humanitarian settings..." Maybe this should read "individuals living in countries affected by humanitarian crises"?
- The note stating that "Indeed, most of the workers appear quite oblivious of obvious signs of physical and psychological distress, as they are not only to continue working and but also dealing with an increased workload during this emergency" strikes me as inappropriate to include in the absence of supportive evidence. Do just be aware that the word "oblivious" in English carries a negative connotation (i.e., the sentence could be read as being a little insulting towards the workers).

References

- Allison, P. D. (2001). *Missing data*. Sage Publications.
- Meade, A. W., & Craig, S. B. (2012). Identifying careless responses in survey data. *Psychological Methods, 17*(3), 437–455. <https://doi.org/10.1037/a0028085>
- Schlomer, G. L., Bauman, S., & Card, N. A. (2010). Best practices for missing data management in counseling psychology. *Journal of Counseling Psychology, 57*(1), 1–10. <https://doi.org/10.1037/a0018082>

Appendix B

Thank you very much for giving me the opportunity to review this protocol on the prevalence of psychological distress and anxiety among residential nursing and care home workers during the first COVID-19 outbreak in Italy. This represents a very timely and important research area and the authors should be commended on deciding to collect data on this. The focus on people working within nursing and care homes makes this study of particular importance due to the disproportionate amount of morbidity and mortality experienced in these settings globally, and in Italy in particular. Please find below some comments:

SUMMARY

Pg. 2, line 27: I would possibly add that it is particularly lethal to the elderly, as well as populations residing in long-term stay facilities in general. For example in the recently published UN COVID-19 mental health policy brief you can read: "According to the International Long-Term Care Policy Network approximately half of all COVID-19- related deaths in Australia, Belgium, Canada, France, Ireland, Norway and Singapore occur among residents of long-term care facilities, with mortality rates ranging from 14% to 64%.²⁵ Many of these long-term facilities are homes catering for people with dementia.". You can find the policy brief here: <http://www.quotidianosanita.it/allegati/allegato6947936.pdf>

INTRODUCTION

Not sure whether you want to have one line on the demographic composition of the population in Italy highlighting how Italy has one of the oldest populations in the world?

Pg. 3, line 4. Does the data provide number of deaths for the equivalent period one year ago? I guess that would allow to possibly provide an estimate for the increase in deaths from previous non-COVID-19 years? Not sure if the survey by the ISS was conducted at other time points but if it was it would be interesting to compare.

Pg. 3, line 8: Is there not more precise data on the level of infection and death experienced by those working inside NHCs?

Pg. 3, line 10: It reads a bit strange that the first reason you give for psychological stress is "staff turnover". Obviously this is a factor that warrants consideration but possibly the first reason I would give is higher risk exposure to the virus and possibility of death? (unless there is data showing that fast turnover is ranked as more stressful by the population under study).

Pg. 3, line 19: I would provide brief definitions of what Phase 1 and Phase 2 stand for (or point the reader to your useful Figure 2 from the start) as these terms are used to mean different phases in different countries (see for example Phase 1 in Spain which would probably correspond more to an Italian Phase 2).

Pg. 3. I would possibly also mention the fact that there is anecdotal evidence (but you might be aware of more systematic evidence) of day and occupational group activities being stopped because they became too dangerous for disease spread. This will have inevitably had consequences for the management of patients/residents.

Pg. 4. You might want to mention a few more studies on healthcare workers mental health during COVID-19 in the final manuscript. You might use this very useful living map of COVID-19 literature which has a mental health section (<http://eppi.ioe.ac.uk/cms/Default.aspx?tabid=3765>).

Pg. 4: I don't find the (long) section on MHPSS support particularly relevant at this stage of the manuscript since the key strength of the study is that of looking at prevalence. Obviously in the discussion you could make some cautious suggestions on how prevalence findings might inform certain aspects of interventions. Also removing the MHPSS section would make the introduction more linear and clear.

Pg. 5, line 47: I would possibly remove this section on comparability with Lai et al study as the strength of your study is that of looking at a population that is quite different from that surveyed by Lai. However Lai et al might be a useful starting point for the power analysis below.

METHODS

Pg. 6, line 42: Due to the fast-moving nature of the situation and to the geographical disparities of the epidemic in Italy, have you thought about ways of controlling for:

1. Abrupt changes in phases, e.g. what if in the middle of your data collection the government decided to go back to phase 1 because of a new surge in cases (possibly also only in certain regions)?
2. Regional variation in regulations for nursing homes (e.g. in regions with very low rates of contagion regional governments might allow family visits but not in regions with high rates of contagion).

Pg. 7, line 11: I think it's interesting that you are including also administrative staff but would make sure in the paper to explain why you did so as I assume their experience and stressors will be very different from those of workers providing care to patients?

Pg. 7, line 60: Have the authors thought about possibly using a more general measure of psychological distress rather than the IES that is so inherently tied to traumatic exposure (and possibly more a measure of post-traumatic stress rather than general distress as the authors rightly point out in the methods)? For example the Kessler Psychological Distress scale (K-6) might be a bit more non-specific and has also been used in relation to health professionals during COVID-19 (<https://www.ncbi.nlm.nih.gov/pmc/articles/PMC7225209/>)? It's also shorter than the IES and might increase completion rate as your participants are likely to be extremely busy at this time. If trauma however is a variable of interest why aren't authors using a measure that is more in line with diagnostic conceptualization of PTSD such as the PCL-5 (DSM-5) or the ITQ (ICD-11)?

Pg. 8, line 15: Have the authors thought about possibly using missing data imputation rather than completely removing scales if for example there is only 1 item missing (especially if you use a long scale like the IES)?

Pg. 8, line 27: I would also be a bit more specific in asking for example whether the participant has had come directly in contact and provided care to a person with COVID-19 as I assume that would be an important aspect to capture? Another possibly meaningful question to ask whether they have had colleagues that have fallen sick/died because of COVID-19?

ANALYSIS

Pg. 8, line 43: In the final manuscript please provide references for these cut-offs.

NOTES

Pg. 9, line 39: This was a good idea from the local ethics committee. I would also suggest the authors had a look at this page (<https://www.traumagroup.org/>) where several UK trauma experts have put together advice and info for health workers during the COVID-19. Unfortunately everything is in English except for an Italian version of their guidance to health workers that I helped translate (https://232fe0d6-f8f4-43eb-bc5d-6aa50ee47dc5.filesusr.com/ugd/6b474f_b169ea1bfa5145b3be901f27974eb28b.pdf). Might be helpful for the section on providing practical advice to people.

MEASURES

Pg. 13, line 12: Why doesn't the age question cover more specific age spans after 40? Since older people seem to be more vulnerable to the virus one might assume that older workers might possibly be more anxious/distressed. With the way the question is framed now there might be some meaningful variation between a worker of 60 years old and a worker of 40 that you would be missing?

Pg. 21, line 20: I find this feedback section a bit strange. Firstly I wouldn't necessarily use a color coding as I would find the red unnecessarily alarmistic to the person receiving the feedback. I would mostly try to normalize their feelings and experience since what you are using are not diagnostic tools and mention that, if they want and if they feel it could be helpful, they can reach out to their manager (or to any other person they would like to open up with from their team, family). The way the feedback for people with high scores is currently framed could unnecessarily cause further distress in my opinion. Also I would definitely provide the number of the free hotline for psychological support during the pandemic (<http://www.salute.gov.it/portale/nuovocoronavirus/dettaglioContenutiNuovoCoronavirus.jsp?lingua=italiano&id=5395&area=nuovoCoronavirus&menu=vuoto>), independently from score. There are many useful suggestions in the UK trauma group guidance that I would draw upon.

Overall I commend the authors for choosing to collect data on such an important and understudied population that has been disproportionately affected by COVID-19 and whose mental health needs should definitely receive more attention. The research question is of scientific validity. The proposed hypothesis is plausible and the methodology is sound, feasible, and clearly explained in detail. All the best with data collection and I look forward to reading the final manuscript.

Kind regards and take care in these times

Alessandro Massazza

Alessandro Massazza

Ph.D student, ESRC DTP

Department of Clinical, Educational and Health Psychology

University College London

26 Bedford Way, Bloomsbury

London, WC1H

[*alessandro.massazza.13@ucl.ac.uk*](mailto:alessandro.massazza.13@ucl.ac.uk)

[*https://www.ucl.ac.uk/epicentre/people/students/alessandro_massazza*](https://www.ucl.ac.uk/epicentre/people/students/alessandro_massazza)

[*@alemassazza*](https://www.ucl.ac.uk/epicentre/people/students/alessandro_massazza)

Appendix C

REVIEWER 1

Major concerns:

R1#1. In section 3.1, the authors describe the estimated sample size needed to detect an estimated proportion of 35% of NCH employees having moderate to severe self-reported symptoms of anxiety and/or distress. They based this expectation on a prevalence survey of anxiety/distress as reported by healthcare workers exposed to COVID-19 in Chinese hospitals. Given the extent of the COVID-19 epidemic in Italy, this may well be reasonable. However, the formula used to estimate the required sample size with a 5 percentage point margin of error ($\pm 5\%$) doesn't seem to recognize that the authors will be relying on a single-stage cluster design. In these types of designs, observations within the same cluster are not independent, but typically correlated in some way. Thus, the expected sample size should account for this intracluster correlation. If it does not, the expected margin of error may well be greater than ± 5 percentage points. (Indeed, a general rule of thumb is that simple random samples will require about 400 respondents to estimate a proportion of 0.5 with a margin of error around 5 percentage points). Thus, the expected sample size of 350 may well be too low to obtain a margin of error of 5 percentage points in this particular survey that seems to employ a single stage cluster design. The authors should consider what the expected intracluster correlation might be and how that might influence estimated variances of estimated proportions when they conduct their statistical analysis. Put another way, while 350 employees might be surveyed, as a result of the clustering of employees within NCH and the similarity of employees within a given NCH means that the "effective" sample size is < 350 (for example, if, in a given NCH, personal protective equipment (PPE) is readily available, this would mean that all employees having direct contact with patients have access to the required PPE, meaning these employees are more alike than two randomly selected employees from two randomly selected NCH).

Reply to R1#1. We thank the Reviewer for raising this issue and for giving us some guidance on how to address it. In the revised manuscript we have now fleshed out our assumptions and calculations (please see section 3.1.1). On this basis, we are now aiming for a minimum sample size of 900. Considering the urgency of the matter and the available resources, our recruitment effort will be close to our maximum capacity. We hope that this adjustment will look acceptable to the Reviewer.

R1#2. Table 1 mentions that a mixture of convenience, snowball, and random sampling methods will be used. However, the methods section implies that probabilistic sampling methods only will be used within three geographic strata. Why does this discrepancy exist? The authors need to consistently describe the sampling methods in both the table and the methods section. Indeed, if the authors only plan to use a single-stage cluster sampling design, then this should be stated as such in Table 1 as well as in the methods section.

Reply to R1#2. Thank you for noticing this inconsistency. We are planning to use a single-stage cluster sampling design and we have now amended Table 1 accordingly.

R1#3. On page 6, lines 59-60, the authors state they will sample a minimum of 32 NCH from an existing frame of NCH in Italy; 23 of these NCH will be sampled from the North (given 72% of all NCH in Italy are from the Northern region), 6 NCH from the Centre and 3 from the South of Italy. While it's somewhat implied, it would better to explicitly state that these 32 NCH will be randomly selected within each geographic stratum. If NCH are not randomly selected, might the possibility of some type of "selection" bias exist?

Reply to R1#3. We are now explicitly stating that we will randomly select NCH. We are also now focusing all our efforts on the most affected region of Italy (i.e. the North).

R1#4. The authors expect to collect data from 37 employees per nursing home, based on an expected response rate of 70% (as reported in the Chinese study by Lai et al). It's not clear whether

an expected 70% response rate is reasonable for Italy, given the vastly different cultures of the two countries. In addition, the Lai et al. study was conducted among healthcare workers, while the planned study is among NCH workers, which may be a more difficult population to survey during the COVID-19 epidemic in Italy. This ultimately could affect the final sample size obtained influencing the statistical reliability of the estimated prevalence of moderate to severe anxiety/distress among NCH throughout Italy. Again, it might be necessary to reconsider the expected sample size for this survey of NCH employees.

Reply to R1#4. Although very high response rates have been previously reported among Italian healthcare workers (e.g. 87% in Alboino et al., 2010), and our survey includes a response facilitator in the form of individual feedback, we have now revised our sampling plan to account for an expected response rate within single NCH of 48%, which is below the average response rate for surveys in the healthcare sector (Baruch and Holtom, 2008). We will now contact a minimum of 120 NCH which, with an expected response rate of 30%, could give us access to 36 NCH. Given an average number of workers of 53 and an expected 48% response rate within NCH, we would reach the minimum target sample size of 900 respondents.

R1#5. The main measures the authors will use to estimate the prevalence of self-reported moderate to severe anxiety/distress are the Generalized Anxiety Disorder (GAD-7) scale and the Impact of Event-Scale (revised: IES-R). The authors state both of these scales are widely used in the existing literature. However, it would be beneficial to report additional information about these scales, such as their reliability and construct validity.

Reply to R1#5. We have now added information on the validation and psychometric properties of the GAD-7 and IES-R in the materials section, together with appropriate references (please see Section 3.2.2).

R1#6. In addition to these two scales, respondents will also be asked about their working conditions in the previous two weeks. However, the questions used to assess working conditions (as outlined in the appendix) seem to lack important detail. For example, the amount of anxiety reported by an NCH employee might well be related to the total amount worked in the previous two weeks, including the number of days worked in the previous two weeks as well as the usual length of a typical shift worked in the previous two weeks. Both of these factors could influence levels of anxiety and it would seem to be beneficial to include this additional information. In addition, given the GAD-7 scale asks about anxiety in the last 7 days and the IES-R scale asks about distress in the past two weeks, it would also seem beneficial to ask respondents how many days (out of 7) they work in the past week and how many days (out of 7) they worked in the week prior to the last week.

Reply to R1#6. We thank the reviewer for the suggestion to quantify more precisely our respondents' working conditions. Incidentally, we suspect that anxiety and distress for workers in NCH are not exclusively linked to being physically working in their centre at this time or to work a great number of hours - even people working from home in quarantine could suffer from severe distress for various reasons: the fear of infecting family members being COVID-19 positive, the feeling of isolation and frustration for not being allowed to go to work, the difficulty in communicating with residents and their families. However, we agree that information about working commitments would be very relevant, in particular the number of days worked over the last week and the week prior to the last (regardless of the total number of hours worked, these require travel to and from home and expose the individual to a heightened risk of contagion). Because we have to be mindful not to try the patience of our respondents, in the revised version of the survey we have eliminated our previous question 8 and replaced it with two questions about the number of days they worked in the last week and the number of days they worked in the seven days prior to the last week. This clearer emphasis on the working routine will also plausibly have the useful side-effect of attuning questionnaire responses onto work-related stressful events.

R1#7. Another question that might be beneficial to ask is whether employees of NCH have access to personal protective equipment (PPE). If so, this could also influence the prevalence of self-reported anxiety/distress. For example, it could well be the case that employees lacking access to PPE experience higher levels of anxiety/distress than employees who have access to PPE. If the data have not yet been collected, it would seem a shame to miss out on an opportunity to ask such a question, since it is potentially related to self-reported levels of anxiety/distress.

Reply to R1#7. We fully agree with this suggestion and we have happily included a further question on availability of PPE equipment (n. 9). Although all NCH - especially in the North of Italy – should now be appropriately equipped with PPE (in line with government regulations), this cannot be given for granted, especially considering the market shortages of PPE that have fueled a lot of tension and debates during the past few months.

R1#8. In the statistical analysis, I wonder whether the objective of this study is to estimate the prevalence of self-reported anxiety/distress overall (among all NCH workers) or by type of work and/or by geographic region and type of worker. The latter seems important information to obtain, as administrative staff not having direct contact with patients might potential face less risk of contracting SARS-CoV-2 than employees having direct contact with NCH residents. This would then influence overall prevalence (especially if there were an imbalance in the proportion of employees sampled, or, put another way, if the sample is not representative of the types of employees across all NCH in Italy). It's also important to stratify by region, as the North region of Italy was the hardest hit by the COVID-19 epidemic in Italy.

Reply to R1#8. The main aim of this study is now to reach an overall prevalence estimate at the level of the most affected region, i.e. the North. In the absence of any other information on the current state of workers in NCH, we believe this already represents an important approach, both at the organizational and at the societal level, to the population of workers in NCH.

There are good reasons (p2 lines 22-42, p3 lines 1-21) to believe that there is not a neat differentiation in the level of stress experienced by different categories of workers in NCH, all of whom work either in close physical contact with the residents (especially technical, healthcare and part of the administrative staff, like nursing directors and administrators) or can form close bonds with them (starting from cleaners, all the way up to the general manager; <https://www.msf.org/why-protecting-staff-care-homes-during-covid-19-so-vital>). The shared educational background of administrators, their management philosophy and their attitudes toward person-centered care represent significant aspects in the adoption of unit care in long-term care settings (Lee & Lee 2012). For those working in NCH, it is very clear that healthcare staff are not the only ones who have been hugely affected from the COVID-19 situation in their work environment. All staff are all part of a close-knit community sharing high level distress. We do agree with the Reviewer that it would be interesting to expand the main scope of this study and aim for prevalence estimates also by worker. However, to do this while achieving acceptable peer-reviewed pre-registration standards, we would have to commit to a target sample size that we simply cannot deliver with the urgency that is required by the situation. We will do our best to achieve and surpass the target sample size, also to enable more fine-grained analyses of the database in future work.

R1#9. The authors state that they will weight their sample to match population proportions of workers from the Italian national healthcare system. The authors will exclude from this database all workers from hospital, university hospitals, and research centres. While this is reasonable (and I can appreciate that the authors are attempt to produce weighted estimates), as a limitation, it would be necessary to consider and potential bias introduced by weighting to a population that could potentially be different than the subset of NCH workers.

Reply to R1#9. We have now considered how our weighting could affect the variance of our prevalence estimate and this is discussed in the revised paper. We are matching to a population based on the most accurate data we can find. We will present our results with and without

weighting. Once the survey is complete, our data will be made public for any re-analysis and comparison to any new population datasets that are produced.

R1#10. For the analysis, the authors state that incomplete surveys will be discarded. However, it's not clear what the criteria will be for excluding incomplete surveys. Do the authors mean that all questionnaire items must be complete to consider a survey "complete"? Or do they mean that respondents failing to answer key questions will be considered incomplete (e.g., employee type, hours worked)? Also, given the GAD-7 and IES-R scales employ a larger number of questions, will complete data for these scales be required to be included for analysis? For example the IES-R scale contains 22 items. It's plausible that some respondents may not answer all items, and so useful data may be discarded. (1) The authors need to clearly define what constitutes an "incomplete" survey and (2) whether it might be possible to retain as much data as possible, possibly using multiple imputation methods. This would seem to be especially important for key measures of anxiety/distress (although if the number of exclusions is low, then this is probably less necessary). In addition, having a larger sample size could potentially mean more data from "complete" surveys for statistical analysis.

Reply to R1#10. We have now defined in the text what constitutes an "incomplete" survey (i.e. a survey in which the demographics, the GAD-7 and the IES-R have not been fully completed) and included checks to detect careless responding. Since this would be a first comprehensive study on NCH, we would like to retain full and possibly accurate information rather than partial information. The proposed sample size has now been increased from 350 to 900 complete surveys but we will still have the chance to consider incomplete ones in further exploratory analyses. Since filling all items on a page is necessary to be able to proceed in the survey, we expect no missing items from respondents who are committed and focused enough to reach completion of the first three sections (demographics, GAD-7 and IES-R).

R1#11. In the statistical analysis, the authors do not seem to recognize that their data are collected from a cluster sampling design and seem to ignore the need to account for the clustering of respondents within NCH. Doing so will produce estimates of variance that are too small and conservative (i.e., narrower) confidence intervals. The clustering of respondents within sampled NCH needs to be recognized in the statistical analysis of these data.

Reply to R1#11. We fully acknowledge that the clustering aspect of our survey was not clear from the first submission of our paper. This has now been rectified in the paper, where we discuss how the variance depends on the design effect and the intracluster correlation. Our clusters will be based on the NCH. When collecting the data, individual NCH will be randomly assigned a cluster identifier, from which it will not be possible to infer the identity of a care home in the dataset that we will release.

R1#12. The authors assume that individual scores (on the GAD-7 and IES-R scales?) will be "added up and presented as a percentage with a 95% confidence interval" assuming a Poisson distribution. Why is a Poisson distribution assumed rather than assuming a binomial distribution? The Poisson distribution assumes the mean equals the variance, but have the authors considered whether estimated means might exhibit over- or under-dispersion (such that the mean does not equal the variance)? If so, might a negative binomial assumption be more relevant, or at least used for a sensitivity analysis (of estimated confidence intervals?) More detail should be added to address these issues.

Reply to R1#12. We had chosen the Poisson distribution as a candidate to represent discrete data but on second thought we agree that a binomial distribution would be a more standard and suitable choice (also given the lack of previous data from this population). We are now using standard estimation procedures in STATA, which assume a binomial distribution.

Minor comment:

R1#13. 1. The introduction seems unnecessarily long and could be tightened up to discuss the most important points and to clearly identify the aims of the study more quickly.

Reply to R1#13. We have now made this alteration.

References

Alboino S, Tartaglia R, Bellandi T, Amicosante AMV, Bianchini E, Biggeri A (2010). Patient safety and incident reporting: survey of Italian healthcare workers. *Qual Saf Health Care* **19**:i8-i12.

Baruch Y and Holtom BC. (2008). Survey response rate levels and trends in organizational research. *Human Relations* 61(8), 1139-1160. doi.org/10.1177/0018726708094863

Lee GH, & Lee JM (2012). Impact of facility managers' characteristics on their intention to introduce a unit care system. *Health and Social Welfare Review*, **32**(4), 94–122.

REVIEWER 2

R2#1. Pg. 2, line 27: I would possibly add that it is particularly lethal to the elderly, as well as populations residing in long-term stay facilities in general. For example in the recently published UN COVID-19 mental health policy brief you can read: “According to the International Long-Term Care Policy Network approximately half of all COVID-19- related deaths in Australia, Belgium, Canada, France, Ireland, Norway and Singapore occur among residents of long-term care facilities, with mortality rates ranging from 14% to 64%. Many of these long-term facilities are homes catering for people with dementia.”. You can find the policy brief here: <http://www.quotidianosanita.it/allegati/allegato6947936.pdf>

Reply to R2#1. We thank the reviewer for this comment. We have now added the suggested sentence in the Summary.

R2#2. Not sure whether you want to have one line on the demographic composition of the population in Italy highlighting how Italy has one of the oldest populations in the world?

Reply to R2#2. We have now added the following sentence in the Introduction: “According to recent statistics of the Population Reference Bureau, Italy has the second largest percentage of older adults (65+) in the world. In 2020, 23.1% of the total population in Italy was estimated to be aged 65 years and older, with elderly people in the Italian society growing constantly”.

R2#3. Pg. 3, line 4. Does the data provide number of deaths for the equivalent period one year ago? I guess that would allow to possibly provide an estimate for the increase in deaths from previous non-COVID-19 years? Not sure if the survey by the ISS was conducted at other time points but if it was it would be interesting to compare.

Reply to R2#3. As far as we know, the same survey has not been conducted at other points in time and we could not find reliable and/or representative comparisons with previous years for NCH specifically. These data would be very interesting indeed. For the Trento area, for example, it appears that, during the month of March, mortality figures in nursing homes show a 100% increase compared to 2019: <https://www.ildolomiti.it/cronaca/2020/nelle-rsa-circa-140-morti-in-piu-rispetto-allanno-scorso-di-questi-solo-67-con-covid-19-e-gli-altri-nava-lipotesi-che-il-coronavirus-sia-responsabile-e-altamente-accreditata>

R2#4. Pg. 3, line 8: Is there not more precise data on the level of infection and death experienced by those working inside NHCs?

Reply to R2#4. Unfortunately, we could not find more details.

R2#5. Pg. 3, line 10: It reads a bit strange that the first reason you give for psychological stress is “staff turnover”. Obviously this is a factor that warrants consideration but possibly the first reason I would give is higher risk exposure to the virus and possibility of death? (unless there is data showing that fast turnover is ranked as more stressful by the population under study).

Reply to R2#5. Thank you. We have re-organized the text according to the Reviewer’s suggestion.

R2#6. Pg. 3, line 19: I would provide brief definitions of what Phase 1 and Phase 2 stand for (or point the reader to your useful Figure 2 from the start) as these terms are used to mean different phases in different countries (see for example Phase 1 in Spain which would probably correspond more to an Italian Phase 2).

Reply to R2#6. We have now added a short explanation as suggested by the Reviewer (p. 2 line 30-33)

R2#7. Pg. 3. I would possibly also mention the fact that there is anecdotal evidence (but you might be aware of more systematic evidence) of day and occupational group activities being stopped because they became too dangerous for disease spread. This will have inevitably had consequences for the management of patients/residents.

Reply to R2#7. We thank the Reviewer for this valuable suggestion. We have now added the following sentence to our introduction (p2 lines 37-42): “Third, the Center for Disease Control and Prevention (CDC) guidelines for preventing the spread of COVID-19 in retirement communities [9] recommended cancelling group activities. However, the elimination of social interactions may increase the risk of adverse mental health outcomes during a stressful event such as a disease outbreak. Therefore, for the group activities that are deemed as absolutely essential, CDC recommended introducing social distancing and limiting the number of attendees. This has had inevitable consequences for the management of residents.”

R2#8. Pg. 4. You might want to mention a few more studies on healthcare workers mental health during COVID-19 in the final manuscript. You might use this very useful living map of COVID-19 literature which has a mental health section (<http://eppi.ioe.ac.uk/cms/Default.aspx?tabid=3765>).

Reply to R2#8. We thank the Reviewer for providing this useful link.

R2#9. Pg. 4: I don’t find the (long) section on MHPSS support particularly relevant at this stage of the manuscript since the key strength of the study is that of looking at prevalence. Obviously in the discussion you could make some cautious suggestions on how prevalence findings might inform certain aspects of interventions. Also removing the MHPSS section would make the introduction more linear and clear.

Reply to R2#9. We have now deleted the section focused on MHPSS interventions as suggested by the Reviewer. We plan to add considerations on evidence-based MHPSS interventions in the discussion section of the full manuscript.

R2#10. Pg. 5, line 47: I would possibly remove this section on comparability with Lai et al study as the strength of your study is that of looking at a population that is quite different from that surveyed by Lai. However, Lai et al might be a useful starting point for the power analysis below.

Reply to R2#10. We have taken on board the Reviewer’s suggestion and removed the section on comparability with Lai et al.’s (2020) study from the Introduction. We are mentioning it as a starting point in the power analysis section, and referring the reader to their Table II for further details.

R2#11. Pg. 6, line 42: Due to the fast-moving nature of the situation and to the geographical disparities of the epidemic in Italy, have you thought about ways of controlling for:

1. Abrupt changes in phases, e.g. what if in the middle of your data collection the government decided to go back to phase 1 because of a new surge in cases (possibly also only in certain regions)?
2. Regional variation in regulations for nursing homes (e.g. in regions with very low rates of contagion regional governments might allow family visits but not in regions with high rates of contagion).

Reply to R2#11. Given the current trend in the data on contagions and the government’s cautious attitude, it seems unlikely that a new surge in cases of a size sufficient to return to phase 1 at the national or broad regional level will happen within the timeframe of our survey. We believe it is unlikely that national government regulations will be relaxed too soon (i.e. within the expected timeframe of this survey, which should really start as soon as possible) given that several asymptomatic individuals may be found in the general population and that every single new case in

NCH could have very dramatic effects. However, it is also true that in areas where the strain has not been felt as much as in the Northern region a more lenient approach to visits in NCH may be gradually adopted by local governors. Therefore, we have now decided to concentrate our efforts purely on the Northern region of Italy which, for our purposes (i.e. to size the psychological burden under the COVID-19 threat) is also the most interesting.

R2#12. Pg. 7, line 11: I think it's interesting that you are including also administrative staff but would make sure in the paper to explain why you did so as I assume their experience and stressors will be very different from those of workers providing care to patients?

Reply to R2#12. Thank you. We have now expanded in our introduction a paragraph to clarify this issue in accordance with the Reviewer's comment and a similar comment by Reviewer 1 (please see pages 2 and 3).

R2#13. Pg. 7, line 60: Have the authors thought about possibly using a more general measure of psychological distress rather than the IES that is so inherently tied to traumatic exposure (and possibly more a measure of post-traumatic stress rather than general distress as the authors rightly point out in the methods)? For example, the Kessler Psychological Distress scale (K-6) might be a bit more non-specific and has also been used in relation to health professionals during COVID-19 (<https://www.ncbi.nlm.nih.gov/pmc/articles/PMC7225209/>)? It's also shorter than the IES and might increase completion rate as your participants are likely to be extremely busy at this time. If trauma however is a variable of interest why aren't authors using a measure that is more in line with diagnostic conceptualization of PTSD such as the PCL-5 (DSM-5) or the ITQ (ICD-11)?

Reply to R2#13. We thank the reviewer for this comment. We would like to administer the IES instrument for the following reasons: 1) as the reviewer rightly pointed out, IES is mostly inherent to exposure to traumatic events. According to preliminary clinical work with NCH staff during the COVID-19 pandemic, staff members were exposed to multiple traumatic events, for example seeing elderly people dying; forced and long periods of isolation (without the possibility of interacting with close family members); 2) there is an Italian validated version of the IES that would allow a feasible and smoothly administration process; 3) K6 is more related to general distress, and we will consider this instrument in the near future, for example in case of follow-up studies. Finally, the use of the planned instruments - that are broadly administered in scientific studies - will allow the collection of harmonized and comparable outcome data with an additional value for readers.

We hope that the reviewer is happy with this justification.

R2#14. Pg. 8, line 15: Have the authors thought about possibly using missing data imputation rather than completely removing scales if for example there is only 1 item missing (especially if you use a long scale like the IES)?

Reply to R2#14. The missing-item case does not apply here, as a response will be required to all items, in order to proceed to the next page.

R2#15. Pg. 8, line 27: I would also be a bit more specific in asking for example whether the participant has had come directly in contact and provided care to a person with COVID-19 as I assume that would be an important aspect to capture? Another possibly meaningful question to ask whether they have had colleagues that have fallen sick/died because of COVID-19?

Reply to R2#15. These are important distinctions to make between respondents. Since we are now focusing on the North of Italy, though, there may be more homogeneity within the population. We feel that the distinction between those who have been in contact with COVID-19 positive colleagues/patients and those who have not is particularly relevant, thus we have now added a question (please see question #8) to address this aspect.

R2#16. Pg. 8, line 43: In the final manuscript please provide references for these cut-offs.

Reply to R2#16. Thank you for the comment. We have now provided those references in the manuscript (please see Sections 3.2.2. & 3.3.1).

R2#17. Pg. 9, line 39: This was a good idea from the local ethics committee. I would also suggest the authors had a look at this page (<https://www.traumagroup.org/>) where several UK trauma experts have put together advice and info for health workers during the COVID-19.

Unfortunately everything is in English except for an Italian version of their guidance to health workers that I helped translate (https://232fe0d6-f8f4-43eb-bc5d-6aa50ee47dc5.filesusr.com/ugd/6b474f_b169ea1bfa5145b3be901f27974eb28b.pdf). Might be helpful for the section on providing practical advice to people.

Reply to R2#17. We are grateful to the Reviewer for this suggestion and for pointing to this very helpful website. We added the practical advice to our survey (please see the revised feedback section) plus the free psychological support phone number.

R2#18. Pg. 13, line 12: Why doesn't the age question cover more specific age spans after 40? Since older people seem to be more vulnerable to the virus one might assume that older workers might possibly be more anxious/distressed. With the way the question is framed now there might be some meaningful variation between a worker of 60 years old and a worker of 40 that you would be missing?

Reply to R2#18. Thank you. We have added three age categories (41-50; 51-60; 61 and over) in the revised version of the survey.

R2#19. Pg. 21, line 20: I find this feedback section a bit strange. Firstly, I wouldn't necessarily use a color coding as I would find the red unnecessarily alarmistic to the person receiving the feedback. I would mostly try to normalize their feelings and experience since what you are using are not diagnostic tools and mention that, if they want and if they feel it could be helpful, they can reach out to their manager (or to any other person they would like to open up with from their team, family). The way the feedback for people with high scores is currently framed could unnecessarily cause further distress in my opinion. Also I would definitely provide the number of the free hotline for psychological support during the pandemic

(<http://www.salute.gov.it/portale/nuovocoronavirus/dettaglioContenutiNuovoCoronavirus.jsp?lingua=italiano&id=5395&area=nuovoCoronavirus&menu=vuoto>), independently from score. There are many useful suggestions in the UK trauma group guidance that I would draw upon.

Reply to R2#19. We thank the Reviewer for these useful suggestions. We agree that the red colour can be misinterpreted: that category has now been eliminated. We are also providing the number of the free hotline for psychological support, rather than suggesting to get in touch directly with their manager (as this may not represent the best option for all). In addition, some very useful advice taken from traumagroup.org website has been included in the feedback page of the revised survey.

Overall I commend the authors for choosing to collect data on such an important and understudied population that has been disproportionately affected by COVID-19 and whose mental health needs should definitely receive more attention. The research question is of scientific validity. The proposed hypothesis is plausible and the methodology is sound, feasible, and clearly explained in detail. All the best with data collection and I look forward to reading the final manuscript.

Kind regards and take care in these times

Alessandro Massazza

*Thank you for your helpful comments Alessandro.
Best wishes, The Authors*

References

Lai J, Ma S, Wang Y, et al. (2020). Factors Associated With Mental Health Outcomes Among Health Care Workers Exposed to Coronavirus Disease 2019. *JAMA Netw Open* **3**(3), e203976

REVIEWER 3

R3#1. Use of the IES-R: In the instructions you state that the spread of COVID-19 infections in Italy is the stressful event. I wonder a) if this is too vague and participants need to state or at least think about a specific event they have found stressful eg. death of a resident, lack of PPE equipment b) similarly, whether you want to focus this on stress related to working in a nursing home, rather than the spread of COVID-19 in general?

Reply to R3#1. Thank you to the Reviewer for this comment, which we found very useful for improving clarity in our manuscript and focus in our survey. The survey has now been edited to include situation-specific questions (e.g. contact with COVID-19 positive patient/colleague and availability of PPE at work) setting up the work environment as a frame of reference. Also, we are now referring to “any stressful events in your nursing/care home due to the recent COVID-19 pandemic” rather than to “the spread of COVID-19 infections in Italy” in the IES-R instructions.

R3#2. Regarding the Interpretation section and recommendations for intervention in Table 1: You seem to suggest that the type of intervention required will depend solely on the prevalence of anxiety and psychological distress symptoms eg. a prevalence of 30-50% would require first and second layers of intervention (psychotherapy and focused psychosocial interventions) and prevalence of 50%+ would need interventions from the first layer (psychotherapy). I have two comments about this. Firstly, does the type of intervention required not depend at least in part on the severity of the symptoms and functional impairment, not just the prevalence? For example, as in an evidence based stepped care model <https://www.nice.org.uk/guidance/cg113/chapter/Key-priorities-for-implementation>. Secondly, the more prevalent, surely the more difficult it is to deliver first layer interventions like psychotherapy for everyone, and interventions from lower levels which can be rolled out more widely should be considered?

Reply to R3#2. We edited the terminology in Table 1, in order to make it clearer that we are relying on prevalence to obtain useful initial indications and that the severity of an individual’s symptoms and functional impairment should be investigated at the next stage. We also appreciate the consideration on the feasibility of delivering complex interventions (as psychotherapy) to broad population groups. In this regard, we added an explanation. We will be able to expand on this in the discussion of the full manuscript.

R3#3. For the GAD-7, you have included the additional 8th question. As you are making decisions based on cut-off scores using the 7 questions, how will this 8th question be used in the analysis and decision making?

Reply to R3#3. Thank you for the comment. The 8th question was not meant to be used in the analysis and decision making but to be considered qualitatively outside the scoring. However, its interpretation would not be straightforward and since we will not have the possibility to talk directly to all respondents and the length of the survey has grown, the 8th question has been now eliminated.

R3#4. You have included a quality of life measure in the appendix but I did not see mention of that in the methods and it doesn't seem to fit your research questions. Is this necessary?

Reply to R3#4. This was explained in a note referred to by a small superscript number, which could have easily gone unnoticed. The inclusion of this measure has been requested by the ethics committee as way to give some potentially useful indication (based on a construct that does not have as strong a clinical connotation unlike anxiety and post-traumatic stress disorder) to respondents. As correctly noticed by the Reviewer, this is not really part of our study and it has been included in the Appendix purely for transparency reasons.

R3#5. Would it be helpful to include a question about access to PPE in your demographics questionnaire?

Reply to R3#5. Thank you for the useful suggestion. We added this question in the revised version of the survey (question 9).

R3#6. Is there any possibility of having a qualitative element to the study? I think this would really help to understand the specifics of what might be causing distress in nursing home workers and inform development of suitable interventions.

Reply to R3#6. We thank the reviewer for this very insightful suggestion. The survey did already offer the possibility to leave comments on page 5, however we are now also suggesting participants to mention something that might have particularly worried them during this period, if so they wished.

R3#7. Feedback to participants on their scores: I don't fully understand where the traffic light percentages have come from - are these based on the outcome measure scores? For amber, when you say seek 'our' advice, does our refer to the research team? But for red the participant should speak to their centre manager? Will the centre managers be trained in how to respond, how to manage any distress? I think it might be helpful to signpost participants to organisations where they can get support for their mental health should they wish to.

Reply to R3#7. The orange percentage in the traffic-light corresponds to the cut off of >5 ($42/100*12$) of the General Health Questionnaires (GHQ-12) originally designed to identify psychological distress in primary care settings and among general medical outpatients. This cut-off score was taken from previous literature (Bratas et al. 2014). However, the red colour, did not correspond to a cut-off reported in literature and we have now deleted it. NCH usually lack a formal support group and psychological difficulties are generally managed autonomously by each centre. We fully agree with the Reviewer regarding the advice/contacts and are now providing the number of the free hotline for psychological support, rather than suggesting to get in touch with us or directly with their manager (as this may not represent the best option for all). In addition, some very useful advice taken from traumagroup.org website has been included in the feedback page of the revised survey.

R3#8. As the questionnaire titles are not the 'real' titles, this could be a little misleading - could you leave them out?

Reply to R3#8. Yes, thank you for this suggestion. They have now been eliminated.

R3#9. The introduction is perhaps longer than needed and I feel could be condensed to make it punchier. In particular from page 3 line 10 (all staff...) to line 24. Lines 25-30 are better and sum up the problems succinctly.

Reply to R3#9. Thank you. We shortened/re-organized the introduction according to your comments and suggestions from other Reviewers.

R3#10. Again, I think the introduction could be condensed from pg 5 line 1 to 17.

Reply to R3#10. Thank you. We shortened/re-organized the introduction according to your comments and suggestions from other reviewers.

R3#11. Pg 3 line 12: turnover - is this turnover of staff? Perhaps make this clearer

Reply to R3#11. It is, thank you. We have now clarified this.

R3#12. Pg 3 line 19: You mention phase 1 and 2 here without referring to figure 2. Perhaps refer to it here for clarity, or not mention the phases as this point (too much detail?).

Reply to R3#12. Thank you, done. We also added a very brief clarification, as also suggested by Reviewer 1.

R3#13. As you refer to the IASC pyramid of support throughout the manuscript, it would be helpful to have a figure showing the pyramid with layers of support. I think it would be more helpful than figure 1 which shows the mortality rates by age, as this doesn't add much more to the text you have written about this.

Reply to R3#13. Thank you for this suggestion. We would like to retain Figure 1 as it shows in an intuitive and immediate way the size of the problem for the elderly population in Italy. We have now added the IASC pyramid as Figure 3 in our manuscript (mhGAP Humanitarian Intervention Guide, WHO).

R3#14. Typos/missing words

1. Pg 2 line 27: suggest adding 'just a' so it reads 'in just a few months'
2. Pg 4 line 23: I would replace 'at' with 'on' when referring to scores on a scale
3. Pg 6 line 23: In table, 'q' is missing from required
4. Pg 6 line 42: Can the dates be added now where there are ??

Reply to R3#14. Thank you, we have now fixed the first three points. We will be able to add the dates just before IPA.

References

Bratas O, Gronning Q and Forbord T. A Comparison of two versions of the General Health Questionnaire Applied in a COPD Population. *Health Care Current Reviews*. 2014;**2**:2

REVIEWER 4

Comments to the Author(s)

The proposed research aims to address an important research question regarding the prevalence of mental health conditions in residential nursing and care home workers during the COVID-19 pandemic. However, several aspects of the design of this study are unclear.

R4#1. The authors have chosen to focus on anxiety and distress in this sample, but do not offer a rationale for why these symptoms have been chosen over other symptoms experienced widely in the general population (such as depression). Additionally, more consideration is needed as to whether it is appropriate to combine these separate symptoms into a single composite prevalence rate. Previous studies have reported higher prevalence rates for distress than anxiety (e.g. Lai et al., 2020).

Reply to R4#1. We appreciate this comment from the Reviewer. The choice of focusing on anxiety and distress was motivated by the prevalence rate of anxiety and PTSD symptoms in population groups exposed to potentially traumatic events. Moreover, there is a urgent need to recognize and treat psychological symptoms of anxiety and PTSD to prevent them from becoming formal psychiatric diagnoses (with risk of chronic suffering and functional impairment). According to available data from populations exposed to potentially traumatic events, PTSD may generate medical retirement from work, changes in work volition and coping strategies (Harris, 2019), and difficulties in interacting with colleagues. (Elmsley et al. 2003). In addition, it has been recently stressed that post-trauma social support and stressors experienced during recovery are the risk factors most strongly predictive of longer-term mental health status (Greenberg et al, 2020). Finally, several mental health conditions are strictly related to anxiety and PTSD, for example depression (with comorbidity rates that may reach 90%; Hunot et al., 2007), also PTSD symptom severity account for the higher levels of depression in ICD- 11 PTSD (Barbano et al., 2019). We consider the Reviewer's comment to be very appropriate but also for ethical issues already discussed with the committee, we decided to make a choice on the scales in order to not to lengthen the already long enough interview which would add fatigue and stress to workers.

R4#2. Finally, all staff members within care homes are invited to participate in this survey. Due to the large variation in job roles it is possible that mental health outcomes would differ according to occupation. Currently, the authors propose to treat these individuals as a homogenous group, but I feel that further justification is required for this approach.

Reply to R4#2. We understand and share the Reviewer's point and we are now addressing it in the manuscript (please see the Introduction, pages 2 and 3), also in response to Reviewers 1 and 3.

The authors hypothesise that they will observe a prevalence rate of 35% for severe to moderate self-reported mental health symptoms of anxiety and/or distress. There are several issues with the hypothesis in its present state:

R4#4. The hypothesis is not operationalised. The authors previously discuss prevalence of severe to moderate self-reported mental health symptoms of anxiety and/or distress, however they do not state which measures or cut-off scores will be used to define this (although this is outlined in later sections).

Reply to R4#4. We have now provided operationalization details (measures and cut-off scores) for our hypothesis (please see Section 2.1).

R4#5. It is also unclear whether the hypothesised prevalence rate of 35% relates to the unweighted or weighted prevalence rate analyses later outlined in the pre-registered data analysis.

Reply to R4#5. The prevalence rate should not differ substantially in the two cases, as in these complex designs (i.e. non-SRS), what changes due to design effects is mainly the standard error of the estimate and thus the size of the 95% confidence interval around the estimate (e.g. Heeringa et al., 2017). We will present weighted and non-weighted estimates in our results.

R4#6. Most problematically, it is unclear whether the authors hypothesis of a 35% prevalence rates originated from. In the paper cited (Lai et al., 2020), prevalence rates are reported as follows: depression (50.4%), anxiety (44.6%), insomnia (34%), and distress (71.5%). The confusion may have originated from Lai et al (2020) discussing a 35% prevalence rate for psychological comorbidities within their sample size calculation. However, this estimate refers to a paper looking at survivors of SARS (Lee et al., 2007). In this paper prevalence rates were 36.3% for depressive symptoms and 36.7%, for anxiety.

Reply to R4#6. The 35% prevalence rate actually comes from Table 2 in Lai et al.'s (2020) paper (we are now also mentioning this in the paper, please see Section 3.1.1), showing that an unweighted average of 35% frontline and second-line workers report moderate-to-severe symptoms on the IES-R. Because the composition of these groups is unequal (there are about 1.5 more second-line than frontline workers), the figure slightly overestimates the true proportion of severe-to-moderate symptoms found in the overall sample (i.e. 29%). However, since we are using it as a guidance to calculate our target sample size, this is a useful feature, due to the fact that any prevalence inferior to 35% would require a smaller sample size, given the same margin of error and level of confidence.

The same Table shows an (unweighted) average of 13% frontline and second-line workers reporting moderate-to-severe symptoms on the GAD-7. Because it is not possible to know whether these "symptomatic" groups are completely or even partly overlapping, we use 35% as a base for our hypothesis (which seems the safest bet given the high comorbidity of anxiety with other conditions).

The Reviewer correctly notes that Lai et al. mention having used 35% in their sample size calculation as expected comorbidity prevalence, and that they refer to Lee et al. (2007) as a source for that rate. However, Lee et al.'s (2007) prevalence rates refer to SARS survivors one year after the outbreak. Technically, Lee et al.'s sample (63 non-healthcare workers and 33 healthcare workers) is different in many respects to Lai et al.'s (2020) and so are the scales they used.

Interestingly, Lee et al. (2007) report prevalence rates for individuals in a chronic phase, whereas Lai et al.'s (2020) prevalence rates refer to an acute phase. Our study lies somewhere in-between, as the acute phase seems to have now ended but we are still very close in time to it (hence we refer to it as a subacute phase in our paper). In Lee et al.'s (2007) study, scores on depression and anxiety subscales are taken (correctly or not) as indicator of "psychological distress". In particular, they report that depressive symptoms were moderate-to-severe in 36.3% of the participants and anxiety symptoms were moderate-to-severe in 36.7% participants. We do not know whether these were (at least partly) the same people. The anxiety figure turns out to be much higher than the comparable figures (i.e. for moderate-to-severe symptoms) reported by Lai et al. (2020) after data collection, which are 15.9% in frontline workers and 9.4% in second-line workers.

Notably, the 44.6% prevalence for anxiety symptoms mentioned by the Reviewer and reported by Lai et al. (2020) in their abstract includes all workers reporting mild-to-severe rather than only those reporting moderate-to-severe symptoms. Similarly, the 71.5% prevalence mentioned by Lai et al. (2020) in their abstract for distress symptoms includes all workers reporting mild-to-severe symptoms reporting mild-to-severe rather than only those reporting moderate-to-severe symptoms. Mindful that there is no current estimate of prevalence for anxiety and/or distress in the population of interest, we believe that 35% is a reasonable starting point for calibrating our sample size.

R4#7. Finally, even if the hypothesised prevalence rate was accurately reflected in the cited papers the samples in these papers differs from the target sample of this study. Lai et al (2020) examined

health care workers in hospitals (nurses and physicians), and Lee et al (2007) examined survivors of SARS. The rationale for the submitted study is that workers in care homes may be particularly susceptible to psychological difficulties. It therefore seems unconvincing that the prevalence rates in the aforementioned studies can be extrapolated to the target sample for this study.

Reply to R4#7. Workers in care homes include a majority of healthcare professionals, and the wider population of healthcare workers is the closest match we could find to our population, which also includes a minority of administrative, technical and professional staff, all of which with potentially some physical and emotional closeness to the residents. We openly recognise that there may be differences but given that we are facing a humanitarian crisis we are using the available data as a starting point this is a useful starting point for an initial inroad into a new population. No prevalence studies have been conducted so far on similar population groups exposed to severe traumatic events in Italy.

R4#8. There are similar issues in the authors' estimations of response rates. They report response rates of 90% in healthcare organisations. However, in the cited materials a mean response rate of 53.8% is reported in health care with a maximum of 94% (Baruch & Holtom, 2008).

Reply to R4#8. We can clarify this point. In the paper, we wrote: "For studies conducted on individuals working in healthcare organizations it is possible to expect very high response rates, up to 90% [39]" meaning that the response rates can be as high as 90% (most likely, only given the right incentives and response facilitators) rather than that the response rate is on average 90%. We did not propose the most optimistic estimate of a 90% response rate but rather an estimate of 70%, based on the literature of reference (Lai et al., 2020) - and on our initial assessment of the potential interest that a survey of the type we are proposing may raise within NCH. To be on even safer ground with our recruitment strategy, however, we are now estimating a 48% response rate, which is below the average reported by Baruch & Holtom (2008).

R4#9. The authors provide sufficient materials that would allow reproducibility of their methods. However, the pre-registered data analysis plan is unclear. The authors state that they will calculate the prevalence of anxiety/distress as a first step, and they will then exclude from the database workers who do not hold an administrative, healthcare, technical or professional job before calculating weighted prevalence rates. However, in the design and recruitment section the authors state that only these individuals will be recruited. It is therefore unclear why these individuals would be retained in the initial prevalence rate calculation. The section on calculating weighted prevalence is rather brief at present which would make replication difficult. Including example analysis code here may be useful.

Reply to R4#9. We have now better specified our pre-registered data analysis plan and, hopefully, given enough detail to allow a clear representation and reproducibility. For the prevalence calculation, we now plan to use STATA. We will produce instructions on how our results can be reproduced with STATA.

R4#10. The materials outlined also include a number of potential moderators not considered in the data analysis plan (e.g. COVID-19 status of care home, engaging in work-related communications, pre-existing staff vs. recruited during the pandemic), increasing the potential for later undisclosed flexible analyses. The authors state that they wish to conduct further exploratory analyses, which presumably include these further measures. However, pre-registered analyses could be considered for these variables.

Reply to R4#10. The potential moderators mentioned by the Reviewer will be the object of careful exploratory analyses, whose results will be presented in a separate section than the preregistered analysis, thus preventing any possible ambiguity as it is customary in registered reports. The main objective of this study consists of sizing the overall burden of psychological symptoms in NCH, and our focus at this stage is on reaching a clear and sufficiently powered design to be able to collect, in

a very timely way, useful data towards answering that main question.

R4#11. There are no details in this submission regarding outcome-neutral conditions for ensuring that the results obtained are able to test the stated hypotheses.

Reply to R4#11. We have now included three control questions from the Wechsler Memory Scale, to be able to detect unreliable/careless respondents.

R4#12. As a minor point, there are errors in the GAD-7 functioning question possible responses ('Not difficult at all' and 'Very difficult' are repeated twice).

Reply to R4#12. We thank the Reviewer for pointing this out, the answers have been changed in the revised version of the survey.

References

Harris JI, Strom TQ, Erbes CR, Ruzek J. Measuring perceived efficacy for coping with posttraumatic stress disorder in the workplace. *Work*. 2019;**63** 283-289.

Elmsley RA, Seedat S, Stein S. Posttraumatic stress disorder and occupational disability in South African security force members. *The Journal of Nervous and Mental Disease*. 2003; 273-41.

Greenberg N, Brooks SK, Wessely S, Tracy, DK. How might the NHS protect the mental health of health-care workers after the COVID-19 crisis? *Lancet Psychiatry*. 2020; Published online May 28.

Hunot V, Churchill R, Silva de Lima M, Teixeira V. Psychological therapies for generalised anxiety disorder. *Cochrane Database Syst Rev*. 2007;**2007**(1):CD001848.

Barbano AC, van der Mei WF, deRoon-Cassini TA, et al. Differentiating PTSD from anxiety and depression: Lessons from the ICD-11 PTSD diagnostic criteria. *Depress Anxiety*. 2019;**36** (6), 490-498.

Heeringa SG, West BT, Berglund PA (2017). *Applied Survey Data Analysis*. Second Ed. CRC Press Taylor & Francis Group (FL).

Lai J, Ma S, Wang Y, et al. Factors Associated With Mental Health Outcomes Among Health Care Workers Exposed to Coronavirus Disease. *JAMA Netw Open*. 2019;**3**(3), e203976

Baruch Y and Holtom BC. Survey response rate levels and trends in organizational research. *Human Relations*. 2008;**61**(8), 1139-1160.

Lee GH, & Lee JM. Impact of facility managers' characteristics on their intention to introduce a unit care system. *Health and Social Welfare Review*. 2012;**32**(4), 94-122.

REVIEWER 5

R5#1. The research question of this study is “What is the prevalence of severe-to-moderate self-reported mental health symptoms of anxiety and/or distress among staff in NCH during the first COVID-19 outbreak in Italy?”

This research question seems reasonably clearly specified. However, I’m somewhat unconvinced that having an answer to it would be of significant scientific or practical utility. All we would learn from the study is that *at one point in time*, some percentage of a sample of residential nursing and care homes in Italy experienced moderate or greater distress or anxiety. We would not know anything about the causal *effect* of COVID-19 on distress or anxiety for this population. Nor would we know whether this population experiences higher levels of distress than other populations. We wouldn’t even really know what levels of anxiety or distress might still be present in this population at the time the results are released to decision-makers who might *do* something to address this anxiety and distress.

Reply to R5#1. The Reviewer is raising a critical issue with clinical implications. We thank her/him for this comment. The assessment of psychological symptoms has relevant scientific and clinical implications for the following reasons: 1) a strong and rigorous body of scientific literature highlighted that the effects of exposure to potentially traumatic events may generate serious long-term consequences on mental health. Although a proportion of subjects might show resilience and spontaneous improvement after the acute phase, large subgroups of vulnerable individuals develop psychological suffering and psychiatric disorders (Betancourt et al. 2019, Panter-Brick et al. 2011, Comtesse et al. 2019, Charlson et al. 2019). 2) the proposed study has an indirect potential value for increasing the implementation of preventative mental health strategies; 3) we would expect that the psychological reactions of our sample might share similarities with those of other population groups exposed to potentially traumatic events. However, it is essential to collect evidence about this instead of indirectly inferring information from other studies in different settings; 4) the present study will partially help to disclose the causal effect of COVID-19 on distress or anxiety for this population. However, a casual pathway could be addressed subsequently considering mediator and moderator analyses on exposure to traumatic events and development and severity of symptoms (using a different study design). We will acknowledge this issue in a paragraph dedicated to the limitations of this study in the discussion of the full manuscript.

R5#2. The authors say “these data [...] will also provide a useful guide for the implementation of timely evidence-based psychological interventions and long-term follow up assessments for NCH workers”—but I struggle to see how that could be the case since these are not things that the planned study directly investigates.

Reply to R5#2. Thank you. We modified the sentence according to the suggestion of the Reviewer (p. 3, lines 38-50). The sentence reads as follows: “The knowledge of detailed socio-demographic and clinical characteristics of this population group has the potential added value of indirectly informing the choice of the intervention category (not the specific intervention type) according to the Inter-Agency Standing Committee pyramid of mental health and psychosocial support interventions in emergency settings (https://www.who.int/mental_health/emergencies/IASC_guidelines.pdf), and long-term follow up assessments for NCH workers. For example, once sized the psychological burden placed on the population, suitable strategies to identify workers possibly requiring an intervention. Workers with lower level of distress might receive basic and/or low-intensity psychosocial interventions on the third or fourth layer of the pyramid, while individuals with higher level of distress might receive structured psychotherapeutic interventions according to resource availability at local level (first layer of the pyramid). This approach is also in line with the WHO mhGAP humanitarian intervention guide (https://apps.who.int/iris/bitstream/handle/10665/162960/9789241548922_eng.pdf)”.

R5#3. As a related note on the original contribution of the study, the authors say “To date, despite the enormous burden of distress and potentially traumatic events experienced by people who work in [nursing and care homes], no studies have documented the prevalence of mental health problems in this population group.” It would be useful to know what search strategy the authors used to determine this.

Reply to R5#3. In order to check for other ongoing similar studies, we performed a preliminary search strategy inspecting the following databases: Pubmed, Medline, Psychinfo, the Cochrane CENTRAL and the Cochrane Library. Moreover, we regularly checked the WHO website, and other sources as the Mental Health and Psychosocial Support Network website, and the Inter-Agency Standing Committee Library. That said, however, we cannot exclude that similar studies are ongoing in Italy or elsewhere. We did not track nor formally screen search records, as this is out of the scope and methodology of the present work.

R5#4. This study does not have any explicit hypotheses. The authors do say “Based on comparable literature on healthcare workers [27], we expect a prevalence of 35%”, but this isn’t a hypothesis per se. The authors do not set any inferential criteria in relation to this prediction.

Reply to R5#4. As the nature of this study is essentially descriptive (with statistical inference concerning population prevalence and confidence intervals), it does not contain a research hypothesis in the style of experimental studies. However, we can propose an inference plan for the possible outcomes given the proportion of people reporting moderate-to-severe psychological symptoms, as acknowledged by this Reviewer in point R5#16.

R5#5. The authors plan to use sample weighting, but the power analysis is for a simple unweighted design with a random sample. Would the weighting not affect the sample size necessary to achieve the desired precision of $\pm 5\%$ around the prevalence estimate?

Reply to R5#5. Thank you for raising this issue, this has now been taken into account in the calculation of our sample size and more details are presented in the revised manuscript.

R5#6. The authors suggest that based on previous studies they expect a response rate of 70% amongst invited participants. I am not familiar with the local cultural context, but based on my own experience this percentage does sound very optimistic. The authors also seem to require permission from “top management” at the various NCHs that would be sampled, but thus far have only spoken with a couple of these managers. As such the feasibility of the recruitment plan is hard to judge.

Reply to R5#6. We thank the reviewer for her/his comment and we are aware that the recruiting phase will require some effort. We have now taken on board the Reviewer’s advice and have revised our estimated response rate. As the recruitment plan is part of our registered report proposal, we could not possibly start before receiving formal approval of our methods. We have informally probed the attitude of other managers in the meantime and found substantial agreement on the urgent need to assess the psychological condition of workers in NCH at this time. Moreover, with the inclusion of a feedback and useful contact information, it was appreciated that our survey does not only fill a research gap but it is also helpful to their employees.

R5#7. There seems to be a lack of clarity about what the IES-R is intended to measure – the authors variously refer to “psychological distress” and “post-traumatic stress”. I do think the IES-R is a slightly awkward fit for this study in that it is usually intended to be used with people who have experienced a *specific* traumatic event that happened in a particular distinct (past) time period—e.g., a car accident, an earthquake, etc. In this study, the stated stimulus is “the spread of COVID-19 infections in Italy”, which is an ongoing and somewhat abstract country-wide phenomenon, not a specific event. This then causes

problems in terms of the applicability of the items. How much sense does it make, for example, to ask participants whether they “found myself acting or feeling like I was back at that time” or “I tried to remove it from my memory” when COVID-19 infections in Italy *are still happening*?

Reply to R5#7. As the Reviewer rightly pointed out, the IES-R is mostly inherent to exposure to traumatic events. According to preliminary clinical work with staff in NCH during the COVID-19 pandemic, staff members were (and still are) exposed to multiple traumatic events, for example seeing elderly people dying suddenly and without any comfort, forced and long periods of isolation (without the possibility of interacting with their close family members), etc. We thus expected the IES-R questions to intuitively make sense to those workers, given the “spread of COVID-19” stimulus. However, we have now taken this Reviewer’s input on board and have rephrased our stimulus in “any stressful events in your nursing/care home due to the recent COVID-19 pandemic”.

R5#8. The authors say that participants “obtaining a score ≥ 10 at the GAD-7 AND/OR obtaining a score ≥ 26 at the IES-R” will be classified as having “moderate to severe symptoms”. These cutoffs have a crucial role in this study, so a clear rationale would have been useful. The authors just say “These categories are based on values established in the literature”, without citations. What *evidence* is there that scores above these cutoffs indicate “moderate to severe symptoms”?

Reply to R5#8. We have now provided additional information and references in Section 3.2.2.

R5#9. More broadly, sumscores from self-report scales have significant limitations as indicators of the presence of mental health conditions. The study aims to estimate “prevalence”, but the authors will be unable to determine the prevalence of any specific mental health disorder or condition (because the scales used are not diagnostic tools). Obviously, every method for measuring psychopathology has limitations, and maybe scores on self-report scales are the best that can be done in this context. But I would have liked to have seen some understanding that alternative methods are available (e.g., diagnostic interviews) and a rationale *for* the selected method.

Reply to R5#9. We agree on the use of diagnostic instruments as an optimal choice for a reliable clinical assessment. However, and due to feasibility issues, it would not be possible to administer time-consuming complex instruments as the MINI Neuropsychiatric interview to such a large group of participants. This would require face-to-face interactions between a clinical psychologist and each participant, that is not recommended by distancing rules, and would require participant identity to be disclosed. Our choice of instruments was also determined by 1) appropriateness of items and psychometric properties of GAD-7 and IES-R; 2) availability of an Italian version; 3) time and resources required for completion (feasibility).

We have now edited the manuscript adding the following sentence (please see Section 3.2.2): “Despite the availability of several valid and reliable instruments to assess the clinical psychological symptoms of participants, as for example the diagnostic MINI Neuropsychiatric Interview, we selected the GAD-7 and IES-R rating scales for feasibility issues, and in consideration of their psychometric properties”. We will address this issue extensively in the discussion of the full manuscript.

R5#10. It appears that translated Italian versions of the measures will be used. This could be fine, but what evidence is there of the reliability and validity of the translated versions?

Reply to R5#10. These instruments have been validated in multiple languages, they are very widely used in the literature and will enable us to compare our results with works of reference on mental health status during COVID-19. Please find relevant references in Section 3.2.2. Despite the availability of several other valid and reliable instruments to assess clinical psychological symptoms

of participants, as for example the diagnostic MINI Neuropsychiatric Interview, we selected the GAD-7 and IES-R for feasibility.

R5#11. The survey contains a number of detailed questions about participants' demographic characteristics, education, work, and living situation. I would be concerned that the information obtained might be sufficient to identify many of the participants if shared openly. It might be necessary to have a separate (and more restrictive) data sharing plan for the demographic information, or to reconsider whether all the information is necessary to collect.

Reply to R5#11. We thank the Reviewer for this comment. We have now eliminated the question on living situation. Also, because geographical areas will be collapsed into broad regions, it will not be possible to guess the identity of NCH, and hence individual workers, from the available data.

R5#12. I was pleased to see the plan to apply sample weighting. However, the authors only attempt to weight on a small number of relevant variables (role type, geographical region, gender; not age or income). Furthermore, the authors do not have access to the proportions in each cross-classification cell in the population about which they intend to make inferences, but only a related population (workers from the Italian national healthcare system as a whole, excluding "hospitals, university hospitals and research centres connected with the national healthcare system"). I can appreciate that the unavailability of relevant population data makes this challenging, but it does add substantial uncertainty to the degree to which the results from the sample can be generalised to the *actual* population of interest.

Reply to R5#12. We agree and have emphasized this limitation in our previous draft; it will be further highlighted in the discussion section of the paper. Though we would like to highlight that good part of the population of reference that we are using coincides with the *actual* population of interest. Although the generalization cannot be as precise as when a population census is available, it will also not be too far from target. This study may also act as a stimulus for a government-led, comprehensive census on workers in NCH. In this case, our raw data will be available to anyone for re-weighting.

R5#13. How is it that the population proportions in Table 2 do not sum to 1?

Reply to R5#13. Previously, these data were normalized by stratum. However, we now focus only on the population in the first row that corresponds to the North of Italy. A previous rounding error, giving a total 0.9999, has been fixed.

R5#14. The authors say that they will discard incomplete surveys (i.e., listwise deletion). Listwise deletion will cause biased estimates unless the data is missing completely at random, which is rarely a plausible assumption (see Allison, 2001; Schlomer et al., 2010). It would be good to see a more sophisticated plan for dealing with missing data (e.g., multiple imputation), or an argument for why listwise deletion is the most appropriate strategy.

Reply to R5#14. Thank you for raising the issue. Every method for dealing with missing data has strengths and weaknesses and no method can fully control lack of randomness. In our survey, only those workers who are responding to every item can proceed to the next page. We may thus have missing pages (e.g. a responder exits the survey after filling the demographic questionnaire and the GAD-7). Because we expect the feedback placed at the end of the survey to be an important target and motivator for this population, we judge that those who take the time complete the whole survey are more likely to respond truthfully and accurately to these self-report scales. We are fully aware that listwise deletion may lead to having to collect more data in order to reach the target sample size but this is also a neat way to improve the accuracy of our estimates from the sample. Surveys with missing pages will be logged, therefore we will still have the possibility to apply a different

approach in further exploratory analyses. We could not afford doing the opposite, instead, as the sample size would then not be sufficient for a listwise deletion.

R5#15. The authors say that a 95% confidence interval will be reported based on an assumed Poisson distribution. I think they might mean a binomial distribution.

Reply to R5#15. The Reviewer is correct. We will now be using STATA to calculate exact confidence intervals, which assumes a binomial distribution.

R5#16. Regarding inferential criteria (bottom row of Table 1): The authors plan to infer what variety of mental health care is necessary for this sector of workers based on the prevalence of moderate to severe anxiety or distress. For example, if the prevalence is between 10 and 30%, they will infer that “focused psychosocial interventions” are required; if the prevalence is greater than 50% then “psychotherapy” will be required; etc. Put bluntly, this interpretation scheme does not make sense. Mental health *varies* from person to person. Even if the prevalence of moderate to severe anxiety or distress is low, some individuals may have very severe needs, and thus require highly specialised supports. You cannot reasonably infer what services are required by considering only one feature of the distribution of mental health conditions in the population (i.e., the proportion above some cutoff on some numeric scale).

Reply to R5#16. We thank the reviewer for this insightful comment. We edited Table 1 in order to make it clearer for readers. We agree that every individual may experience different mental health conditions and that there are individual and social determinants of mental health that should inform the implementation of interventions for each individual. We are also aware on the enormous treatment gap still existing in mental health, so that more than 70% of people with mental disorders (and up to 90% in low resource settings) do not receive care at all (<https://apps.who.int/iris/bitstream/handle/10665/275386/9789241514811-eng.pdf?ua=1>). We will address this issue extensively in the discussion of the full manuscript.

R5#17. The study is reasonably simple in nature (cross-sectional survey, no experimental manipulation), so replicating it theoretically wouldn't be overly difficult. Nevertheless, the fact that the actual (Italian) materials aren't currently available is one obvious issue.

Reply to R5#17. We have added the Italian materials to the Appendix.

R5#18. The authors allude to exclusions in a few places, but no comprehensive list of exclusion criteria seems to have been provided. It would be useful to consider issues like detecting careless responses (see Meade & Craig, 2012), whether there are sufficient checks to detect participants outside of the intended population, detecting duplicate responses, etc.

Reply to R5#18. We believe that the survey topic should be sufficiently relevant and motivating for our target population. This is in contrast to other populations, such as students, for which survey participation and/or survey contents may not be particularly important (Meade & Craig 2012). However, we will tackle the issue of inattentive or careless responses by adding three extra questions (one at the end of the demographic questionnaire, one at the end of the GAD-7, one at the end of the IES-R) taken from the Information section of the Italian Wechsler Memory scale. These are very simple questions (Who is the president of Italy? Who was the president of Italy before the current one? What is the name of the current Pope?). If a respondent does not answer correctly to at least 2 of the three questions, their data will be excluded.

We will also use IP tracking, to detect and prevent duplicate responses (no more than one response can be sent from the same device).

Participants will be contacted by the researchers through their organization (in which they usually form a close-knit community), they will be requested to avoid environmental distraction when completing the survey and to possibly use their personal rather than work devices (as a further

measure to preserve anonymity). Given the level of commitment required, the specialist/professional tuning of the demographic questionnaire and the circulation of the survey from within the organizational community, we do not expect a significant number of participants outside of the intended population.

R5#19. Relatedly, the authors say that incomplete responses will be discarded, but it's unclear what "counts" as incomplete (Missing an item on the GAD-7 or IES-R? Missing *any* item on the survey?)

Reply to R5#19. As we have now better specified in the text, "incomplete" essentially means missing the demographic questionnaire, the GAD-7 and the IES-R questionnaires. We will not consider incomplete responses those missing the final wellbeing questionnaire instead.

R5#20. The algorithm/method and software that will be used for the sample weighting is not specified. A data processing and analysis script could have allowed for a much clearer indication that the analysis pipeline has been adequately specified.

Reply to R5#20. We have now provided more detailed information on our weighting method and software. We will be using STATA for our calculations and will produce instructions on how to reproduce our results.

R5#21. This is not an experiment so some of the considerations that might typically apply here (e.g., positive controls, manipulation checks) are not relevant. The primary concern I have is the lack of a plan to deal with the scenario of slow recruitment – e.g., if the data collection end date is reached and the sample size is less than the target of 350, what would the authors do? Would they conclude that the sample is not sufficient to answer the stated research question?

Reply to R5#21. We thank the Reviewer for this comment. The target sample size has now increased and we are aware that it will be challenging to reach in a relatively brief time. However, in case of slower than expected recruitment, it seems unlikely that we would have to push back the end date into a possible Phase 3, provided that the study can start soon enough. Additionally, we are now planning a more capillary and intensive sampling approach, within the same broad region as that of our probe NCH.

R5#22. The authors say that "Anxiety is characterized by the excessive worry about everyday events and problems to the point at which the individual experiences considerable distress in performing day to day tasks." This sounds more like generalized anxiety disorder than "anxiety" the emotion or a general description of anxiety disorders.

Reply to R5#22. We have now edited the sentence in accordance with the Reviewer's comment.

R5#23. The authors say "Last but not least, this survey will represent a first important step for workers to open up about their mental health at work and to become aware that, even in their work environment, there are psychologists who can listen to their difficulties, in person, and provide support accordingly to help them face the remainder of this emergency and beyond." I find it hard to see how a participant filling out an anonymous online survey with fixed-choice items would be likely to receive those benefits.

Reply to R5#23. We thank the Reviewer for this comment. Psychological problems in the working environment are a very sensitive issue - most of workers do not like to show their problems to the entire community. In this sense, the feedback we are offering could be a first step for them to become aware of a particular state. We agree with the Reviewer, however, and for this reason we added supportive feedback from a useful website <https://www.traumagroup.org/> as suggested by Reviewer 2. We also provide the phone number for the national free psychological support line. Further, participants are given the possibility to write to the authors or leave any comments on page 5 of the survey. We believe that this is a good way to establish contact with them (if they wish to contact), to give them a chance to ask for and find help.

R5#24. Page 4, “A Cochrane systematic review of 33 randomized trials on psychological therapies for individuals living in humanitarian settings...” Maybe this should read “individuals living in countries affected by humanitarian crises”?

Reply to R5#24. We have now deleted this section on suggestion from other Reviewers.

R5#25. The note stating that “Indeed, most of the workers appear quite oblivious of obvious signs of physical and psychological distress, as they are not only to continue working and but also dealing with an increased workload during this emergency” strikes me as inappropriate to include in the absence of supportive evidence. Do just be aware that the word “oblivious” in English carries a negative connotation (i.e., the sentence could be read as being a little insulting towards the workers).

Reply to R5#25. This is an interesting point, which we were completely oblivious to. Following your advice, we have now replaced “oblivious” with “unaware”. Indeed, evidence of workers being unaware of their psychological distress has been reported during the COVID-19 outbreak in Wuhan: “*Moreover, individual nurses showed excitability, irritability, unwillingness to rest, and signs of psychological distress, but refused any psychological help and stated that they did not have any problems. In a 30-min interview survey with 13 medical staff at The Second Xiangya Hospital, several reasons were discovered for this refusal of help*” (Chen et al. 2020).

References

Betancourt TS, Thomson DL, Brennan RT, Antonaccio CM, Gilman SE, VanderWeele TJ. Stigma and Acceptance of Sierra Leone's Child Soldiers: A Prospective Longitudinal Study of Adult Mental Health and Social Functioning. *J Am Acad Child Adolesc Psychiatry*. 2019;S0890-8567(19)30392-2.

Panter-Brick C, Goodman A, Tol W, Eggerman M. Mental health and childhood adversities: a longitudinal study in Kabul, Afghanistan. *J Am Acad Child Adolesc Psychiatry*. 2011;50(4):349-363.

Comtesse H, Powell S, Soldo A, Hagl M, Rosner R. Long-term psychological distress of Bosnian war survivors: an 11-year follow-up of former displaced persons, returnees, and stayers. *BMC Psychiatry*. 2019;19(1):1.

Charlson F, van Ommeren M, Flaxman A, Cornett J, Whiteford H, Saxena S. New WHO prevalence estimates of mental disorders in conflict settings: a systematic review and meta-analysis. *Lancet*. 2019;394(10194):240- 248.

Chen Q, Liang M, Li Y, et al. Mental health care for medical staff in China during the COVID-19 outbreak. *Lancet Psychiatry*. 2020;7(4):e15- e16.

Appendix D

Thank you for revising the current manuscript. I am satisfied with how the authors addressed most of my comments and with the overall content of the manuscript. I think that the focus on the North of Italy is a particularly good choice that will make the study more focused and also more feasible. Glad that you found the COVID trauma group information helpful.

I have one more substantial comment (here below) and then some minor ones:

Pg. 1, line 38: I still find it a bit confusing that you move interchangeably from PTSD to psychological distress throughout the manuscript and I would try and maintain a clear distinction between these constructs. Psychological distress can be a perfectly normal reaction to extreme events that most people will likely experience during and in the aftermath of such events. PTSD on the other hand is obviously different. I know that the IES is sometimes referred to as a measure of subjective distress (caused by traumatic event) but I think that describing it as a measure of post-traumatic stress reactions, or post-traumatic symptomatology might be more correct (as the authors rightly stress in the materials section). The items in the IES are clearly connected to PTSD-like symptoms (avoidance, re-experiencing, hyper-arousal) and I think that referring to it simply as “psychological distress” is misleading. I would suggest either to substitute “psychological distress” with “post-traumatic symptomatology” or “post-traumatic stress reactions” throughout the manuscript or to use a measure that taps into psychological distress rather than post-traumatic symptomatology.

Pg. 1, line 51: I feel that these next lines are quite number heavy and the reader kind of gets a bit lost in all these different estimates. Possibly try and focus on the most significant numbers only for clarity's sake.

Pg. 3, figure: Very minor point. The increasing saturation of red makes it look like it has some meaning (rather than simply showing time moving on) and readers might interpret the color as a coding for the increasing severity of epidemic (which would make sense up to April-May but then should start becoming lighter?). You could simply choose different colors for different months?

Pg. 4, line 2: You already mention above that patients and workers have strong bonds so would remove here.

Pg. 4, line 17: I would be cautious about saying that up to 40% of populations in humanitarian settings have PTSD. Are these all studies using diagnostic assessments administered by clinicians or simple screening tools? If it's the latter I would talk about “probable rates of PTSD” or “PTSD symptomatology”.

Pg. 4, line 39: I am not very convinced by how studying prevalence might inform the choice of the IASC pyramid level. Interventions across all IASC levels should be present and interconnected during an emergency independently from prevalence. Also, even if you found that most people would require clinical services that doesn't mean they also don't need social considerations in basic services and security right (and vice-versa)? Same comment applies to the interpretation section in Table 1.

Pg. 4, line 46: I feel that these kind of interventions (e.g. online courses etc.) would not fall in the first layer of the IASC pyramid? The first layer is specifically about “security, adequate governance and services that address basic physical needs”. I would possibly place them between 2nd (key community and family supports) and 3rd layer (more focused individual, family or group interventions by trained and supervised workers) but definitely not 1st?
https://interagencystandingcommittee.org/system/files/legacy_files/Checklist%20for%20field%20use%20IASC%20MHPSS.pdf

Pg 4, line 52: I would add family, friends to psychologists as these are the figures that people will most likely go to when in distress rather than psychologists?

Pg. 5, figure (caption): I think my point above on level of IASC intervention also arises from some confusion on how the layers of the pyramid are numbered. I believe that generally in the IASC guidelines the bottom is considered the first layer and top is fourth but in the caption there seems to be some confusion in the numbering?

Pg. 5, line 15: reference for GAD-7 cut-off still missing here?

Pg. 7, Table 1, interpretation section: Not sure what is meant by “prevalence < 10: unlikely to be an organization-related burden”? Also a prevalence of 9% of severe-to-moderate mental health symptoms still seems quite substantial to me and worthy of being addressed?

Pg. 8, line 43: Not sure why hyper-arousal mentioned as a “third component of PTSD”? Are authors referring to ICD-11 classification? Could simply put “a core component of PTSD”.

Pg. 9, line 10: I think that you might also want to ask whether they themselves have contracted COVID-19? I think that would be a very important stressor to consider. I guess you would have to ask if they *think* they have contracted COVID as many might have not been tested but possibly still a very meaningful thing to consider (especially if you are looking at Northern Italy?)

Pg. 9, line 19: I am not hugely convinced by these attention check questions. First, you might inadvertently exclude people from lesser educated background that might not be up to date with Italian politics. Secondly you might inadvertently exclude international migrant workers that might be a relevant proportion in certain of your job categories as they might be less in touch with Italian politics? What about less loaded questions that would still work as attention checks, like for example “what is the capital of Italy?”, “in which region is Milan?”

Pg. 17, question 5. Couldn't you simply include the regions of Northern Italy as options and say here that if they work outside of these regions unfortunately they cannot participate but still provide them with the details to the psychosocial support that you have at the end?

Besides some minor points I am satisfied with how the authors have addressed my comments. I wish you all the best with data collection.

Alessandro Massazza

Appendix E

REVIEWER 1

Thank-you for revising the manuscript and for providing sufficient responses and revisions to all of my original concerns.

We are grateful to the Reviewer for this positive feedback.

REVIEWER 2

Thank you for revising the current manuscript. I am satisfied with how the authors addressed most of my comments and with the overall content of the manuscript. I think that the focus on the North of Italy is a particularly good choice that will make the study more focused and also more feasible. Glad that you found the COVID trauma group information helpful.

I have one more substantial comment (here below) and then some minor ones:

R2#1: Pg. 1, line 38: I still find it a bit confusing that you move interchangeably from PTSD to psychological distress throughout the manuscript and I would try and maintain a clear distinction between these constructs. Psychological distress can be a perfectly normal reaction to extreme events that most people will likely experience during and in the aftermath of such events. PTSD on the other hand is obviously different. I know that the IES is sometimes referred to as a measure of subjective distress (caused by traumatic event) but I think that describing it as a measure of post-traumatic stress reactions, or post-traumatic symptomatology might be more correct (as the authors rightly stress in the materials section). The items in the IES are clearly connected to PTSD-like symptoms (avoidance, re-experiencing, hyper-arousal) and I think that referring to it simply as “psychological distress” is misleading. I would suggest either to substitute “psychological distress” with “post-traumatic symptomatology” or “post-traumatic stress reactions” throughout the manuscript or to use a measure that taps into psychological distress rather than post-traumatic symptomatology.

Reply to R2#1: *We have now harmonized the terminology across the manuscript, using the term “post-traumatic symptomatology” as suggested, instead of distress or psychological distress (which we agree might be misleading).*

R2#2: Pg. 1, line 51 (corresponds to “In 2020, 23.1% of the total population in Italy..” and following): I feel that these next lines are quite number heavy and the reader kind of gets a bit lost in all these different estimates. Possibly try and focus on the most significant numbers only for clarity’s sake.

Reply to R2#2: *We have now removed two of these statistics – the proportion of deaths in people aged 60+ and the proportion of the Italian population living in the North. They can now be found in the legend of Figure 1 and in Note 1 respectively.*

R2#3: Pg. 3, figure: Very minor point. The increasing saturation of red makes it look like it has some meaning (rather than simply showing time moving on) and readers might interpret the color as a coding for the increasing severity of epidemic (which would make sense up to April-May but then should start becoming lighter?). You could simply choose different colors for different months?

Reply to R2#3: *The saturation gradient has been removed from Figure 2.*

R2#4: Pg. 4, line 2 (in NCH, resident patients and staff tend to form close relationships, due to their prolonged contact): You already mention above that patients and workers have strong bonds so would remove here.

Reply to R2#4: *We have removed this sentence.*

R2#5: Pg. 4, line 17 “In individuals who had been exposed to traumatic events in humanitarian settings, rates of PTSD and anxiety may reach a prevalence of 30-40% (according to the type and amount of stressors [21,22]).”: I would be cautious about saying that up to 40% of populations in humanitarian

settings have PTSD. Are these all studies using diagnostic assessments administered by clinicians or simple screening tools? If it's the latter I would talk about "probable rates of PTSD" or "PTSD symptomatology".

Reply to R2#5: *We agree with the Reviewer. Since in humanitarian settings it is not always feasible to administer diagnostic instruments, we cautiously added the word "symptomatology" after PTSD and anxiety, as suggested.*

R2#6: Pg. 4, line 39 "The knowledge of detailed socio-demographic and clinical characteristics of this population group has the potential added value of indirectly informing the choice of the intervention category (not the specific intervention type) according to the Inter-Agency Standing Committee (IASC) pyramid of mental health and psychosocial support interventions in emergency settings [52]": I am not very convinced by how studying prevalence might inform the choice of the IASC pyramid level. Interventions across all IASC levels should be present and interconnected during an emergency independently from prevalence. Also, even if you found that most people would require clinical services that doesn't mean they also don't need social considerations in basic services and security right (and vice-versa)? Same comment applies to the interpretation section in Table 1.

Reply to R2#6: *This is a very important point and we thank the Reviewer for this comment. The IASC guidelines have been mainly developed to respond to humanitarian emergencies. The core idea behind the work of the IASC taskforce is that even basic social supports are essential to protect and support the mental health and psychosocial well-being of people exposed to potentially traumatic events. The recommended selected psychosocial, psychological and psychiatric interventions at the top levels are intended to be used for specific problems (e.g. difficulties that need personalized assistance), and not as a substitution of basic human rights considerations or social support placed at lower levels. The layers are defined as complementary not as mutually exclusive.*

We have now clarified this in a new sentence on page 5, just under the pyramid (Figure 3 caption: "Interventions across layers are intended to be complementary and not mutually exclusive").

R2#7: Pg. 4, line 46 "Psychological interventions covering the areas of anxiety and distress fall in the first two layers of the IASC pyramid and might include online training courses to help staff to deal with psychological problems and online psychological group or individual activities to release stress [32]": I feel that these kind of interventions (e.g. online courses etc.) would not fall in the first layer of the IASC pyramid? The first layer is specifically about "security, adequate governance and services that address basic physical needs". I would possibly place them between 2nd (key community and family supports) and 3rd layer (more focused individual, family or group interventions by trained and supervised workers) but definitely not 1st?

https://interagencystandingcommittee.org/system/files/legacy_files/Checklist%20for%20field%20use%20IASC%20MHPSS.pdf

Reply to R2#7: *We have now replaced the term "online training courses" with "online psychotherapeutic interventions".*

R2#8: Pg 4, line 52 "inside or outside their work environment, there are psychologists who can listen to their difficulties, in person,": I would add family, friends to psychologists as these are the figures that people will most likely go to when in distress rather than psychologists?

Reply to R2#8: *Thanks for pointing this out. However, we have now removed this paragraph in a restructuring of this section to improve clarity.*

R2#9: Pg. 5, figure (caption): I think my point above on level of IASC intervention also arises from some confusion on how the layers of the pyramid are numbered. I believe that generally in the IASC guidelines the bottom is considered the first layer and top is fourth but in the caption there seems to be some confusion in the numbering?

Reply to R2#9: *We fixed this issue in the manuscript text and in the caption of Figure 3. We have now also numbered the layers in Figure 3. We apologize for the confusion caused by our previous wording.*

R2#10: Pg. 5, line 15: reference for GAD-7 cut-off still missing here?

Reply to R2#10: *All the references – both those referring to GAD-7 and those referring to IES-R - are given at the end of section 3.2.2: "These categories are based on values established in the literature [28, 41, 69,*

63, 70, 27]”, which is where the reader was previously referred to on page 5, after the IES-R acronym. We have made it clearer that this reference also applies to GAD-7.

R2#11: Pg. 7, Table 1, interpretation section: Not sure what is meant by “prevalence < 10: unlikely to be an organization-related burden”? Also a prevalence of 9% of severe-to-moderate mental health symptoms still seems quite substantial to me and worthy of being addressed?

Reply to R2#11: *We agree with this comment from the Reviewer. Even a prevalence rate below 10% is clinically important and requires due attention. This would probably fall outside the scope of the present work though, as it would reflect the mental health condition of the general population.*

In accordance with this Reviewer’s comment, we edited Table 1 as follows: “Prevalence ≤ 10; Unlikely to be an emergency-related burden due to the pandemic. Probable need to liaise with local public health services to explore psychological state and provide appropriate intervention”.

R2#12: Pg. 8, line 43: Not sure why hyper-arousal mentioned as a “third component of PTSD”? Are authors referring to ICD-11 classification? Could simply put “a core component of PTSD”.

Reply to R2#12: *We have edited the text according to the Reviewer’s suggestion, and now indicate hyper-arousal as “a core component of PTSD”.*

R2#13: Pg. 9, line 10: I think that you might also want to ask whether they themselves have contracted COVID-19? I think that would be a very important stressor to consider. I guess you would have to ask if they *think* they have contracted COVID as many might have not been tested but possibly still a very meaningful thing to consider (especially if you are looking at Northern Italy?)

Reply to R2#13: *We agree with the Reviewer that personally contracting COVID-19 would be an important stressor. However, having previously consulted the ethics committee on this point, we prefer not to ask this question for privacy reasons. Positive COVID-19 people were forced to stay at home in quarantine, they are easily recognizable and may feel marginalized. Asking whether they think they have contracted COVID maybe more subtle but would still be problematic, as any workers with suspicious symptoms are meant to report this to their manager and they may feel uncomfortable responding truthfully (although we tell them that the survey is completely anonymous, respondents may prefer to be extra-cautious and not respond truthfully to this question). Last but not least, it is a loaded question that may trigger anxiety and act as a deal breaker (causing respondents to withdraw from the survey at this very point).*

R2#14: Pg. 9, line 19: I am not hugely convinced by these attention check questions. First, you might inadvertently exclude people from lesser educated background that might not be up to date with Italian politics. Secondly you might inadvertently exclude international migrant workers that might be a relevant proportion in certain of your job categories as they might be less in touch with Italian politics? What about less loaded questions that would still work as attention checks, like for example “what is the capital of Italy?”, “in which region is Milan?”

Reply to R2#14: *We thank the Reviewer for this suggestion. As far as questions about national politics are concerned, we think it is true that non-Italian staff may have difficulties. When a question is too specific, also geography may perhaps be problematic. Therefore, we have replaced the two questions about Italian presidents with a basic question about international politics (Who is the current President of the United States?) which should be known by anyone regardless of their level of education or their nationality, and a simple question about geography (What is the capital of Italy?).*

R2#15: Pg. 17, question 5. Couldn’t you simply include the regions of Northern Italy as options and say here that if they work outside of these regions unfortunately they cannot participate but still provide them with the details to the psychosocial support that you have at the end?

Reply to R2#15: *We appreciate the Reviewer’s comment. However, our survey will remain available to any Italian nursing or care home. If some respondents are not located in the North of Italy, their data will not be included in the statistical analyses for this report.*

Besides some minor points I am satisfied with how the authors have addressed my comments. I wish you all the best with data collection.

REVIEWER 4

R4#1: Thank you to the authors for taking the time to address my concerns in detail, and for clarifying the sources of the estimated prevalence rate. The majority of my previous concerns have been addressed, however I believe it may still be worth considering whether it would be appropriate to report anxiety and distress prevalence rates separately. The authors have provided a clear rationale in the introduction as to why they have chosen to focus on anxiety and distress symptoms. However, there is little detail for the decision to combine these into a single prevalence rate and why this is more beneficial than considering these symptoms independently. As similar studies in this area (e.g. Lai et al, 2020) have reported distress and anxiety separately I think it would be worth outlining the decision process for this.

Reply to R4#1: *We would prefer to keep anxiety and PTSD combined in one index, as we expect that symptoms of anxiety and PTSD in this population group could present similarities and overlap. We reasoned that considering anxiety and PTSD in one index will provide a more precise estimate of the psychological state of participants for the following line of arguments: 1) we are administering the questionnaire some weeks after the emergency peak. At this time, while PTSD symptomatology may unfortunately still be present, it is also possible that participants have developed symptoms of anxiety; 2) in the available literature studies on this topic, for example, Lai et al. (2020) found high levels of PTSD symptoms and lower levels of anxiety during the emergency peak, while Lee et al. (2007) found higher levels of anxiety months after the emergency peak; 3) in addition please note that also in high profile prevalence studies different disorders have been combined in one prevalence rate, for example in the recent WHO prevalence estimates of mental disorders in conflict settings (Charlson et al., 2019, table 2; https://pubmed.ncbi.nlm.nih.gov/31200992/?utm_source=gquery&utm_medium=referral&utm_campaign=CitationSensor). That said, we plan to conduct secondary subgroup analyses to check prevalence rates separately.*

We have now added the following statement to the Introduction: “PTSD and anxiety symptomatology will be considered in one prevalence index to capture a more comprehensive estimate of the psychological state of participants. Available studies on this topic, for example, found high levels of PTSD symptoms and lower levels of anxiety during the emergency peak [27], while in other studies conducted months after the acute emergency levels of anxiety were higher [15]”.

R4#2: As a minor point, the introduction would benefit from further refinement at times. For example, the last paragraph on page 2 of the manuscript, line 38 onwards, the authors conclude ‘This has had inevitable consequences for the management of residents’. However, it is unclear exactly what exactly these inevitable consequences are.

Reply to R4#2: *The inevitable consequences refer to the limitation of social activities in terms of frequency and management (e.g. wearing a mask). We have now added examples, with references, in correspondence with this point in the manuscript.*

R4#3: There are also some minor typos in the English translation of study materials:

- Item 7 of the GAD-7 English translation, ‘MORE THAN AHALF THE DAYS’
- Question 3 on the quality of life scale, ‘Where you productive (comleting many tasks) in most activities’.

Reply to R4#3: *Thanks, the typos have now been corrected.*

REVIEWER 5

The authors have provided a fast revision, with thoughtful responses to my comments and those of the other (4!) reviewers. I consider the authors’ responses to many of my points to be satisfactory, and have therefore focused this review on what I see as the remaining issues.

R5#1 (R5#1) (my comments regarding the utility of the study): Some of what the authors write in their response seems very sensible. However, in their response letter they state “(4) the present study will partially help to disclose the causal effect of COVID-19 on distress or anxiety for this population.” This is not the case; in Cook and Campbell’s terminology (Shadish, Cook, & Campbell, 2001), the study is a “one-shot case study”. The authors won’t be able to establish a correlation between anxiety or distress and COVID-19, and

certainly can't establish a causal effect (not even tentatively – no statistic in the planned analysis is an estimator of a causal effect). While the causal inference does not seem to be implied in the current manuscript (just the response letter) it is important that the authors do not claim that the study tells us something about the causal effect of COVID-19 in either the discussion section of the eventual manuscript or any ensuing media coverage.

Reply to R5#1: *We thank the Reviewer for this insightful comment. We are aware that our study in isolation does not present the characteristics for discovering a causal relationship between the exposure to the COVID-19 pandemic and the mental health condition of participants. We will consider this comment carefully when interpreting and presenting the results.*

R5#2 (R5#7): I appreciate the author's useful revisions to the instructions to the participants for the IES-R. However, it remains the case that the questions in the IES-R ask participants about a single event (e.g., "pictures about it popped into my mind"), while the customised instructions relate to a (plural) set of multiple possible events ("any stressful events in your nursing/care home due to the recent COVID-19 pandemic"). The customised instructions thus don't quite fit the items grammatically. Whether this problem is obviated in the Italian translation, I don't know. I'm also left a little unsure whether participants will have a clear idea of which events they're meant to be reflecting on given the rather open-ended instructions.

Reply to R5#2: *Thank you for raising this issue. We have now eliminated the grammatical inconsistency by referring to a single event ("any stressful event"), which mirrors more closely the Italian translation. We have also added some examples to the instructions ("e.g. the death of a resident or of a colleague, the shortage of personal protective equipment at work, having had to follow new health and safety protocols, such as wearing a protective mask, maintaining social distancing sanitizing rooms, liaising with relatives remotely and/or communicating possible deaths, etc."), to give participants a better idea of the type of the potentially traumatic event they should focus on and to make reference to in the following questions.*

R5#3: As per my comment in the first round there is still inconsistency within the manuscript in relation to what the IES-R is meant to be measuring ("distress", "psychological distress", "post-traumatic stress").

Reply to R5#3: *The Reviewer highlighted an important issue for improving manuscript clarity. As suggested also by Reviewer 2, we have now harmonized the terminology across the manuscript, and avoided mixing words such as "distress" and "psychological distress" and "post-traumatic stress". We followed the suggestion from Reviewer 2 using the term "PTSD symptomatology", which is in line with our aim and with the instrument (IES-R) that will be administered.*

R5#4 (R5#16): I appreciate the author's attempt to add some more nuance to the interpretation scheme in Table 1, but my concern here remains. Mental health varies from person-to-person. For example, even with a prevalence of moderate distress/anxiety under 10%, some individuals may still experience severe psychological dysfunction and require structured and intensive psychological (or psychiatric) interventions. I don't believe that the inferences the authors are seeking to draw about which services should be offered are ones that logically follow from the statistic they plan to report (i.e., a prevalence estimate). I leave this to the editor's judgment but my recommendation is that this interpretation scheme be removed.

Reply to R5#4: *We agree that even a prevalence rate below 10% is clinically important and requires due attention. According to this comment and also to a comment from Reviewer 2, we edited Table 1 as follows: "prevalence \leq 10 Unlikely to be an emergency-related burden due to the pandemic. Probable need to liaise with local public health services for exploring psychological status and providing appropriate intervention". That said, we leave the final decision to keep or remove the table to the Editor.*

R5#5 (R5#21): I can appreciate the author's attention to the feasibility of the sample size planning, but this wasn't quite what I was asking for (perhaps I didn't express myself entirely clearly). In a registered report, the inferential criteria need to make it clear how a failure to achieve the target sample size will affect the inferences drawn. I.e., will the study still be reported if the sample size is less than 900? How would this affect how you describe the results? (The editor may have suggestions here).

Reply to R5#5: *The minimum sample size required to obtain an estimate with the set confidence and margin of error is actually 847 (which we rounded up to 900 to be conservative). In our power calculations we have also kept the estimated prevalence reasonably high (please see our reply to R4#6 in the previous round of*

reviews). If the actual prevalence in our sample was lower than .35, the minimum sample size required to obtain an estimate with the set confidence and margin of error would also be lower (e.g. *coeteris paribus*, it would be about 780 for a prevalence of .30, about 698 for a prevalence of .25, about 595 for a prevalence of .20, and about 475 for a prevalence of .15). On the other hand, if the actual prevalence was slightly higher than .35 (say .41) we would need a larger sample (about 900) respondents to estimate population parameters with the same confidence and margin of error (this actually corresponds to our current target sample size). If it was (unexpectedly) much higher than .35 (say .50), the required sample would be even larger (about 930). For any prevalence higher than .50, we will need a sample size smaller than 930.

Would we not report our data in this case? Of course we would, but our margin of error would be slightly higher than the initially set 5% (i.e. the brackets of the confidence interval will be slightly larger). The same applies to the case in which we were unable to reach our target sample size. If the actual prevalence turns out to be .35, and we only collect data from about 590 respondents, the margin of error of our estimate will increase from 0.05 to 0.06. This has now been spelled out in section 3.1.1.

Rovereto, 4th September 2020

Appendix F

Dear Professor Chambers,

Please find enclosed our Stage 2 manuscript, in which we have extended the Stage 1 manuscript by adding Result, Discussion and Conclusion sections. All the other sections are identical to the Stage 1 accepted manuscript with the following exceptions:

1. A change to the Author list.

Due to unforeseen circumstances and timing issues, Angela Federico could not help with participant recruitment (i.e. the contribution specified in the Author Credits list submitted with our Stage 1 manuscript); Chiara Bove stepped in and took on the task in her stead, thus she now appears as a coauthor rather than Angela Federico. As we did specify in the submission system at Stage 1 that the status of Angela's contribution was only "expected", this change has not been flagged with a note in our Stage 2 manuscript.

2. A typographical error in the relvariance formula – with no bearings on the reported calculations - has been fixed (please see note 3 in the manuscript).
3. The start and end dates of our survey have now been included in Figure 2 (timeline of COVID-19 events in Italy) and in the main text.
4. All verb tenses have been updated where opportune.
5. The previous abstract has been tweaked and updated, to make space for an overview of the main results.

Any changes to the previously approved text have been highlighted in the manuscript for your and the reviewers' convenience. We have included the survey data file and the scripts to perform statistical analyses as electronic supplementary materials. These will be moved

UNIVERSITY
OF TRENTO - Italy

Department of Psychology
and Cognitive Science

to the OSF repository upon acceptance (link provided in the manuscript). The materials used to collect the data and their English translation are provided with the manuscript as appendices, as in the Stage 1 report.

We are very mindful of the urgency of publishing this work to inform the field on the one hand and to enable the timely implementation of intervention studies on the other.

Therefore, we have submitted our report ahead of schedule.

We hope that you and the reviewers will find the Stage 2 manuscript to be suitable for publication in the Royal Society Open Science journal and we will be looking forward to hearing from you.

Thank you very much in advance for your time and your feedback,

Marianna Riello, Marianna Purgato, Chiara Bove,

David MacTaggart & Elena Rusconi

Appendix G

Review of RSOS-200880 R3 (Stage 2)

In this Stage 2 Registered Report, the authors report the results of their preregistered analyses as well as some additional exploratory analyses. I have a small number of points to raise at this stage:

- It's good to see that the raw data and syntax files seem to be provided, but I couldn't open or check these as I don't have Stata. Ideally the syntax would be provided with an open source language (e.g., R). At the least the *data* should be available in a non-proprietary format (e.g. .csv).
- Page 12, line 21, is the phrase "had an educational level *up* to high school (752 [70.1%])" correct? Surely at least some participants had some tertiary education?
- Section 4.1.3 reports the actual preregistered data analysis; from what I can tell they follow the preregistered plan. However, the vast majority of the results section covers additional exploratory analyses. I can appreciate that exploratory analyses can sometimes be appropriate in a Registered Report, but they then need to be clearly justified, and techniques suitable for exploratory analysis employed. Here, the "exploratory" analyses rely heavily on statistical hypothesis testing (i.e., *p* values), even though the hypotheses tested weren't preregistered. There also isn't a very explicit rationale for the analyses: Why are these analyses so important to include that they need to take up most of the Results section, while *not* being important enough to have been preregistered? How can the reader be assured that these analyses are adequately powered, especially considering the small sample sizes in some cells? I would actually suggest that almost all of the exploratory analyses be removed, bar section 4.2.4 and the first two paragraphs of section 4.2.2 (which just report some relevant descriptive statistics).
- The fact that the logistic regression includes only those predictors which had a bivariate association with the outcome with $p < 0.1$ is especially questionable. Selecting analyses to report based on which coefficients are statistically significant will result in biased estimates, and is the exact practice that a registered report is designed to avoid.
- The study ultimately concludes a very high prevalence "43% for anxiety and/or PTSD symptoms of moderate-to-severe intensity". In the first round of review I asked, "What *evidence* is there that scores above these cut-offs indicate "moderate to severe symptoms"?" This concern was never directly addressed; the authors said "These categories are based on values established in the literature [27, 28, 41, 63, 69, 70]", but just giving citations to papers that have suggested or used these cut-offs before is not a very convincing rationale. In the discussion section, I would now suggest that the authors reflect on the evidence basis for these cut-offs – what **evidence** do we have that these thresholds have high sensitivity and specificity in the identification of moderate anxiety and or PTSD? Consider, for example, that if the specificity of the cut-offs is low, then the prevalence estimate reported could be a substantial overestimate.
- "Based on the above comparisons, we speculate that our prevalence *datum* may represent a conservative estimate of the psychological burden in Northern Italian NCH": Is a "psychological burden" really an estimable quantity? The study certainly does *not* represent a conservative estimate of *diagnosable* PTSD or anxiety disorders – if anything, the opposite is true.

- Section 5.4: The authors speculate on potential reasons for the higher prevalence of PTSD symptoms as opposed to anxiety symptoms. One possible explanation is simply that the cut-off for PTSD symptoms may have had lower specificity (i.e., that it was a more “lenient” cut-off).
- Section 5.5: It’s good to see limitations identified clearly. It would be worth noting that the design of the study makes it impossible to determine the causal *effect* of COVID-19 on PTSD or anxiety symptoms.
- Data accessibility statement: The link to supporting data and research materials is not working (I suspect the OSF project is still set to private)

Appendix H

Thank you very much for giving me the possibility to review this manuscript on the mental health of people working in nursing and care homes in Northern Italy during the COVID-19 pandemic. As the authors rightly point out, this is a valuable study on an under-researched population and the authors have been able to design and conduct the study in a very short time span. I believe the data are able to test the authors' proposed hypothesis, that the rationale is the same as that approved at Stage 1, that the authors adhered precisely to the registered procedures, and that the authors' conclusions are justified by the data.

There are a number of issues I would like to highlight that I hope will help the authors in improving the manuscript.

My main concern is around the collapsing of the two conditions (PTSD and anxiety) in one prevalence rate. The rationale behind this choice does not seem hugely clear to me. Especially in light of DSM-5 classification specifically removing PTSD from the anxiety disorders cluster. These are two distinct conditions with distinct sets of symptoms, so I find this approach somewhat unusual as it feels a bit like mixing apples and pears. This is especially the case since you highlight in the intro (and confirm in your data) how the two conditions might have different trajectories and prevalence (27). Furthermore, simply based on the combined prevalence, it is impossible for the reader to know whether the "moderate-to-severe mental health" cases are mostly due to anxiety or to PTSD which is an important distinction (as you highlight yourself in the Discussion). Similarly, I feel it is impossible to properly interpret the logistic regression with the combined prevalence as one cannot establish whether the effect is due to the association with anxiety or PTSD specifically. Furthermore, the collapse of the two measures makes it look like more people have moderate to severe mental health problems (44%) whereas the actual prevalence of moderate-to-severe anxiety is 22% and of moderate-to-severe PTSD 40%. Therefore, I would suggest avoiding the combined prevalence rate of PTSD-anxiety by removing 4.1.3, 4.2.1, and the overall prevalence rate in Table 6, and only reporting separate results for PTSD and anxiety as in 4.2.2 as I am struggling to see what the combined prevalence is adding to the results. Indeed, the interpretation of the results is much clearer when you disaggregate the two conditions (e.g. contact with COVID-19 patients being more related to PTSD than anxiety which makes sense and is an interesting finding). If you really need to keep the combined prevalence in order to match protocol, I would try and prioritize the separate data throughout the manuscript.

Some smaller points below:

SUMMARY

pg. 1, line 10: I would remove "which were taken off guard". While it's true that Italy did not have the benefit of time as in other countries, "being take off guard" is not the only reason why nursing homes were hit so hard. There were governmental and systemic reasons that led to nursing homes being left in a particularly vulnerable position, e.g. under-funding of public health system, delays in PPE delivery etc.

pg. 1, line 21: I would avoid including "Phase 2" in abstract as might not be clear for international audience. You could maybe specify the months when data collection took place and say something like "X months after national lockdown was declared".

Expand the acronym NCH at some point in summary and introduction.

INTRO

pg. 2, line 4: Please update with more recent death count (i.e. August/September)

pg. 2, line 39: “Moreover...”. Sentence too long, please shorten.

pg. 3, line 9: I am not sure pantomime is what you mean? Pantomime is a form of theatrical entertainment?

pg. 3, line 32: Provide exact dates.

pg. 4, line 19: substitute “thoughts” with “memories” and possibly say “avoidance of internal and external reminders of the trauma” as more in line with DSM-5 and ICD-11

pg. 4, line 20: singular “mood”

pg. 4, line 21: Please specify these rates are from US.

pg. 4, line 22: Please update with PTSD prevalence rates that are more relevant to this study. Most of the studies included in the reviews referenced (21,22) are on conflict or disaster settings which might have very different characteristics to the pandemic in Italy. There is quite some literature on PTSD in the context of Ebola and other epidemics among health-care workers that might be more relevant. Some possible references that might provide you with some useful data-points:

Allan, S. M., Bealey, R., Birch, J., Cushing, T., Parke, S., Sergi, G., ... Meiser-Stedman, R. (2020). The prevalence of common and stress-related mental health disorders in healthcare workers based in pandemic-affected hospitals. A rapid systematic review and meta-analysis. *medRxiv*. doi: <https://doi.org/10.1101/2020.05.04.20089862>

Bell, V. & Wade, D. (2020). Mental health of clinical staff working in high-risk epidemic and pandemic health emergencies: A rapid review of the evidence and meta-analysis. *medRxiv*. doi: <https://doi.org/10.1101/2020.04.28.20082669>

pg. 4, line 24: “However, precise information...”. True, but unclear how this links to the paragraph?

pg. 4, line 30: Unclear what is meant by “trauma recognition” being a risk factor for PTSD?

pg. 5, line 10: “In the current...”. To me this would appear more appropriate in the Discussion section? Additionally, as said previously, I don’t understand the point made concerning how prevalence rates could inform the choice of intervention category. Independently from whether you find a prevalence of 10% or 90% ,as the IASC guidelines highlight, “*All layers of the pyramid are important and should ideally be implemented concurrently*”. Even if someone received a formal diagnosis of PTSD/GAD etc. they would still need layers i, ii, and iii on top of layer iv.

pg. 5., line 28: Figure 3: “At the bottom...”, wrong-labelling of layers’ number, please invert.

pg. 6, line 3: Please specify “data on staff working in NHCs”.

pg. 7, line 20: Table 1: In “Question”, please substitute psychological distress with “PTSD symptomatology”

MATERIALS AND METHODS

pg. 8, line 23: Please specify how randomization of NHCs took place

pg. 8, line 42: Might be useful to provide Cronbach’s alpha for GAD-7 and IES-R in the current study as well.

pg. 9, line 11: You could say “Total scores range from 0 to 21” and same for IES-R

RESULTS

pg. 14: I am not sure I understand the added value of Section 4.2.3 after 4.2.2 as it feels like they are answering the same question with the same data using a slightly different analytical approach? Indeed, the results are the same in the two sections. I would possibly remove section 4.2.3 and only keep 4.2.2.

pg. 14. Section 4.2.4. Same point as for 4.2.3. Not sure what this section is adding conceptually to the paper?

DISCUSSION

pg. 15, line 13: In primis, mostly Italian expression

pg. 17, line 4: I would add as a reason for high rates of symptoms the higher likelihood of being exposed to the virus as a healthcare professional (this is supported in your exploratory analysis in relation to PTSD).

pg. 17, line 18: I would specify that De Sio et al. administered their questionnaire at the peak of the first wave in Italy (1-12 April) which might explain the very high rates of distress. Also the rate of 93.8% is only for the doctors living in most affected areas (n = 212) (see Table 1 in their paper: <https://www.europeanreview.org/wp/wp-content/uploads/7869-7879.pdf>) whereas the overall percentage for the entire sample is 89.06%.

pg. 17, line 23: I am not sure which rates the authors are reporting from the Di Tella study (97%)? From Table 1 of that study (<https://onlinelibrary.wiley.com/doi/epdf/10.1111/jep.13444>) it seems that they report anxiety rates above cut-off at 71% and PTSD rates above cut-off at 26.2%. I would avoid clumping together different diagnoses.

pg. 17, line 28: As above, in the Rossi et al study I would report the percentage of participants for each separate condition and not aggregate it together.

pg. 17: All this section on studies among Italian health-workers can be shortened considerably (e.g. less methodological detail on each study)

pg. 17, line 34: Not sure what is meant by “comparable calculation”, please specify

pg. 17, line 41: I don't think that your estimates are "conservative", quite the opposite. Rates of probable PTSD at 39% are much much higher than general population and there is a real risk of over-estimation issues due to the time at which data was collected (i.e. during the emergency itself) + self-report.

pg. 18: line 9: As mentioned in the Intro, it might be best to keep comparison of rates limited to similar populations (i.e. healthcare workers during epidemics) rather than expanding to humanitarian workers who are likely to be a very different sub-population with different stressors (e.g. humanitarian workers aware they are going into a war zone, nurses did not sign up to work during a pandemic).

pg. 18, line 17: All this section on gender differences can be shortened considerably

pg. 18, line 42: This is the reason why it is confusing to aggregate anxiety and PTSD because it makes it look like "workers who had contact with COVID-positive colleagues/patients report more severe symptom levels compared to workers who had not had contact with COVID-positive colleagues/patients", which is a bit misleading as this effect is really driven by the relation that COVID-19 exposure has with PTSD specifically and not with anxiety.

pg. 19, line 5: This is very interesting

pg. 19, line 15: Very interesting and with important implications for interventions in health settings.

pg. 19, line 33: If true, wouldn't have this however resulted in lower PTSD rates among people working in NHCs where family members were allowed in?

pg. 19, line 35: I think this section could possibly be left out.

pg. 20: line 13: Unclear how your study speaks to formal PTSD since you used a self-report measure?

pg. 21, line 1: Another thing that should definitely be measured in future studies is functioning.

pg. 21, line 6: Yes, but I would separate these two points, i.e. 1. You used self-report which is vulnerable to over-reporting when compared to structured clinical assessment, especially in emergency settings (see Bromet et al. (<https://pubmed.ncbi.nlm.nih.gov/27573281/>) review showing PTSD prevalence of 20-40% when using self-report versus 3-5% when measured in proper clinical assessments) 2. Your design doesn't really allow you to 100% say that these rates are due to COVID-19 because you don't have a pre-event mental health measurement (https://www.researchgate.net/publication/289725686_Methodological_challenges_in_studying_the_mental_health_consequences_of_disasters).

pg. 21, line 12: Please remove "vulnerable personalities who struggled to deal with the pandemic". Many of these issues have little to do with personality and it risks perpetuating stigma.

pg. 21: Another aspect, that isn't really a limitation, but more of a note of caution, is that your rates are higher than those in the general population probably also because of the time in which you collected this data. Experiencing high levels of anxiety and PTS-symptoms would have probably still been completely normal when your participants were still in a semi-emergency context. Indeed, as a general note, I think it might be worth thinking about whether you were measuring PTS-symptoms or simply acute stress symptoms. Indeed, there was yet no "post" when you collected your data as people were still working in the context of a continuing pandemic. This is not to say that high level of acute stress shouldn't be addressed via interventions.

Overall the discussion is excessively long and could be made more succinct.

EDITORIAL COMMENTS

Some sentences are generally too long and sometimes feel a bit convoluted.

I hope this feedback will be useful for the authors and again congratulations on collecting the data despite context. I wish you all the best with the manuscript.

Kind regards,

Alessandro Massazza

Appendix I

Dear Editor,

We have now made substantial revisions based on the comments of all the referees. All changes to the manuscript are highlighted (including deleted text) and extra files (data and outputs) have been added to the supplementary materials.

Based on your recent comments, we would like to bring some general points to your attention and will now do so by reviewing the points of each reviewer in turn:

Reviewer 2 provided us with 43 points to address and we have done this in detail. We are in disagreement with him on some of his main points. As you highlighted in your decision letter, Reviewer 2 appears now mainly concerned about the collapsing of anxiety and PTSD together. However, we had already argued, pre-IPA, why this approach is suitable for our study. We would also like to highlight that the point was originally raised by Reviewer 4, which appears largely satisfied with how the analyses were conducted. We believe that it would put the review process in crisis and essentially undermine the pre-registration process, if we were to continue to re-address previous points. This problem is amplified if reviewers start picking up other reviewers' points as their main objection to publication. If this were to happen in another round of Reviews, we would ask you to make a decision.

Reviewer 4 raised only two minor points, which we have answered. We believe that s/he is satisfied with our work.

Reviewer 5 is the most critical of our work and seems to be, in principle, against this type of study. Although we have answered Reviewer 5's comments and have edited the manuscript based on most of these, we do not agree with the substantial reductions that s/he is asking. Reviewer 5 is asking us to remove all exploratory analysis that is not directly linked to our pre-registered report. This would effectively leave us with a couple of lines stating the main prevalence result and very little comment about it. And of course the Limitation section, which is what Reviewer 5 liked of the manuscript. We do not believe that this is (a) enough for a journal of this quality and (b) instrumental to improve the rigour and the significance of our contribution. We disagree with Reviewer 5's definition of exploratory analysis and with his/her stance that it should not really be included in registered reports. We do believe that it is important to make sure that any methodology is stated clearly and that the reader has enough details and materials (codes, instructions, etc.) to be able to reproduce the results, and to explore the data further if they wish. The exploratory analysis that we have performed is truly exploratory. Contrary to what Reviewer 5 states, we are performing exploratory logistic regressions and non-parametric tests to discover if several demographic and work-related variables may be connected to our result. Hence, the inclusion of exploratory analyses provides a nice complement to our pre-registered analysis. Our exploratory analysis is presented in much more detail compared to similar studies and its inclusion in a stage 2 pre-registered report is fully justified.

After more than 40 pages of reviewers' questions and replies, it is fair to say that we have undergone a rigorous review process. However, we now see this starting to run in circles. Ideally, we would like to avoid another round of "major revisions" that involve having to answer points that we have already addressed, debate the philosophy of registered reports or take on the task to justify, within our manuscript, the sheer existence of a whole field of research. If this were to occur, we would ask you to intervene and make a decision. Also, if the review-revision process is delayed (e.g. due to the start of teaching), this would be very damaging for this work as the results need to be made public now in order to have some impact.

After all of this, we would like to thank both you and all the reviewers for taking the time to edit and review this paper. Even though we disagree with some of the reviewers' comments, we genuinely appreciate their effort in helping us to improve this paper.

Thank you for your time.

Yours sincerely,

The Authors.

P.S.: For review purposes, research materials are provided at the following private link:
https://osf.io/gmba/?view_only=50ddf0065efa48b7932167bbcde48b08. The OSF project will be made public upon manuscript acceptance.

REVIEWER 2

Thank you very much for giving me the possibility to review this manuscript on the mental health of people working in nursing and care homes in Northern Italy during the COVID-19 pandemic. As the authors rightly point out, this is a valuable study on an under-researched population and the authors have been able to design and conduct the study in a very short time span. I believe the data are able to test the authors' proposed hypothesis, that the rationale is the same as that approved at Stage 1, that the authors adhered precisely to the registered procedures, and that the authors' conclusions are justified by the data.

There are a number of issues I would like to highlight that I hope will help the authors in improving the manuscript.

R2#1: My main concern is around the collapsing of the two conditions (PTSD and anxiety) in one prevalence rate. The rationale behind this choice does not seem hugely clear to me. Especially in light of DSM-5 classification specifically removing PTSD from the anxiety disorders cluster. These are two distinct conditions with distinct sets of symptoms, so I find this approach somewhat unusual as it feels a bit like mixing apples and pears. This is especially the case since you highlight in the intro (and confirm in your data) how the two conditions might have different trajectories and prevalence (27).

Reply to R2#1: This is a very important point to address. However, this point was originally raised by R4 and we have already addressed it before IPA.

We refer the Reviewer to our previous response:

"We would prefer to keep anxiety and PTSD combined in one index, as we expect that symptoms of anxiety and PTSD in this population group could present similarities and overlap. We reasoned that considering anxiety and PTSD in one index will provide a more precise estimate of the psychological state of participants for the following line of arguments: 1) we are administering the questionnaire some weeks after the emergency peak. At this time, while PTSD symptomatology may unfortunately still be present, it is also possible that participants have developed symptoms of anxiety; 2) in the available literature studies on this topic, for example, Lai et al. (2020) found high levels of PTSD symptoms and lower levels of anxiety during the emergency peak, while Lee et al. (2007) found higher levels of anxiety months after the emergency peak; 3) in addition please note that also in high profile prevalence studies different disorders have been combined in one prevalence rate, for example in the recent WHO prevalence estimates of mental disorders in conflict settings (Charlson et al., 2019, table 2; https://pubmed.ncbi.nlm.nih.gov/31200992/?utm_source=gquery&utm_medium=referral&utm_campaign=CitationSensor). That said, we plan to conduct secondary subgroup analyses to check prevalence rates separately.

We have now added the following statement to the Introduction: "PTSD and anxiety symptomatology will be considered in one prevalence index to capture a more comprehensive estimate of the psychological state of participants. Available studies on this topic, for example, found high levels of PTSD symptoms and lower levels of anxiety during the emergency peak [27], while in other studies conducted months after the acute emergency levels of anxiety were higher [15]".

We have now also included a brief reminder of our rationale in the Discussion, at the beginning of section 5.2.

R2#2: Furthermore, simply based on the combined prevalence, it is impossible for the reader to know whether the "moderate-to-severe mental health" cases are mostly due to anxiety or to PTSD which is an important distinction (as you highlight yourself in the Discussion). Similarly, I feel it is impossible to properly interpret the logistic regression with the combined prevalence as one cannot establish whether the effect is due to the association with anxiety or PTSD specifically.

Reply to R2#2: We understand the Reviewer's point, which is a corollary of R2#2. However, we believe the logistic regression is meaningful for the following reasons:

1) From an organizational and professional perspective, any psychological issue will require a different type of investment and approach compared to other types of issues (e.g. medical).

2) In emergency settings, the majority of individual and group interventions that can prevent the development of a wide range of mental health conditions have been developed as supportive interventions independently of the type of mental health symptomatology. The main intervention components are intended to support individuals affected by losses, traumatic events or chronic/acute stressors that occur in emergency situations. For example, international guidelines list among scalable interventions the following: 1. Brief and non-specialist delivered basic versions of psychological interventions; 2. Self-help materials and guided self-help programmes. These interventions have been proven to be effective and safe for large population groups suffering disabling stress and anxiety (WHO, 2017).

We do acknowledge the important concern raised by Reviewer in relation to the distinction between PTSD and anxiety. However, this applies mainly in a clinical setting when a formal diagnosis of mental disorder is established, in order to offer the most appropriate and structured psychotherapeutic intervention.

3) The main focus of this paper is on sizing up the magnitude of the problem on the whole (pre-registered analysis). A secondary aim is to identify groups of individuals that may need targeted intervention (logistic regression in exploratory analysis, which highlights the association between certain demographic characteristics and the presence of severe-to-moderate symptoms in one or both the conditions).

In summary, our primary interest is not in the specific mechanism underlying workers' distress. In the absence of any previous evidence we would like to know first what proportion of workers may be experiencing a problem. Naturally, the data lend toward a further exploration of the components of this problem, which is what we have done in further exploratory analyses.

R2#3: Furthermore, the collapse of the two measures makes it look like more people have moderate to severe mental health problems (44%) whereas the actual prevalence of moderate-to-severe anxiety is 22% and of moderate-to-severe PTSD 40%. Therefore, I would suggest avoiding the combined prevalence rate of PTSD-anxiety by removing 4.1.3, 4.2.1, and the overall prevalence rate in Table 6, and only reporting separate results for PTSD and anxiety as in 4.2.2 as I am struggling to see what the combined prevalence is adding to the results. Indeed, the interpretation of the results is much clearer when you disaggregate the two conditions (e.g. contact with COVID-19 patients being more related to PTSD than anxiety which makes sense and is an interesting finding). If you really need to keep the combined prevalence in order to match protocol, I would try and prioritize the separate data throughout the manuscript.

Reply to R2#3: Please note that as in previous responses this point has already been addressed and cannot be changed after IPA. However, we would like to clarify a possible misunderstanding here. In order to answer our main research question of interest, this way of presenting things is more conservative than the standard way. Indeed, it actually returns a lower prevalence of individuals with likely mental health issues than the disaggregate figures considered together. In other terms, it directly answers the question: what proportion of our respondents is in a significant state of distress and currently at risk of developing a psychiatric condition? The disaggregated prevalence estimates, either singularly or in combination, cannot directly answer this question. 44% is the real proportion of individuals reporting issues of any kind (based on the available measures). 62% (i.e. 22% anxiety + 40% PTSD) is not, because the disaggregated prevalence figures do not take into account repeat counts (i.e. those individuals who reach the threshold both for one and for the other measure). Therefore, they do not answer the main question we are interested in, i.e. how many individuals report significant symptoms (and as a consequence may require a psychological intervention in NCH post-COVID). The disaggregated figures can help answering other questions (e.g. re: the

specific characteristics and possible underlying mechanisms of mental health symptoms), which we have addressed with exploratory analyses and in the appropriate section of our Discussion.

Some smaller points below:

SUMMARY

R2#4: pg. 1, line 10: I would remove “which were taken off guard”. While it’s true that Italy did not have the benefit of time as in other countries, “being take off guard” is not the only reason why nursing homes were hit so hard. There were governmental and systemic reasons that led to nursing homes being left in a particularly vulnerable position, e.g. under-funding of public health system, delays in PPE delivery etc.

Reply to R2#4: We agree with the Reviewer and “which were taken off guard” has now been replaced with “which were hit hard”.

R2#5: pg. 1, line 21: I would avoid including “Phase 2” in abstract as might not be clear for international audience. You could maybe specify the months when data collection took place and say something like “X months after national lockdown was declared”. Expand the acronym NCH at some point in summary and introduction.

Reply to R2#4: Done. The acronym has now been eliminated from the Abstract and introduced on Page 2, line 13.

INTRO

R2#6: pg. 2, line 4: Please update with more recent death count (i.e. August/September)

Reply to R2#6: The sentence about death counts is intended to set up the context for our data collection - its update would defy this purpose and require rather extensive modifications to the preregistered document, which we are not allowed to do.

R2#7: pg. 2, line 39: “Moreover...”. Sentence too long, please shorten.

Reply to R2#7: Done.

R2#8: pg. 3, line 9: I am not sure pantomime is what you mean? Pantomime is a form of theatrical entertainment?

Reply to R2#8: To avoid ambiguity this has now been replaced with “miming”.

R2#9: pg. 3, line 32: Provide exact dates.

Reply to R2#9: Done.

R2#10: pg. 4, line 19: substitute “thoughts” with “memories” and possibly say “avoidance of internal and external reminders of the trauma” as more in line with DSM-5 and ICD-11

Reply to R2#10: Done.

R2#11: pg. 4, line 20: singular “mood”

Reply to R2#11: Done.

R2#12: pg. 4, line 21: Please specify these rates are from US.

Reply to R2#12: Done.

R2#13: pg. 4, line 22: Please update with PTSD prevalence rates that are more relevant to this study.

Most of the studies included in the reviews referenced (21,22) are on conflict or disaster settings which might have very different characteristics to the pandemic in Italy. There is quite some literature on PTSD in the context of Ebola and other epidemics among health-care workers that

might be more relevant. Some possible references that might provide you with some useful data-points:

Allan, S. M., Bealey, R., Birch, J., Cushing, T., Parke, S., Sergi, G., ... Meiser-Stedman, R. (2020). The prevalence of common and stress-related mental health disorders in healthcare workers based in pandemic-affected hospitals. A rapid systematic review and meta-analysis. *medRxiv*. doi: <https://doi.org/10.1101/2020.05.04.20089862>

Bell, V. & Wade, D. (2020). Mental health of clinical staff working in high-risk epidemic and pandemic health emergencies: A rapid review of the evidence and meta-analysis. *medRxiv*. doi: <https://doi.org/10.1101/2020.04.28.20082669>

Reply to R2#13: Thank you for pointing these out. We have now added the suggested studies. In addition, we have cited the review of Kisely and colleagues, 2020 (Kisely S, Warren N, McMahon L, Dalais C, Henry I, Siskind D. Occurrence, prevention, and management of the psychological effects of emerging virus outbreaks on healthcare workers: rapid review and meta-analysis. *BMJ*. 2020;369:m1642. Published 2020 May 5. doi:10.1136/bmj.m1642).

R2#14: pg. 4, line 24: “However, precise information...”. True, but unclear how this links to the paragraph?

Reply to R2#14: The sentence is now better connected with the paragraph and a relevant reference has been provided. The sentence now reads as follows: “However, precise information on the time since trauma exposure, that could heavily impact symptom intensity (King et al., 2006), is often not available from the current literature”

R2#15: pg. 4, line 30: Unclear what is meant by “trauma recognition” being a risk factor for PTSD?

Reply to R2#15: We replaced “recognition” with “awareness” and rephrased that sentence.

R2#16: pg. 5, line 10: “In the current...”. To me this would appear more appropriate in the Discussion section? Additionally, as said previously, I don’t understand the point made concerning how prevalence rates could inform the choice of intervention category.

Independently from whether you find a prevalence of 10% or 90%, as the IASC guidelines highlight, “*All layers of the pyramid are important and should ideally be implemented concurrently*”. Even if someone received a formal diagnosis of PTSD/GAD etc. they would still need layers i, ii, and iii on top of layer iv.

Reply to R2#16: We appreciate this comment. However, as this paragraph belongs to the pre-registered manuscript we are unable to move it to the Discussion section. The Reviewer is right in pointing out this issue, which has implications at the clinical level and has been addressed before IPA. Our work may only indirectly provide information on the most appropriate intervention category, given the context. The aim is to inform policy makers and organizations on the size of the problem, which requires monitoring and further assessment of the psychological status of staff in NCH and, ideally, the implementation of large scale intervention strategies in the next steps. This paragraph, therefore, suggests that depending on the severity of symptoms within the population, organizations may invest on different types of intervention (with different levels of the pyramid enabling timely interventions at a different scale). Indeed, these organizations cannot afford to take responsibility for providing “all layers of the pyramid”. We are well aware that our study design (survey) is appropriate for collecting large scale data on prevalence in a timely way but it is only able to provide indirect information at the clinical/intervention level.

R2#17: pg. 5., line 28: Figure 3: “At the bottom...”, wrong-labelling of layers’ number, please invert.

Reply to R2#17: We double-checked the labelling of the layers in the original version of the IASC guidelines (https://www.who.int/mental_health/emergencies/guidelines_iasc_mental_health_psychosocial_jun

e_2007.pdf?ua=1) and we believe that the numbering reported in our manuscript is correct (please see p. 11-13 of the document available at the provided link).

R2#18: pg. 6, line 3: Please specify “data on staff working in NHCs”.

Reply to R2#18: Done.

R2#19: pg. 7, line 20: Table 1: In “Question”, please substitute psychological distress with “PTSD symptomatology”

Reply to R2#19: Done.

MATERIALS AND METHODS

R2#20: pg. 8, line 23: Please specify how randomization of NHCs took place

Reply to R2#20: Thanks, we have now added this specification in the manuscript.

R2#21: pg. 8, line 42: Might be useful to provide Cronbach’s alpha for GAD-7 and IES-R in the current study as well.

Reply to R2#21: Thanks. We have now reported the values we found (0.89 for GAD-7 and 0.93 for IES-R; Intrusion: 0.90 Avoidance: 0.81 Hyperarousal: 0.84) in the Result section (4.2.4).

R2#22: pg. 9, line 11: You could say “Total scores range from 0 to 21” and same for IES-R

Reply to R2#22: Done.

RESULTS

R2#23: pg. 14: I am not sure I understand the added value of Section 4.2.3 after 4.2.2 as it feels like they are answering the same question with the same data using a slightly different analytical approach? Indeed, the results are the same in the two sections. I would possibly remove section 4.2.3 and only keep 4.2.2.

Reply to R2#23: These address similar but not identical questions and provide evidence that even when looking at the data across different scales of aggregation and using different analytical approaches, results are consistent. The extra Table presented in section 4.2.3 shows a cross-tabulation of data that would not otherwise be available to the reader and is meant to ease comparability across studies. Last but not least, our exploratory analysis follows a similar approach to that adopted by Lai et al. (2020), for direct comparability.

R2#24: pg. 14. Section 4.2.4. Same point as for 4.2.3. Not sure what this section is adding conceptually to the paper?

Reply to R2#24: 4.2.3 provides an extra prevalence estimate, 4.2.4 focuses on scale and subscale scores (which enable separate analyses for the IES-R subscales). Also in this case the sections address similar but not identical issues and provide converging evidence at different scales of data aggregation. For comparability, we followed a similar approach to that used in Lai et al. (2020).

DISCUSSION

R2#25: pg. 15, line 13: In primis, mostly Italian expression

Reply to R2#25: The Reviewer is correct – this would only be clear to Italians (or classical scholars). The expression has now been replaced with “Firstly”.

R2#26: pg. 17, line 4: I would add as a reason for high rates of symptoms the higher likelihood of being exposed to the virus as a healthcare professional (this is supported in your exploratory analysis in relation to PTSD).

Reply to R2#26: In this paragraph we mention possible reasons why the prevalence in NCH is particularly high by emphasising differences with a hospital setting. We have therefore added the

point suggested by the Reviewer in the opening sentence: “The high levels of anxiety and/or PTSD symptoms found in the present survey might be influenced by the interplay of several factors, in addition to the higher likelihood of being exposed to the virus as healthcare professionals.”

R2#27: pg. 17, line 18: I would specify that De Sio et al. administered their questionnaire at the peak of the first wave in Italy (1-12 April) which might explain the very high rates of distress. Also the rate of 93.8% is only for the doctors living in most affected areas (n = 212) (see Table 1 in their paper: <https://www.europeanreview.org/wp/wp-content/uploads/7869-7879.pdf>) whereas the overall percentage for the entire sample is 89.06%.

Reply to R2#27: The Reviewer is correct. We have now replaced 94% with 89% and specified that the study was conducted around the peak of contagions in Italy.

R2#28: pg. 17, line 23: I am not sure which rates the authors are reporting from the Di Tella study (97%)? From Table 1 of that study (<https://onlinelibrary.wiley.com/doi/epdf/10.1111/jep.13444>) it seems that they report anxiety rates above cut-off at 71% and PTSD rates above cut-off at 26.2%. I would avoid clumping together different diagnoses.

Reply to R2#28: As here we are comparing the data available from the literature with our main outcome (with the aim to return an estimate of the proportion of individuals with relevant symptomatology of anxiety and/or PTSD), it is necessary to consider these together – with the caveat that some individuals are likely to be counted twice (i.e. they contribute both to the prevalence rate of one type of symptoms and to the prevalence rate of the other). Hence the specification in text about unknown comorbidity rates.

R2#29: pg. 17, line 28: As above, in the Rossi et al study I would report the percentage of participants for each separate condition and not aggregate it together.

Reply to R2#29: Please see above.

R2#30: pg. 17: All this section on studies among Italian health-workers can be shortened considerably (e.g. less methodological detail on each study)

Reply to R2#30: Given the heterogeneity of approaches in the literature tackling similar issues, methodological details are essential for a reader to understand the scope of the reported percentages and to fully appreciate the differences in outcomes across studies. We have now slightly shortened this section, while maintaining all the necessary details.

R2#31: pg. 17, line 34: Not sure what is meant by “comparable calculation”, please specify

Reply to R2#31: We have now replaced this with “same calculation”, as we are adding the proportion of individuals reporting severe anxiety to the proportion of individuals reporting severe PTSD symptomatology for comparison purposes with Rossi et al. (2020).

R2#32: pg. 17, line 41: I don’t think that your estimates are “conservative”, quite the opposite. Rates of probable PTSD at 39% are much much higher than general population and there is a real risk of over-estimation issues due to the time at which data was collected (i.e. during the emergency itself) + self-report.

Reply to R2#32: This paragraph ends a section where we show that other Italian studies measuring self-reported symptomatology of anxiety and PTSD among healthcare workers in the midst of Phase 1 tend to report higher percentages than ours. However, the prevalence data reported in the aforementioned Italian studies are also much higher than those reported in Lai et al. (healthcare, peak of contagions, China) – which are only slightly lower than ours.

Among the possible (all speculative) reasons behind these discrepancies, we suspect that the way in which participants have been recruited in our study – which is more similar to Lai’s than to the other available Italian studies – may have led to a more conservative (lower) estimate than the

recruitment method used in the aforementioned studies. We have now specified in the manuscript against what benchmark (i.e. the “extant literature on Italian healthcare workers”) we are defining our *datum* as “conservative”.

R2#33: pg. 18: line 9: As mentioned in the Intro, it might be best to keep comparison of rates limited to similar populations (i.e. healthcare workers during epidemics) rather than expanding to humanitarian workers who are likely to be a very different sub-population with different stressors (e.g. humanitarian workers aware they are going into a war zone, nurses did not sign up to work during a pandemic).

Reply to R2#33: We have followed the Reviewer’s suggestion and we eliminated the comparison with a systematic review by Strohmeier and Scholte (p. 18, lines 8-15). This also helps to shorten the Discussion as suggested by this Reviewer in other comments (and also by R4).

R2#34: pg. 18, line 17: All this section on gender differences can be shortened considerably

Reply to R2#34: We have now shortened this section by eliminating the following lines: page 18, lines 23-25; page 18, lines 32-33 and 34-36.

R2#35: pg. 18, line 42: This is the reason why it is confusing to aggregate anxiety and PTSD because it makes it look like “workers who had contact with COVID-positive colleagues/patients report more severe symptom levels compared to workers who had not had contact with COVID-positive colleagues/patients”, which is a bit misleading as this effect is really driven by the relation that COVID-19 exposure has with PTSD specifically and not with anxiety.

Reply to R2#35: Please see our Reply to R2#2.

pg. 19, line 5: This is very interesting

pg. 19, line 15: Very interesting and with important implications for interventions in health settings.

R2#36: pg. 19, line 33: If true, wouldn’t have this however resulted in lower PTSD rates among people working in NHCs where family members were allowed in?

Reply to R#36: Not necessarily. Indeed, it is also possible to expect higher rates in NCH where family visits were allowed, because of the heightened risk of contagion. As stated at the end of this paragraph, this, combined with evidence from a recent Dutch study, suggests that “family visits” may be an ambiguous factor (which would explain why we did not find it either positively or negatively related to PTSD symptomatology).

R2#37: pg. 19, line 35: I think this section could possibly be left out.

Reply to R#37: Agreed. This has now been eliminated.

R2#38: pg. 20: line 13: Unclear how your study speaks to formal PTSD since you used a self-report measure?

Reply to R#38: The reference to our study has been eliminated.

R2#39: pg. 21, line 1: Another thing that should definitely be measured in future studies is functioning.

Reply to R#39: We agree, indeed we mention functioning on page 21, line 12.

R2#40: pg. 21, line 6: Yes, but I would separate these two points, i.e. 1. You used self-report which is vulnerable to over-reporting when compared to structured clinical assessment, especially in emergency settings (see Bromet et al. (<https://pubmed.ncbi.nlm.nih.gov/27573281/>) review showing PTSD prevalence of 20-40% when using self-report versus 3-5% when measured in proper clinical assessments) 2. Your design doesn’t really allow you to 100% say that these rates are due to COVID-19 because you don’t have a pre-event mental health measurement

(https://www.researchgate.net/publication/289725686_Methodological_challenges_in_studying_the_mental_health_consequences_of_disasters).

Reply to R2#40: This is a useful suggestion to improve clarity and we edited the Discussion accordingly. The sentence now reads as follows: “Second, we did not use diagnostic tools (e.g. Mini International Neuropsychiatric Interview) or other clinician-administered tools but self-administered instruments which could be vulnerable to over-reporting when compared to structured clinical assessment, especially in emergency settings (Bromet et al. 2017). We were also unable to distinguish preexisting mental health disturbances from new symptoms caused essentially by the pandemic, as we did not have pre-pandemic data from the participants involved in this survey (Galea et al., 2008)”.

R2#41: pg. 21, line 12: Please remove “vulnerable personalities who struggled to deal with the pandemic”. Many of these issues have little to do with personality and it risks perpetuating stigma.

Reply to R2#41: We have now replaced the wording “vulnerable personalities who struggled to deal with the pandemic” with “subgroups of individuals”.

R2#42: pg. 21: Another aspect, that isn't really a limitation, but more of a note of caution, is that your rates are higher than those in the general population probably also because of the time in which you collected this data. Experiencing high levels of anxiety and PTS-symptoms would have probably still been completely normal when your participants were still in a semiemergency context. Indeed, as a general note, I think it might be worth thinking about whether you were measuring PTS-symptoms or simply acute stress symptoms. Indeed, there was yet no “post” when you collected your data as people were still working in the context of a continuing pandemic. This is not to say that high level of acute stress shouldn't be addressed via interventions.

Overall the discussion is excessively long and could be made more succinct.

Reply to R2#42: Thank you for this consideration. We slightly edited a sentence, to integrate at least partially the Reviewer's comment. The edited sentence reads as follows: “Moreover, potentially traumatic events are sudden and unexpected, by definition, with an impairment that is strong but temporary, and the psychological symptoms that we detected might represent normal reactions during a situation like the COVID-19 pandemic.”

We also shortened the Discussion in several places.

EDITORIAL COMMENTS

R2#43: Some sentences are generally too long and sometimes feel a bit convoluted.

Reply to R2#43: We have now made substantial changes to the manuscript based on all of the Reviewers' comments and we hope that our descriptions are now clearer.

I hope this feedback will be useful for the authors and again congratulations on collecting the data despite context. I wish you all the best with the manuscript.

Kind regards,

Alessandro Massazza

References

Charlson F, van Ommeren M, Flaxman A, Cornett J, Whiteford H, Saxena S. (2019). New WHO prevalence estimates of mental disorders in conflict settings: a systematic review and meta-analysis. *Lancet* **394**(10194):240-248. doi:10.1016/S0140-6736(19)30934-1.

WHO 2017. Scalable psychological interventions for people in communities affected by adversity
A new area of mental health and psychosocial work at WHO

Kisely S, Warren N, McMahon L, Dalais C, Henry I, Siskind D. (2020). Occurrence, prevention, and management of the psychological effects of emerging virus outbreaks on healthcare workers: rapid review and meta-analysis. *BMJ* **369**:m1642. Published 2020 May 5. doi:10.1136/bmj.m1642

King DW, King LA, McArdle JJ, Grimm K, Jones RT, Ollendick TH. (2006). Characterizing time in longitudinal trauma research. *J Trauma Stress* **19**(2):205-215. doi:10.1002/jts.20112

IASC (2020). Interim Briefing Note ADDRESSING MENTAL HEALTH AND PSYCHOSOCIAL ASPECTS OF COVID-19 OUTBREAK Version 1.5 February 2020 IASC Reference Group on Mental Health and Psychosocial Support in Emergency Settings

NICE (2018), Post-traumatic stress disorder. Nice guideline [NG116].

Iasc Guidelines on Mental Health and Psychosocial Support in Emergency Settings (2007).

Galea S, Maxwell AR, Norris F. (2008). Sampling and design challenges in studying the mental health consequences of disasters. *Int J Methods Psychiatr Res* **17** Suppl 2(Suppl 2):S21-S28. doi:10.1002/mpr.267

REVIEWER 4

The introduction, rationale and stated hypotheses are the same as the approved stage 1 submission, and the authors adhered to the registered procedures. The data collected by the authors adequately allowed them to test their proposed hypotheses. The unregistered exploratory statistical analyses seem justified.

R4#1: I feel that the discussion could be more concise and at times more objective (e.g. referring to organisations that chose not to participate as ‘less virtuous’ seems speculative).

Reply to R4#1: We have now shortened the Discussion, also in accordance with comments from other Reviewers. We replaced the term “virtuous” with “worker-oriented”, as in NCH where the situation turned particularly dramatic, organizations tended to keep relevant information from the public (e.g. by not reporting the actual number of COVID-19-related deaths). In the time window of our survey, we were also approached by workers who got wind of the initiative and asked (begged) for the possibility take part, independently of their organization. We had to decline, as this was a considerable deviation from our chosen methodology, but we have strong grounds to suspect that a much more dramatic picture would have emerged, had we followed this alternative path.

R4#2: The confidence intervals for the effect of gender on anxiety in the logistic regression model came close to overlapping with the null. Highlighting the strength of evidence for this particular effect would be useful.

Reply to R4#2: It is true that the effect of gender is near the null but since it is (a) within the specified bounds after being determined by the same method used for all other variables, (b) a variable that is significant consistently in all the main tests and (c) a variable that has been reported to be significant in similar studies for anxiety, we deem that this result is not controversial. If it had not been this particular variable, we would have to be much more cautious.

REVIEWER 5

In this Stage 2 Registered Report, the authors report the results of their preregistered analyses as well as some additional exploratory analyses. I have a small number of points to raise at this stage:

R5#1: It's good to see that the raw data and syntax files seem to be provided, but I couldn't open or check these as I don't have Stata. Ideally the syntax would be provided with an open source language (e.g., R). At the least the *data* should be available in a non-proprietary format (e.g. .csv).

Reply to R5#1: We have now included the data in Excel format. Please note that a trial version of Stata can be downloaded for free and this can be used to run the codes. We have maintained the scripts in their original format (*.do) as they can be read easily in any text editor (e.g. NotePad++). If you do not wish to download STATA to run the codes, we have now also included the output of the codes in separate *.txt files.

R5#2: Page 12, line 21, is the phrase “had an educational level *up* to high school (752 [70.1%])” correct? Surely at least some participants had some tertiary education?

Reply to R5#2: This depends on what it is meant by tertiary education. In the UK, both colleges and universities are sometimes included under the umbrella of tertiary education. In Italy, there is not a college system and 70% of our participants did not obtain a university degree (thus they only completed up to high school). To clarify this further we have modified the phrase to read: “had an educational level between primary and high school (752 [70.1%])”.

R5#3: Section 4.1.3 reports the actual preregistered data analysis; from what I can tell they follow the preregistered plan. However, the vast majority of the results section covers additional exploratory analyses. I can appreciate that exploratory analyses can sometimes be appropriate in a Registered Report, but they then need to be clearly justified, and techniques suitable for exploratory analysis employed. Here, the “exploratory” analyses rely heavily on statistical hypothesis testing (i.e., *p* values), even though the hypotheses tested weren't preregistered. There also isn't a very explicit rationale for the analyses: Why are these analyses so important to include that they need to take up most of the Results section, while *not* being important enough to have been preregistered? How can the reader be assured that these analyses are adequately powered, especially considering the small sample sizes in some cells? I would actually suggest that almost all of the exploratory analyses be removed, bar section 4.2.4 and the first two paragraphs of section 4.2.2 (which just report some relevant descriptive statistics).

Reply to R5#3: In the first round of reviews (R5#4), the Reviewer objected that our pre-registered analysis did not use inferential statistics, therefore s/he had deemed it not appropriate. In the current round, the Reviewer objects that inferential statistics should not be used in exploratory analyses. But this does not correspond to the reality of registered reports and “statistical hypothesis testing (i.e. *p* values)” can be “exploratory” (and it is appropriately used as such here). Equating preregistered with “*p* values” and exploratory with “not *p* values” is a simplification which would greatly reduce the scope and breadth of the registered report format. We designed our study to obtain an accurate prevalence estimate. Then, in line with the extant literature on the prevalence of mental health symptomatology, we have performed some extra investigation into what factors are related to our main prevalence result. Performing extra calculations would not tell us anything more than the *p*-values and confidence intervals that we have already produced (e.g. Hoenig & Heisey (2001), *The American Statistician*, 55, 19). The extra investigation has been included in a distinct section and clearly discussed as separate from the pre-registered and fully-powered result. Most of the space taken up by the exploratory analyses section is actually taken up by tables, which provide, in a succinct format, a useful overview of our data and will enable direct comparability with similar

studies. This is an important and new area that merits more than the publication of a couple of statistics without comment. Nowhere in the manuscript do we claim that our exploratory analysis is the final word on this subject (just like any statistical study), but for this study (the first of its kind for NCH in Italy) we do identify variables that have also been found to be important in other studies. Such results can help to inform us what specific subgroups should be offered preventative interventions in NCH and so they provide a good starting point for further developments. Taking them out would simply undermine our contribution.

R5#4: The fact that the logistic regression includes only those predictors which had a bivariate association with the outcome with $p < 0.1$ is especially questionable. Selecting analyses to report based on which coefficients are statistically significant will result in biased estimates, and is the exact practice that a registered report is designed to avoid.

Reply to 5#4: The practice of performing bivariate analyses prior to a logistic regression and selecting variables with p -values less than some (relatively large) number (typical values in the literature are 0.1 and 0.25) is a standard approach (e.g. Heeringa et al., *Applied Survey Data Analysis*, 2017, CRC Press). It is important to realise that the logistic regressions that we present are part of an *exploratory analysis*. In such an analysis, we are at liberty to pursue various methods as long as we present each step clearly for these methods to be repeatable. We have chosen an approach used by many distinguished practitioners in survey design and analysis, so our approach is not without foundation. Further, our results pick out a consistent picture – the importance of gender and COVID-19 contact. These variables have been found to be important in other studies and are not unrealistic candidates to be related to anxiety and/or PTSD.

As mentioned above, all of our data and codes are available for others to use for further analysis. If, for example, we adopt a different approach to the logistic regression for our main result and include all the variables in the analysis, we get:

Survey: Logistic regression

Number of strata	=	1	Number of obs	=	1,071
Number of PSUs	=	33	Population size	=	1,068.6736
			Design df	=	32
			F(9, 24)	=	2.73
			Prob > F	=	0.0241

result	Odds Ratio	Linearized Std. Err.	t	P> t	[95% Conf. Interval]	
job	1.209403	.1493141	1.54	0.133	.940489	1.555207
1.gender	2.217534	.494397	3.57	0.001	1.408132	3.492187
age	1.039327	.056939	0.70	0.486	.9295835	1.162027
education	1.161722	.1320002	1.32	0.196	.9216939	1.464259
daysworkedlastwk	1.016698	.0676995	0.25	0.805	.887742	1.164387
daysworkedprevwk	.9950412	.053957	-0.09	0.928	.8909868	1.111248
familyvisits	1.044878	.1738665	0.26	0.794	.744499	1.466448
covid19	1.68919	.3850424	2.30	0.028	1.061773	2.687356
ppe	1.064287	.215993	0.31	0.761	.7039259	1.609129
_cons	.1137052	.0718705	-3.44	0.002	.0313781	.4120353

Note: _cons estimates baseline odds.

As you can see, this change of procedure does not change our main finding.

In short, we apply a standard method which produces results that have a clear interpretation. Our approach is described in an exploratory analysis section and enough detail is given for it to be

reproduced. If others wish to use our data, extend our analysis or criticise it, we give them the opportunity.

R5#5: The study ultimately concludes a very high prevalence “43% for anxiety and/or PTSD symptoms of moderate-to-severe intensity”. In the first round of review I asked, “What *evidence* is there that scores above these cut-offs indicate “moderate to severe symptoms”?” This concern was never directly addressed; the authors said “These categories are based on values established in the literature [27, 28, 41, 63, 69, 70]”, but just giving citations to papers that have suggested or used these cut-offs before is not a very convincing rationale. In the discussion section, I would now suggest that the authors reflect on the evidence basis for these cut-offs – what **evidence** do we have that these thresholds have high sensitivity and specificity in the identification of moderate anxiety and or PTSD? Consider, for example, that if the specificity of the cut-offs is low, then the prevalence estimate reported could be a substantial overestimate.

Reply to 5#5: We thank the Reviewer for giving us the opportunity to clarify this point. In fact, the required information about available sensitivity and specificity of the questionnaires used was already included in our manuscript (please see section 3.2.2). However, with the occasion, we would also like the Reviewer to consider that specificity and sensitivity values can be deceitfully straightforward, as they depend heavily on the methodology employed and on the diagnosis against which they are measured (e.g. GAD-7 sensitivity and specificity are 89% and 82% for generalized anxiety disorder in primary care patients, but they are 66% and 81% for post-traumatic stress disorder in Spitzer et al., 2006; additionally, Rutter and Brown (2017) report 79.5% sensitivity and 44.7% specificity for generalized anxiety disorder in outpatients using the same cut-off as Spitzer et al.’s, 2006). They are of very limited value here as (a) they have not been tested in the population of interest (b) they are not being used for diagnostic purpose. In other words, there is no evidence that the available values apply to this population for diagnosable PTSD or anxiety disorders and, therefore, any evaluation of their possible impact on our prevalence estimate of subjective symptomatology is quite speculative. If we used them as a proxy for the diagnosis of a mental health condition, our prevalence could well be an “overestimate”. But it could also be an “underestimate” as sensitivity never reaches 100% with our cutoffs. Last but not least, we are not using these questionnaires as diagnostic tools and our prevalence estimate is not intended to return the proportion of individuals with a mental health condition. It provides an initial indication, based on an adequate sample size, on the proportion of individuals reporting symptomatology of potential clinical interest in the context of this emergency. It provides a starting point for preventative interventions and further clinical assessments, and our estimate should be compared with and interpreted in the context of similar epidemiological studies, as we have done.

R5#6: “Based on the above comparisons, we speculate that our prevalence *datum* may represent a conservative estimate of the psychological burden in Northern Italian NCH”: Is a “psychological burden” really an estimable quantity? The study certainly does *not* represent a conservative estimate of *diagnosable* PTSD or anxiety disorders – if anything, the opposite is true.

Reply to 5#6: Our statement should be read in the context where it is placed. Compared to similar studies (i.e. survey-based epidemiological studies), our estimate appears to be conservative (section 5.2).

Re: “psychological burden”. Actually, yes, it can be a measurable quantity, as any psychological construct, provided its scope is defined. Our intention here was to use it in general terms, as a mental heavy load carried by workers during the COVID-19 emergency (which we measure as anxiety and PTSD symptomatology). Since however, it seems that a group of Korean academics developed a scale to measure “psychological burden” in the workplace, and have provided an

operationalization for the construct (Kim et al. 2018), we have now replaced “psychological burden” with “self-reported psychological symptoms”.

Finally, we fully appreciate the Reviewer’s preoccupation but we are not claiming that 43% is the prevalence of individuals with a diagnosable mental health disorder.

R5#7: Section 5.4: The authors speculate on potential reasons for the higher prevalence of PTSD symptoms as opposed to anxiety symptoms. One possible explanation is simply that the cutoff for PTSD symptoms may have had lower specificity (i.e., that it was a more “lenient” cutoff).

Reply to 5#7: This is valid point but we respectfully refer the Reviewer to our previous Reply to point 5#5.

R5#8: Section 5.5: It’s good to see limitations identified clearly. It would be worth noting that the design of the study makes it impossible to determine the causal *effect* of COVID-19 on PTSD or anxiety symptoms.

Reply to 5#8: Nowhere in the manuscript do we claim that our study is suitable to test for causal effects. We have now added, as an extra note of caution, the Reviewer’s point in the Limitations section.

R5#9: Data accessibility statement: The link to supporting data and research materials is not working (I suspect the OSF project is still set to private)

Reply to 5#9: For review purposes, research materials are now provided at the following private link: https://osf.io/gmbasa/?view_only=50ddf0065efa48b7932167bbcde48b08. The OSF project will be made public upon manuscript acceptance.

References

Hoening & Heisey (2001). The Abuse of Power: The Pervasive Fallacy of Power Calculations for Data Analysis. *The American Statistician*, 55, 19.

Heeringa et al., *Applied Survey Data Analysis*, 2017, CRC Press.

Spitzer RL, Kroenke K, Williams JB, Löwe B. A brief measure for assessing generalized anxiety disorder: the GAD-7. *Arch Intern Med*. 2006 May 22;166(10):1092-7. doi: 10.1001/archinte.166.10.1092. PMID: 16717171.

Rutter LA, Brown TA. Psychometric Properties of the Generalized Anxiety Disorder Scale-7 (GAD-7) in Outpatients with Anxiety and Mood Disorders. *J Psychopathol Behav Assess*. 2017 Mar;39(1):140-146. doi: 10.1007/s10862-016-9571-9. Epub 2016 Sep 10. PMID: 28260835; PMCID: PMC5333929.

Kim et al. (2018). Developing a Basic Scale for Workers' Psychological Burden from the Perspective of Occupational Safety and Health. *Saf Health Work*. 2018 Jun; 9(2): 224–231.